# New Lower Bounds for Private Estimation and a Generalized Fingerprinting Lemma[*]

**Gautam Kamath**
Cheriton School of Computer Science
University of Waterloo
Waterloo, ON N2L 3G1, Canada
g@csail.mit.edu

**Argyris Mouzakis**
Cheriton School of Computer Science
University of Waterloo
Waterloo, ON N2L 3G1, Canada
amouzaki@uwaterloo.ca

**Vikrant Singhal**
Cheriton School of Computer Science
University of Waterloo
Waterloo, ON N2L 3G1, Canada
vikrant.singhal@uwaterloo.ca

## Abstract

We prove new lower bounds for statistical estimation tasks under the constraint of $(\varepsilon, \delta)$-differential privacy. First, we provide tight lower bounds for private covariance estimation of Gaussian distributions. We show that estimating the covariance matrix in Frobenius norm requires $\Omega(d^2)$ samples, and in spectral norm requires $\Omega(d^{3/2})$ samples, both matching upper bounds up to logarithmic factors. We prove these bounds via our main technical contribution, a broad generalization of the fingerprinting method [BUV14] to exponential families. Additionally, using the private Assouad method of Acharya, Sun, and Zhang [ASZ21], we show a tight $\Omega(d/(\alpha^2 \varepsilon))$ lower bound for estimating the mean of a distribution with bounded covariance to $\alpha$-error in $\ell_2$-distance. Prior known lower bounds for all these problems were either polynomially weaker or held under the stricter condition of $(\varepsilon, 0)$-differential privacy.

## 1 Introduction

The last several years have seen a surge of interest in algorithms for statistical estimation under the constraint of *Differential Privacy* (DP) [DMNS06]. We now have a rich algorithmic toolbox for private estimation of mean, covariance, and entire distributions, in a variety of settings.

While the community has enjoyed much success designing *algorithms* for estimation tasks, *lower bounds* have been much harder to come by, and consequently, the existing literature lacks a rigorous understanding of some core problems. Lower bounds against the strong privacy constraint of *pure* DP (i.e., $(\varepsilon, 0)$-DP) are generally not too challenging to prove, most often relying upon the well-known packing technique [HT10]. Such packing lower bounds are optimal in a broad range of settings (see, e.g., [BKSW19]), and thus are frequently effective at providing tight minimax sample complexity lower bounds.

However, pure DP is a very strong privacy constraint: in most practical circumstances, it suffices to only require the weaker privacy notion of *approximate* DP (i.e., $(\varepsilon, \delta)$-DP) [DKM+06], in which case packing lower bounds are no longer applicable. Proving lower bounds under approximate DP

---

[*]Authors are listed in alphabetical order.

36th Conference on Neural Information Processing Systems (NeurIPS 2022).

is much more challenging, and accordingly, the state of affairs is less satisfying. There exist only a couple of techniques which apply in this setting, including the fingerprinting method [BUV14] and differentially private analogues of Le Cam's method and Assouad's lemma [ASZ18, ASZ21]. These techniques can be quite brittle and thus unable to prove lower bounds for some fairly basic settings. For example, both techniques generally require that the underlying distribution has independent marginals and are thus ineffective at proving tight lower bounds for problems involving correlations such as private covariance estimation. Furthermore, the fingerprinting method often needs the distribution's marginals to have a precise functional form, such as Gaussian or Bernoulli. These restrictions make the approaches somewhat brittle for proving lower bounds which differ too much from existing results. Consequently, we lack approximate DP lower bounds for some fundamental settings which seem to qualitatively differ, including Gaussian covariance estimation and mean estimation of heavy-tailed distributions.

## 1.1 Our Results

We circumvent these barriers and fill a number of gaps in the literature by providing tight lower bounds for several estimation tasks under the constraint of approximate differential privacy. Specifically, we address problems including covariance estimation for Gaussians and mean estimation with heavy-tailed data. Our main technical tool for covariance estimation is a novel generalization of the fingerprinting method to exponential families, while we prove our mean estimation lower bound via DP Assouad's lemma.

Our first result is a lower bound for covariance estimation in Frobenius norm.[2]

**Theorem 1.1** (Informal). *For any $\alpha = \mathcal{O}(1)$, any $(\varepsilon, \delta)$-DP mechanism with $\varepsilon, \delta \in [0, 1]$, and $\delta \leq \mathcal{O}(\min\{1/n, d^2/(n\log(n/d^2))\})$ that takes $n$ samples from an arbitrary $d$-dimensional Gaussian $\mathcal{N}(0, \Sigma)$ and outputs $M(X)$ satisfying $\mathbb{E}_{X,M}\left[\|M(X) - \Sigma\|_F^2\right] \leq \alpha^2$ requires $n \geq \Omega\left(\frac{d^2}{\alpha\varepsilon}\right)$.*

We point out that the above lower bound also implies a lower bound for density estimation of Gaussians with known mean and unknown covariance (see Theorem 1.1 of [DMR18]). This nearly matches the $\widetilde{\mathcal{O}}(d^2)$ upper bound of [KLSU19] up to polylogarithmic factors.[3] Previously, $\Omega(d^2)$ lower bounds for Gaussian covariance estimation were only known under the more restrictive constraint of pure differential privacy [KLSU19, BKSW19].

Our result for Gaussian covariance estimation in Frobenius norm implies the following result for spectral estimation.

**Theorem 1.2** (Informal). *For any $\alpha = \mathcal{O}(1/\sqrt{d})$, any $(\varepsilon, \delta)$-DP mechanism with $\varepsilon, \delta \in [0, 1]$, and $\delta \leq \mathcal{O}(\min\{1/n, d^2/(n\log(n/d^2))\})$ that takes $n$ samples from an arbitrary $d$-dimensional Gaussian $\mathcal{N}(0, \Sigma)$ and outputs $M(X)$ satisfying $\mathbb{E}_{X,M}\left[\|M(X) - \Sigma\|_2^2\right] \leq \alpha^2$ requires $n \geq \Omega\left(\frac{d^{1.5}}{\alpha\varepsilon}\right)$.*

In Theorem 3.8 of [KLSU19], the authors give guarantees for an individual iteration of a private recursive preconditioner. In each iteration, the data is rescaled, resulting in constant-factor progress in privately reducing the condition number of the underlying covariance matrix to a constant. Identifying the preconditioning matrix necessitates performing spectral estimation of the underlying covariance matrix. Thus, the sample complexity upper bound of $\widetilde{\mathcal{O}}(d^{3/2})$ which comes up in the guarantees of the preconditioning steps is also an upper bound for spectral estimation for $\alpha = \mathcal{O}(1)$. This implies that our lower bound is tight up to polylogarithmic factors in the regime $\alpha = \mathcal{O}(1/\sqrt{d})$.

---

[2]Recall that the Mahalanobis norm of $M$ with respect to $\Sigma$ is $\|M\|_\Sigma = \left\|\Sigma^{-1/2}M\Sigma^{-1/2}\right\|_F$. For the lower bound constructions we consider, $\Omega(1)\,\mathbb{I} \preceq \Sigma \preceq \mathcal{O}(1)\,\mathbb{I}$. In this regime, the Mahalanobis and Frobenius distances are equivalent up to a constant factor. Thus, we will often use them interchangeably and our lower bound for Frobenius estimation also implies a lower bound for Mahalanobis estimation.

[3]We note that all our lower bounds in this work are stated in terms of mean squared error, while most of the upper bounds guarantee error at most $\alpha$ with probability $1 - \beta$ for some $\beta > 0$. In many natural cases (such as when the estimator's range is naturally bounded), lower bounds against MSE can be converted to constant probability statements via a boosting argument (outlined, e.g., in the proof of Theorem 6.1 of [KLSU19]). The details of such a conversion generally standard when applicable and we do not discuss it further here.

Note that the sample complexity of the same problem sans privacy constraints is $\Theta(d)$, and thus the cost of privacy is polynomial in the dimension $d$. This is in contrast to Gaussian mean estimation in $\ell_2$-norm and covariance estimation in Frobenius norm, which maintain their non-private sample complexities of $\mathcal{O}(d)$ and $\mathcal{O}(d^2)$, respectively.

$\Omega(d^{3/2})$-sample lower bounds for approximate DP covariance estimation in spectral norm were previously known for *worst-case* distributions [DTTZ14]. Our work relaxes this assumption significantly to the Gaussian case while maintaining the same lower bound. A priori, it was not clear that such a result was even true, as there are frequently gaps between the sample complexity of private estimation for worst-case versus well-behaved distributions for even basic settings, including distributions over the hypercube (see, e.g., Remark 6.4 of [BKSW19]).

This lower bound has implications for estimation tasks with scale-dependent error. For example, consider Gaussian mean estimation in Mahalanobis distance. A natural way to approach this problem is to first estimate the covariance matrix spectrally, and then estimate the mean after an appropriate rescaling (see, e.g., [KLSU19]). Our lower bound shows that any such approach must incur the $\Omega(d^{3/2})$ cost of spectral estimation, which is greater than the $\mathcal{O}(d)$ minimax sample complexity of the problem. Indeed, some recent works [BGS$^+$21, LKO21] manage to circumvent this roadblock by adopting more direct (but computationally inefficient) methods.

Finally, we prove new lower bounds for mean estimation of distributions with bounded second moments (the definition is in Appendix A).

**Theorem 1.3** (Informal). *For any $\alpha \leq 1$, any $(\varepsilon, \delta)$-DP mechanism with $\delta \leq \varepsilon$ that takes $n$ samples from an arbitrary distribution over $\mathbb{R}^d$ with second moments bounded by $1$ and outputs $M(X)$ satisfying $\mathbb{E}_{X,M}\left[\|M(X) - \mu\|_2^2\right] \leq \alpha^2$ requires $n \geq \Omega\left(\frac{d}{\alpha^2 \varepsilon}\right)$.*

Similar $\Theta(d/(\alpha^2 \varepsilon))$ upper and lower bounds were previously known for distributions with bounded second moments under the stricter constraint of pure differential privacy [BD14, KSU20, HKM21]. Also, a similar bound is shown in [KLZ21] for concentrated DP, but under a weaker moment assumption which involves only coordinate-wise projections. Our result shows that the same bound holds under the weaker constraint of approximate DP for distributions satisfying the stronger moment assumption, and thus no savings can be obtained by relaxing the privacy notion. In contrast to our other results, we show this lower bound via the DP Assouad method of [ASZ21], thus demonstrating the promise of different approaches for proving lower bounds for differentially private estimation.

## 1.2 Related Work

The fingerprinting technique for proving lower bounds was introduced by Bun, Ullman, and Vadhan [BUV14], which relied on the existence of *fingerprinting codes* [BS98, Tar08]. Since then, the technique has been significantly simplified and refined in a number of ways [SU17a, DSS$^+$15, BSU17, SU17b, KLSU19, CWZ19], including the removal of fingerprinting codes and distilling the main technical component into the *fingerprinting lemma*. Beyond the aforementioned settings of mean estimation of Gaussians and distributions over the hypercube [BUV14, SU15, DSS$^+$15, KLSU19], fingerprinting lower bounds have also been applied in settings including private empirical risk minimization [BST14] and private spectral estimation [DTTZ14]. Differentially private analogues of Fano, Le Cam, and Assouad were first considered in the local DP setting [DJW13], and more recently under the constraint of central DP [ASZ18, ASZ21].

On the upper bounds side, the most relevant body of work is that which focuses on mean and covariance estimation for distributions satisfying a moment bound, see, e.g., [BD14, KV18, BKSW19, BS19, KLSU19, CWZ19, KSU20, WXDX20, DFM$^+$20, BDKU20, AAK21, BGS$^+$21, KLZ21, HLY21, KMS$^+$21, AL21]. Our lower bounds nearly match these upper bounds for Gaussian covariance estimation [KLSU19] and heavy-tailed mean estimation [KSU20, HKM21]. Previously, these bounds were conjectured to be optimal, albeit on somewhat shaky grounds including evidence from lower bounds proven under pure DP or non-rigorous connections with lower bounds in related settings. Our results rigorously prove optimality and confirm these conjectures. Other works study private statistical estimation in different and more general settings, including mixtures of Gaussians [KSSU19, AAL21], graphical models [ZKKW20], discrete distributions [DHS15], and median estimation [AMB19, TVGZ20]. Some recent directions involve guaranteeing user-level

privacy [LSY$^+$20, LSA$^+$21], or a combination of local and central DP for different users [ADK19]. See [KU20] for further coverage of DP statistical estimation.

## 1.3 Organization

We start by providing our generalized fingerprinting technique in Section 2. Then we show its novel application for proving the lower bound for covariance estimation of Gaussians in Mahalanobis norm, followed by the lower bound for spectral norm estimation in Section 3. Finally, we sketch the proof of the lower bound for mean-estimation of heavy-tailed distributions in Section 4. We describe the notations and preliminaries for privacy and exponential families in Appendix A. After that, we detail the missing proofs from Sections 2, 3, and 4 in Appendices B, C, and D, respectively. Furthermore, we state facts from linear algebra and geometry in Appendix E, and facts from probability and statistics in Appendix F. Finally, in Appendix G, we include the proofs of older results obtained in [KLSU19], but using our generalized results in the language of exponential families.[4]

## 2 Fingerprinting Proofs for Exponential Families

This section is split into three parts. Section 2.1 provides an overview of existing fingerprinting proofs and motivation for our result, and introduces all the necessary notation. Then, in Section 2.2, our new fingerprinting lemma is stated. Finally, in Section 2.3, we give a general recipe for applying it.

### 2.1 Overview of Fingerprinting Proofs and Intuition for Our Result

At the heart of fingerprinting proofs lies the trade-off between accuracy and privacy. For simplicity, assume we have a univariate data-generating distribution and coming from a family parameterized by its mean $\mu$, which we want to estimate. In contrast to minimax lower bounds in classical statistics (which are commonly obtained by constructing hard testing instances), fingerprinting proofs assume the existence of a subset of the domain of $\mu$ (commonly an interval $[\pm R]$) and that a value is drawn $\mu \sim \mathcal{U}([\pm R])$. Then, $n$ independent samples $X_1, \ldots, X_n$ are drawn from $\mathcal{D}_\mu$. Now, given an estimator $f : \mathbb{R}^n \to [\pm R]$ for $\mu$, we consider the quantity $Z_i \coloneqq c(R, \mu)(f(X) - \mu)(X_i - \mu), \forall i \in [n]$, which measures the correlation between the output of $f$ and the $i$-th sample ($c(R, \mu) > 0$ is a scaling factor that exists for purely technical reasons). Defining $Z \coloneqq \sum_{i \in [n]} Z_i$, all proofs first show a lower bound $L$ on $\mathbb{E}[Z + (f(X) - \mu)^2]$, implying that our estimator cannot at the same time be accurate (with respect to the Mean-Squared-Error (MSE)) and exhibit low correlation with individual samples. This component of the proof is traditionally called the *fingerprinting lemma*.

Privacy comes up in the second step of the proof, where we upper-bound the correlation terms $\mathbb{E}[Z_i]$.

We assume that $f$ is a private mechanism with MSE at most $L/2$. Due to privacy, all correlation terms $Z_i$ should be upper-bounded by the same $U = U(\varepsilon, \delta)$. Applying this for each sample leads to an inequality of the form $nU \geq L/2 \iff n \geq L/(2U)$, so the only thing that remains is to identify the appropriate parameter range for $\delta$ to prove the desired bound.

Our work is motivated by a fundamental weakness of existing fingerprinting proofs. To prove lower bounds for high-dimensional distributions via fingerprinting, past works have assumed that the underlying distribution is a product distribution. This made it possible to work component-wise using techniques for single-dimensional distributions. This assumption is somewhat restrictive and makes it difficult to generalize the approach to settings with a richer correlation structure.

Our main technical contribution is providing a broad generalization of the celebrated fingerprinting method to exponential families (the definition is in Appendix A). Distributions belonging in the same exponential family are parameterized by their *natural parameter vector* $\eta$, instead of their mean $\mu$, for which it is uncommon to have a general closed-form expression. Given $n$ samples $X_1, \ldots, X_n$ from $p_\eta$, it suffices to know the values of the *sufficient statistics* $T(X_1), \ldots, T(X_n)$ instead of the samples themselves in order to estimate $\eta$ (see, e.g., [Jor10]). We denote the mean and covariance of $T$ by $\mu_T$ and $\Sigma_T$, respectively.

---

[4]The bounds we recover in Appendix G are all from [KLSU19]. These include one lower bound for mean estimation of binary product distributions, and one for Gaussian mean estimation. In the process of proving the latter, we identified a bug in the original proof from [KLSU19], which we correct in this manuscript. This resulted in a slightly different range for $\delta$ than the one given in the original theorem and proof.

Based on the above remarks, our fingerprinting lemma provides lower bounds for estimating $\eta$ instead of $\mu$. We assume the existence of vectors $\eta^{(1)}, \eta^{(2)}$ with $\eta_j^{(1)} \leq \eta_j^{(2)}, \forall j \in [k]$, and define the intervals $I_j := \left[\eta_j^{(1)}, \eta_j^{(2)}\right]$ for each coordinate $j \in [k]$. Assuming that the hyperrectangle $\bigotimes_{j \in [k]} I_j$ is a subset of $\mathcal{H}$, we draw a natural parameter vector $\eta$ uniformly at random from it. Observe that $I_j$ may not be centered around the origin, which poses a minor technical difficulty. To deal with this, we assume that our estimator does not estimate $\eta$ itself, but rather the deviation of $\eta$ from the *midpoint* $m := (\eta^{(1)} + \eta^{(2)})/2$. The two problems are equivalent because any estimator $f : S^n \to \bigotimes_{j \in [k]}[-(\eta_j^{(2)} - \eta_j^{(1)})/2, (\eta_j^{(2)} - \eta_j^{(1)})/2]$ for the latter problem can be transformed to an estimator for the former, simply by adding $m$ to it (and vice-versa). Defining $R$ to be the *width vector* $\eta^{(2)} - \eta^{(1)}$, the correlation between the estimate for $\eta_j$ and the sample $X_i$ becomes $Z_i^j := [R_j^2/4 - (\eta_j - m_j)^2][f_j(X) - (\eta_j - m_j)](T_j(X_i) - \mu_{T,j}), \forall i \in [n], j \in [k]$.

Observe that now, $Z_i^j$ measures the correlation between $f$ and the sufficient statistics of the samples. This is in contrast to older fingerprinting proofs, in which the $Z_i^j$ measure correlation between $f$ and the samples themselves. We consider this to be natural in the context of exponential families. Indeed, we know that, if we want to estimate the natural parameter vector of a distribution that belongs to an exponential family given a dataset $X = (X_1, \ldots, X_n)$, it suffices to know the values $T(X_1), \ldots, T(X_n)$ instead of the samples themselves (see [Jor10]). Thus, despite $f$ receiving $X$, correlation ends up being measured with respect to $T(X_i), i \in [n]$.

In addition to the above, we introduce three more correlation terms:

$$Z_i := \sum_{j \in [k]} Z_i^j = \sum_{j \in [k]} [R_j^2/4 - (\eta_j - m_j)^2][f_j(X) - (\eta_j - m_j)](T_j(X_i) - \mu_{T,j}),$$

$$Z^j := \sum_{i \in [n]} Z_i^j = [R_j^2/4 - (\eta_j - m_j)^2][f_j(X) - (\eta_j - m_j)] \sum_{i \in [n]} (T_j(X_i) - \mu_{T,j}),$$

$$Z := \sum_{i \in [n]} Z_i = \sum_{j \in [k]} Z^j.$$

We motivate each of the above. $Z_i$ describes the correlation between $X_i$ and our estimates for $\eta$. Moreover, $Z^j$ describes the correlation between our estimate for $\eta_j$ and $X$. Finally, $Z$ describes the total correlation between the output of $f$ and $X$. Our fingerprinting lemma for exponential families will involve upper and lower-bounding the quantity $\mathbb{E}_{\eta,X}\left[Z + \|f(X) - (\eta - m)\|_2^2\right]$.

## 2.2 The Main Lemma and Our Approach to Lower Bounds

We prove our new fingerprinting lemma in two parts. First, we assume we have a distribution $p_\eta$ that belongs to an exponential family and we want to estimate $\eta_j$ whereas $\eta_{-j}$ is fixed (Lemma 2.1). This setting is similar to the ones considered in [BSU17, KLSU19], which feature similar lemmas. We then leverage Lemma 2.1 to get a result when the goal is to estimate all of $\eta$ (Lemma 2.2).

**Lemma 2.1.** *Let $p_\eta$ be a distribution over $S \subseteq \mathbb{R}^d$ belonging to an exponential family $\mathcal{E}(T, h)$ with natural parameter vector $\eta \in \mathcal{H} \subseteq \mathbb{R}^k$. We assume that $I_j$ is non-degenerate, and that $\eta_j$ is randomly generated by drawing $\eta_j \sim \mathcal{U}(I_j)$, whereas $\eta_{-j}$ is fixed. Then, for any function $f_j : S^n \to [\pm R_j/2]$ that takes as input a dataset $X := (X_1, \ldots, X_n) \sim p_\eta^{\otimes n}$, we have:*

$$\mathbb{E}_{\eta_j,X}\left[Z^j + [f_j(X) - (\eta_j - m_j)]^2\right] \geq R_j^2/12.$$

*Proof.* Let $g_j : \bigotimes_{j \in [k]} I_j \to [\pm R_j/2]$ with $g_j(\eta) := \mathbb{E}_X[f_j(X)]$. $g_j$ can equivalently be written as:

$$g_j(\eta) = \int_{\mathbb{R}^d} \cdots \int_{\mathbb{R}^d} \left(\prod_{i \in [n]} p_\eta(x_i)\right) f_j(x_1, \ldots, x_n) \, dx_1 \ldots dx_n.$$

Observe that:

$$\left(\prod_{i\in[n]} p_\eta\left(x_i\right)\right) = \left(\prod_{i\in[n]} h\left(x_i\right)\right)\exp\left(\left\langle\eta, \sum_{i\in[n]} T\left(x_i\right)\right\rangle - nZ\left(\eta\right)\right)$$

$$\implies \frac{\partial}{\partial\eta_j}\left(\prod_{i\in[n]} p_\eta\left(x_i\right)\right) = \left(\prod_{i\in[n]} p_\eta\left(x_i\right)\right)\sum_{i\in[n]}\left(T_j\left(x_i\right) - \frac{\partial}{\partial\eta_j}\left(Z\left(\eta\right)\right)\right)$$

$$= \left(\prod_{i\in[n]} p_\eta\left(x_i\right)\right)\sum_{i\in[n]}\left(T_j\left(x_i\right) - \mu_{T,j}\right).$$

Calculating the partial derivative of $g_j$ with respect to $\eta_j$ and changing the order of differentiation and integration, we get:

$$\frac{\partial}{\partial\eta_j}\left(g_j\left(\eta\right)\right) = \int_{\mathbb{R}^d}\cdots\int_{\mathbb{R}^d}\left(\prod_{i\in[n]} p_\eta\left(x_i\right)\right) f_j\left(x_1,\ldots,x_n\right)\sum_{i\in[n]}\left(T_j\left(x_i\right) - \mu_{T,j}\right)\,dx_1\ldots dx_n$$

$$= \mathop{\mathbb{E}}_X\left[f_j\left(X\right)\sum_{i\in[n]}\left(T_j\left(X_i\right) - \mu_{T,j}\right)\right]$$

$$\overset{(a)}{=} \mathop{\mathbb{E}}_X\left[f_j\left(X\right)\sum_{i\in[n]}\left(T_j\left(X_i\right) - \mu_{T,j}\right)\right] - \mathop{\mathbb{E}}_X\left[\left(\eta_j - m_j\right)\sum_{i\in[n]}\left(T_j\left(X_i\right) - \mu_{T,j}\right)\right]$$

$$= \mathop{\mathbb{E}}_X\left[\left[f_j\left(X\right) - \left(\eta_j - m_j\right)\right]\sum_{i\in[n]}\left(T_j\left(X_i\right) - \mu_{T,j}\right)\right],$$

where $(a)$ relies on the observation that, conditioned on $\eta_j$, $\eta_j - m_j$ and $\sum_{i\in[n]}\left(T_j\left(X_i\right) - \mu_{T,j}\right)$ are independent and $\mathop{\mathbb{E}}_X\left[\sum_{i\in[n]}\left(T_j\left(X_i\right) - \mu_{T,j}\right)\right] = 0$.

We proceed to define the function $G_j\colon \bigotimes_{j\in[k]} I_j \to \mathbb{R}$ as:

$$G_j\left(\eta\right) := \mathop{\mathbb{E}}_X\left[Z^j\right] = \left[R_j^2/4 - \left(\eta_j - m_j\right)^2\right]\mathop{\mathbb{E}}_X\left[\left[f_j\left(X\right) - \left(\eta_j - m_j\right)\right]\sum_{i\in[n]}\left(T_j\left(X_i\right) - \mu_{T,j}\right)\right]$$

$$= \left[R_j^2/4 - \left(\eta_j - m_j\right)^2\right]\frac{\partial g_j}{\partial\eta_j}\left(\eta\right).$$

Calculating the expectation of $G_j$ with respect to $\eta_j$ yields:

$$\mathop{\mathbb{E}}_{\eta_j}\left[G_j\left(\eta\right)\right] = \mathop{\mathbb{E}}_{\eta_j}\left[\left[R_j^2/4 - \left(\eta_j - m_j\right)^2\right]\frac{\partial g_j}{\partial\eta_j}\left(\eta\right)\right]$$

$$= \frac{1}{|I_j|}\int_{\eta_j^{(1)}}^{\eta_j^{(2)}}\left[R_j^2/4 - \left(\eta_j - m_j\right)^2\right]\frac{\partial g_j}{\partial\eta_j}\left(\eta_1,\ldots,\eta_j,\ldots,\eta_k\right)\,d\eta_j$$

$$\overset{(a)}{=} \frac{1}{|I_j|}\left[R_j^2/4 - \left(\eta_j - m_j\right)^2\right]g_j\left(\eta_1,\ldots,\eta_j,\ldots,\eta_k\right)\Bigg|_{\eta_j=\eta_j^{(1)}}^{\eta_j=\eta_j^{(2)}}$$

$$+ 2\frac{1}{|I_j|}\int_{\eta_j^{(1)}}^{\eta_j^{(2)}}\left(\eta_j - m_j\right)g_j\left(\eta_1,\ldots,\eta_j,\ldots,\eta_k\right)\,dt$$

$$= 2\mathop{\mathbb{E}}_{\eta_j}\left[\left(\eta_j - m_j\right)g_j\left(\eta\right)\right].$$

where $(a)$ uses integration by parts that $m_j$ has been defined to be equal to $(\eta_j^{(1)} + \eta_j^{(2)})/2$.

Based on this, we remark that the inequality we wish to prove can be written in the form:

$$\mathop{\mathbb{E}}_{\eta_j, X} \left[ 2 \left( \eta_j - m_j \right) f_j \left( X \right) + \left[ f_j \left( X \right) - \left( \eta_j - m_j \right) \right]^2 \right] \geq R_j^2/12. \tag{1}$$

Appealing to linearity of expectation, we focus on the second term and get:

$$
\begin{aligned}
\mathop{\mathbb{E}}_{\eta_j, X} \left[ \left[ f_j \left( X \right) - \left( \eta_j - m_j \right) \right]^2 \right] &= \mathop{\mathbb{E}}_{\eta_j, X} \left[ f_j^2 \left( X \right) - 2 \left( \eta_j - m_j \right) f_j \left( X \right) + \left( \eta_j - m_j \right)^2 \right] \\
&\geq -2 \mathop{\mathbb{E}}_{\eta_j, X} \left[ \left( \eta_j - m_j \right) f_j \left( X \right) \right] + \mathop{\mathbb{E}}_{\eta_j} \left[ \left( \eta_j - m_j \right)^2 \right] \\
&= -2 \mathop{\mathbb{E}}_{\eta_j, X} \left[ \left( \eta_j - m_j \right) f_j \left( X \right) \right] + \mathop{\mathrm{Var}}_{\eta_j} \left( \eta_j \right) \\
&= -2 \mathop{\mathbb{E}}_{\eta_j, X} \left[ \left( \eta_j - m_j \right) f_j \left( X \right) \right] + R_j^2/12.
\end{aligned}
$$

Substituting this to (1) yields the desired result. ∎

We now generalize the previous lemma in the setting where all the components of $\eta$ are drawn independently and uniformly at random from intervals.

**Lemma 2.2.** *Let $p_\eta$ be a distribution over $S \subseteq \mathbb{R}^d$ belonging to an exponential family $\mathcal{E}\left(T, h\right)$ with natural parameter vector $\eta \in \mathcal{H} \subseteq \mathbb{R}^k$. We assume that $\eta$ is randomly generated by drawing independently $\eta_j \sim \mathcal{U}\left(I_j\right), \forall j \in [k]$. Then, for any function $f \colon S^n \to \bigotimes_{j \in [k]} \left[\pm R_j/2\right]$ that takes as input a dataset $X := \left(X_1, \ldots, X_n\right) \sim p_\eta^{\otimes n}$, we have:*

$$\mathop{\mathbb{E}}_{\eta, X} \left[ Z + \left\| f\left(X\right) - \left(\eta - m\right) \right\|_2^2 \right] \geq \left\| R \right\|_2^2 / 12.$$

The proof is merely a component-wise application of Lemma 2.1 and is deferred to Appendix B.1.

## 2.3 From the Lemma to Lower Bounds

We leverage Lemma 2.2 to give a general recipe for proving lower bounds for private estimation. The overall structure of our argument is similar to that of past fingerprinting proofs but, along the way, uses specific properties of exponential families to deal with the fact that we do not have a product distribution.

We assume the existence of an $(\varepsilon, \delta)$-DP mechanism $M \colon S^n \to \bigotimes_{j \in [k]} \left[\pm R_j/2\right]$ with $\varepsilon \in [0, 1]$ and $\delta \geq 0$ such that, for any distribution $p_\eta$ with $\eta \in \bigotimes_{j \in [k]} I_j$, we have for $X \sim p_\eta^{\otimes n}$:

$$\mathop{\mathbb{E}}_{X, M} \left[ \left\| M\left(X\right) - \left(\eta - m\right) \right\|_2^2 \right] \leq \alpha^2 \leq \left\| R \right\|_2^2 / 24. \tag{2}$$

For the above, Lemma 2.2 with $f \equiv M$ yields:

$$\mathop{\mathbb{E}}_{\eta, X, M} \left[ Z \right] = \mathop{\mathbb{E}}_{M} \left[ \mathop{\mathbb{E}}_{\eta, X} \left[ Z \right] \right] \geq \left\| R \right\|_2^2 / 12 - \mathop{\mathbb{E}}_{\eta, X, M} \left[ \left\| M\left(X\right) - \left(\eta - m\right) \right\|_2^2 \right] \geq \left\| R \right\|_2^2 / 24. \tag{3}$$

Thus, our intent is to upper-bound the LHS by a function of $n$ using the definition of privacy, which will allow us to identify a lower bound to get the desired accuracy. To do so, we first need to upper-bound $\mathop{\mathbb{E}}_{\eta, X, M} \left[ Z_i \right]$.

**Lemma 2.3.** *Conditioning on $\eta$, for any $\varepsilon \in [0, 1], \delta \geq 0$, and $T > 0$, it holds that:*

$$\mathop{\mathbb{E}}_{X, M} \left[ Z_i \right] \leq 2\delta T + 2\varepsilon \sqrt{ \mathop{\mathbb{E}}_{X_{\sim i}, M} \left[ s^\top \Sigma_T s \right] } + 2 \int_T^\infty \mathop{\mathbb{P}}_{X_i} \left[ \left\| T\left(X_i\right) - \mu_T \right\|_2 > (4t)/(\left\| R \right\|_\infty^3 \sqrt{k}) \right] dt,$$

*where $s \in \mathbb{R}^k$ with $s_j := \left[ R_j^2/4 - \left( \eta_j - m_j \right)^2 \right] \left[ M_j \left( X_{\sim i} \right) - \left( \eta_j - m_j \right) \right], \forall j \in [k]$.*

The proof involves splitting $Z_i$ in its positive and negative components and applying the definition of DP to each of them. The argument is standard and has appeared in previous works (e.g., see [SU17b]), but we need to apply specific properties of exponential families to deal with the fact that we don't have a product distribution. The proof is lengthy so we omit it and is deferred to Appendix B.2.

Combining Lemma 2.3 with (3) yields the main result of this section (see Appendix B for the full proof):

**Theorem 2.4.** *Let $p_\eta$ be a distribution over $S \subseteq \mathbb{R}^d$ belonging to an exponential family $\mathcal{E}(T, h)$ with natural parameter vector $\eta \in \mathcal{H} \subseteq \mathbb{R}^k$. Also, let $\eta^{(1)}, \eta^{(2)} \in \mathcal{H}$ and let $I_j := \left[\eta_j^{(1)}, \eta_j^{(2)}\right], \forall j \in [k]$ be a collection of intervals and $R := \eta^{(2)} - \eta^{(1)}, m := (\eta^{(1)} + \eta^{(2)})/2$ be the corresponding width and midpoint vectors, respectively. Assume that $\bigotimes_{j \in [k]} I_j \subseteq \mathcal{H}$ and that $\eta$ is drawn from the distribution $\mathcal{U}(\bigotimes_{j \in [k]} I_j)$. Moreover, assume that we have a dataset $X \sim p_\eta^{\otimes n}$ and an independently drawn point $X_i' \sim p_\eta$ and $X_{\sim i}$ denotes the dataset where $X_i$ has been replaced with $X_i'$. Finally, let $M: S^n \to \bigotimes_{j \in [k]} [\pm R_j/2]$ be an $(\varepsilon, \delta)$-DP mechanism with $\varepsilon \in [0, 1], \delta \geq 0$ with $\mathbb{E}_{X, M}\left[\|M(X) - (\eta - m)\|_2^2\right] \leq \alpha^2 \leq \|R\|_2^2/24$. Then, for any $T > 0$, it holds that:*

$$n\left\{2\delta T + 2\varepsilon \mathbb{E}_\eta\left[\sqrt{\mathbb{E}_{X_{\sim i}, M}[s^\top \Sigma_T s]}\right] + 2\int_T^\infty \mathbb{P}_{X_i}\left[\|T(X_i) - \mu_T\|_2 > \frac{4t}{\|R\|_\infty^3 \sqrt{k}}\right] dt\right\} \geq \frac{\|R\|_2^2}{24}.$$

In order to apply the above theorem, the main idea is to identify values for $T$ and $\delta$ such that $\delta T \geq 2\int_T^\infty \mathbb{P}_{X_i}\left[\|T(X_i) - \mu_T\|_2 > (4t)/(\|R\|_\infty^3 \sqrt{k})\right] dt$, and $3\delta T n \leq \frac{\|R\|_2^2}{48}$.

The reasoning behind this is that the first and third terms of (2.4) both depend on $T$, with the latter term becoming smaller as $T$ increases. Thus, balancing these two terms involves identifying a value for $T$ that is as small as possible and results in $\delta T$ dominating the other term. At the same time, we want the sum of those two terms to be smaller than the half of the RHS, so that we get a lower bound that is as tight as possible, leading to the constraint on $\delta$.

The above result in the inequality $n\mathbb{E}_\eta\left[\sqrt{\mathbb{E}_{X_{\sim i}, M}[s^\top \Sigma_T s]}\right] \geq \|R\|_2^2/(96\varepsilon)$.

Consequently, obtaining the desired sample complexity lower bounds boils down to upper-bounding the term $\mathbb{E}_{X_{\sim i}, M}\left[s^\top \Sigma_T s\right]$. In the applications we will see, the upper bounds will be worst-case in terms of $\eta$, so the outer expectation $\mathbb{E}_\eta[\cdot]$ will be largely ignored.

We point the reader to the end of Appendix B for three remarks. The first notes the rotationally invariant nature of Theorem 2.4 with respect to the parameter space $\mathcal{H}$ (Remark B.2), the second details how to upper-bound the last term of (2.4) when $\delta$ is $> 0$ (Remark B.3), and the third Theorem 2.4 for $\delta = 0$ (Remark B.4).

## 3 Lower Bounds for Private Gaussian Covariance Estimation

Here, we provide lower bounds for covariance estimation of high-dimensional Gaussians with respect to the Mahalanobis and spectral norms under the constraint of $(\varepsilon, \delta)$-DP.

### 3.1 Estimation with Respect to the Mahalanobis Norm

In this section, we characterize the sample complexity of private Gaussian covariance estimation with respect to the Mahalanobis norm under $(\varepsilon, \delta)$-DP. Previous lower bounds either assumed a stricter notion of privacy (e.g., $\varepsilon$-DP - see [KLSU19]) or exhibited significant gaps with known upper bounds (see Section 1.1.1 of [AAK21] for further discussion). We start by stating our main result.

**Theorem 3.1.** *There exists a distribution $\mathcal{D}$ over covariance matrices $\Sigma \in \mathbb{R}^{d \times d}$ with $\mathbb{I} \preceq \Sigma \preceq 2\mathbb{I}$ such that, given $\Sigma \sim \mathcal{D}$ and $X \sim \mathcal{N}(0, \Sigma)^{\otimes n}$, for any $\alpha = \mathcal{O}(1)$ and any $(\varepsilon, \delta)$-DP mechanism $M: \mathbb{R}^{n \times d} \to \mathbb{R}^{d \times d}$ with $\varepsilon, \delta \in [0, 1]$, and $\delta \leq \mathcal{O}(\min\{1/n, d^2/(n\log(n/d^2))\})$ that satisfies $\mathbb{E}_{X, M}\left[\|M(X) - \Sigma\|_\Sigma^2\right] \leq \alpha^2$, it must hold that $n \geq \Omega\left(\frac{d^2}{\alpha \varepsilon}\right)$.*

Instead of giving the full proof, we give a sketch. The full argument can be found in Appendix C.1.

*Proof Sketch.* The outline of the proof involves (1) writing the Gaussian distribution $\mathcal{N}(0, \Sigma)$ as an exponential family, (2) defining a distribution from which $\eta$ is drawn, (3) showing that estimating the deviation of the natural parameter vector from the midpoint $m$ reduces to covariance estimation (so lower bounds for the former also apply for the latter), (4) upper-bounding the term $\mathbb{E}\left[s^{\top} \Sigma_T s\right]$, and (5) applying Theorem 2.4 as described at the end of the previous section.

Step (1) is non-trivial because the standard way to write $\mathcal{N}(0, \Sigma)$ as an exponential family involves setting $\eta$ to be equal to the *canonical flattening* of the *precision matrix* $\left(\Sigma^{-1}\right)^{\flat}$ (Fact F.3). However, we cannot use this representation, because $\Sigma^{-1}$ is symmetric, preventing the components of $\eta$ from being generated independently, as is required by Theorem 2.4. Thus, we consider an alternative parameterization. This involves identifying an upper-triangular matrix $U$ such that $\Sigma^{-1} = U + U^{\top}$ and setting the parameter vector to be equal to $2U^{\flat}$. The two parameterizations are equivalent (see Lemma C.1).

Step (2) employs a variant of a construction that has appeared in [KLSU19] (see Algorithm 1). The process consists of sampling the elements of the precision matrix uniformly at random from intervals of width $1/(2d)$. For the diagonal elements, these intervals are centered at $3/4$, whereas for the non-diagonal elements they are centered at the origin. The process is analyzed in Lemma C.2.

Step (3) requires us to describe an algorithm which, given oracle access to $(\varepsilon, \delta)$-DP estimate $\widehat{\Sigma}$ of $\Sigma$ that is $\alpha$-accurate in Mahalanobis norm, produces an $(\varepsilon, \delta)$-DP estimate of the natural parameter vector that is $\mathcal{O}(a)$-accurate in $\ell_2$-norm. The idea is to calculate $\widehat{\Sigma}^{-1}$, obtain a projection $\widetilde{\Sigma}^{-1}$ of it onto the support of the output of the algorithm described in step (2), find an upper-triangular matrix $\widetilde{U}$ such that $\widetilde{\Sigma}^{-1} = U + U^{\top}$ and output $2\widetilde{U}^{\flat} - m$. For the full algorithm, see Algorithm 2, and for the propositions that formally establish the reduction see Lemma C.4 and Corollary C.5.

Step (4) is given in Lemma C.7. The calculation involves appealing to a technical result from [DKK+16] (Proposition C.6).

Step (5) requires us to upper-bound the tail of $\|T(X_i) - \mu_T\|_2$. This is done in Lemma C.8. The strategy we follow is the one described in Remark B.3. We use a net-based argument (see Definition E.1), and an application of the Hanson-Wright inequality (see Fact F.7). Then, we proceed to identify the appropriate range for $\delta$ and obtain our sample complexity lower bound by working as described at the end of Section 2.3. ∎

## 3.2 Estimation with Respect to the Spectral Norm

Here, we prove a lower bound for covariance estimation of high-dimensional Gaussians in spectral norm under the constraint of approximate DP. Employing a reduction-based approach from Mahalanobis estimation to spectral estimation, we directly leverage the results of Section 3.1.

**Theorem 3.2.** *There exists a distribution $\mathcal{D}$ over covariance matrices $\Sigma \in \mathbb{R}^{d \times d}$ with $\mathbb{I} \preceq \Sigma \preceq 2\mathbb{I}$ such that, given $\Sigma \sim \mathcal{D}$ and $X \sim \mathcal{N}(0, \Sigma)^{\otimes n}$, for any $\alpha = \mathcal{O}(1/\sqrt{d})$ and any $(\varepsilon, \delta)$-DP mechanism $M: \mathbb{R}^{n \times d} \to \mathbb{R}^{d \times d}$ with $\varepsilon, \delta \in [0, 1]$, and $\delta \leq \mathcal{O}(\min\{1/n, d^2/(n \log(n/d^2))\})$ that satisfies*
$$\mathbb{E}_{X,M}\left[\left\|\Sigma^{-\frac{1}{2}}(M(X) - \Sigma)\Sigma^{-\frac{1}{2}}\right\|_2^2\right] \leq \alpha^2, \text{ it must hold that } n \geq \Omega\left(\frac{d^{1.5}}{\alpha \varepsilon}\right).$$

The proof is a reduction via the property $\|A\|_F \leq \sqrt{d}\|A\|_2$. We defer it to Appendix C.2.

# 4 Lower Bound for Private Heavy-Tailed Mean Estimation

In this section, we state the lower bound for mean-estimation of distributions with bounded second moment with respect to the $\ell_2$ norm and sketch its proof, but defer the details to Appendix D.

**Theorem 4.1.** *Let $\mathcal{D}$ be a distribution over $\mathbb{R}^d$ with second moments bounded by $1$ and unknown mean $\mu$. Then, for any $\alpha \leq 1$, any $(\varepsilon, \delta)$-DP mechanism with $\delta \leq \varepsilon$ that takes $X \sim \mathcal{D}^{\otimes n}$ as input and outputs $M(X)$ satisfying $\mathbb{E}_{X,M}\left[\|M(X) - \mu\|_2^2\right] \leq \alpha^2$ requires $n \geq \Omega\left(\frac{d}{\alpha^2 \varepsilon}\right)$.*

*Proof Sketch.* We use the DP Assouad lemma of [ASZ21]. To apply the method, we construct a family of distributions indexed by the points of the binary hypercube $\mathcal{E}_d \coloneqq \{-1, 1\}^d$. Define $p = (2\alpha^2)/d$ and $t = \sqrt{d}/(\sqrt{2}\alpha)$. Then for each $v \in \mathcal{E}_d$, we define a distribution $\mathcal{D}_v$ over $\mathbb{R}^d$, such that for $X \sim \mathcal{D}_v$, the following holds.

$$X_i = \begin{cases} v_i t, & \text{with probability } p \\ 0, & \text{with probability } 1 - p \end{cases}, \forall i \in [d].$$

It can be shown that the second moment of $\mathcal{D}_v$ is at most $pt^2 = 1$.

For $u, v \in \mathcal{E}_d$, we define the loss function between $\mathcal{D}_u$ and $\mathcal{D}_v$ to be $\ell(\theta(\mathcal{D}_u), \theta(\mathcal{D}_v)) = \|\theta(\mathcal{D}_u) - \theta(\mathcal{D}_v)\|_2^2$. This implies that (as defined in Lemma A.4) $\tau = 2p^2 t^2 = 2p$. On rearranging the terms in Lemma A.4, the above gives us $D \geq 0.04/(\varepsilon + \delta)$.

The final task is to show that there exists a coupling $(X, Y)$ between the mixture distributions $p_{+i}$ and $p_{-i}$ (as defined in Lemma A.4) with the appropriate value of $D = \mathbb{E}[d_{\text{Ham}}(X, Y)]$. Here, we just construct such a coupling, but verify the bound on $D$ in the appendix. Let $(X, Y)$ be as follows. For every row $X_j$ in $X$, if $X_j^i \neq 0$, then $Y_j^i = -X_j^i$, otherwise $Y_j^i = X_j^i$. For all other coordinates $k \neq i$, $Y_j^k = X_j^k$. We show that in this case, $D = np$. This gives us $D/p = n \geq d/[50(\varepsilon + \delta)\alpha^2)]$, which proves the theorem. $\blacksquare$

## 5 Conclusions and Open Problems

In this work, we gave nearly tight lower bounds for private covariance estimation with respect to the Frobenius and spectral norms under the Gaussian distribution and for mean estimation with respect to the $\ell_2$-norm under distributions with bounded second moments. This constitutes important progress in the lower bounds literature under approximate differential privacy. On a technical level, we significantly generalized the fingerprinting method. An obvious question is whether the fingerprinting method can be further generalized and what are its limitations. A second obvious question is to generalize Theorem 4.1 for higher-order moments.

Another question involves expanding upon the results of Theorems 3.1 and 3.2 to hold in the regimes $\alpha = \mathcal{O}(\sqrt{d})$ and $\alpha = \mathcal{O}(1)$, respectively. For Frobenius estimation, the condition $\alpha = \mathcal{O}(1)$ is a consequence of the distribution over covariance matrices that we used to prove the result, whereas for spectral estimation the condition $\alpha = \mathcal{O}(1/\sqrt{d})$ is an artifact of our reduction-based approach. Especially in the latter case, it would be interesting to come up with a "direct" way to obtain the lower bound, instead of reducing from Frobenius estimation.

Finally, an interesting open question may be to obtain lower bounds for the problems considered here in the *high-probability regime*. Specifically, the guarantees in our lower bounds are expressed in terms of the MSE and in Section 1.1 we sketched the argument to convert them to constant probability guarantees. It would be interesting to come up with techniques to prove tight lower bounds when the probability of success is $1 - \beta$ for $\beta \in (0, 1)$. This was done in the context of pure differential privacy in [HKM21].

## Acknowledgments

The authors would like to thank Jonathan Ullman for helpful feedback on the manuscript, Haoshu Xu for identifying a bug in the proof of Lemma 2.3 which affected how the range of $\delta$ is identified in Theorems 3.1 and G.4, as well as the anonymous reviewers at NeurIPS for their comments.

GK was supported by an NSERC Discovery Grant, an unrestricted gift from Google, and a University of Waterloo startup grant. AM was supported by an NSERC Discovery Grant and a David R. Cheriton Graduate Scholarship. VS was supported by an NSERC Discovery Grant.

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
