# A Preliminaries

**General Notation.** We use the notations: $[n] \coloneqq \{1, 2, \ldots, n\}, [a \pm R] \coloneqq [a - R, a + R]$, and $\mathbb{R}_+ \coloneqq [0, \infty)$. Given any distribution $\mathcal{D}$, $\mathcal{D}^{\otimes n}$ denotes the *product measure* where each *marginal distribution* is $\mathcal{D}$. Thus, if we are given $n$ independent samples from $\mathcal{D}$, we write $(X_1, \ldots, X_n) \sim \mathcal{D}^{\otimes n}$. Also, depending on the context, we may use capital Latin characters like $X$ to denote either an individual sample from a distribution or a collection of samples $X \coloneqq (X_1, \ldots, X_n)$. To denote the $j$-th component of a vector, we will use either a subscript (e.g., $X_j$, if the vector is $X$) or a superscript (e.g., $X_i^j$, if the vector is $X_i$). The convention used will be clear depending on the context. The *unit $\ell_2$-sphere* in $d$-dimensions that is centered at the origin is denoted by $\mathbb{S}^{d-1}$. Finally, we give the following definition for distributions with bounded $k$-th moments:

**Definition A.1.** Let $\mathcal{D}$ be a distribution over $\mathbb{R}^d$. We say that $\mathcal{D}$ has $k$-th moments bounded by $C$ if

$$\underset{X \sim \mathcal{D}}{\mathbb{E}} \left[ \left| \left\langle X - \underset{X \sim \mathcal{D}}{\mathbb{E}} [X], v \right\rangle \right|^k \right] \leq C, \forall v \in \mathbb{S}^{d-1}.$$

**Linear Algebra Preliminaries.** For a vector $v \in \mathbb{R}^d$, $v_i$ refers to its $i$-th component and $v_{-i} \in \mathbb{R}^{d-1}$ describes the vector one gets by removing the $i$-th component from $v$. We use $\vec{1}_d$ to refer to the all 1s vector in $\mathbb{R}^d$. For a pair of vectors $x, y \in \mathbb{R}^d$, we write $x \leq y$ if and only if $x_i \leq y_i, \forall i \in [n]$. Given a matrix $M \in \mathbb{R}^{d \times d}$, we denote the element at the $i$-th row and the $j$-th column by $M_{ij}$ or $(M)_{ij}$. Also, we denote the *canonical flattening* operation of $M$ into a vector by $v = M^\flat \in \mathbb{R}^{d^2}$ and the inverse operation by $M = v^\# \in \mathbb{R}^{d \times d}$. We denote its *spectral norm* by $\|M\|_2 \coloneqq \sup_{v \in \mathbb{S}^{d-1}} \|Mv\|_2$. For symmetric matrices $M$, this reduces to $\|M\|_2 = \sup_{v \in \mathbb{S}^{d-1}} |v^\top M v|$, which is known to be equal to the largest eigenvalue of $M$ (in absolute value). Also, we denote the *trace* of $M$ by $\mathrm{tr}(M) \coloneqq \sum_{i \in [d]} M_{ii}$, and its *Frobenius norm* by $\|M\|_F \coloneqq \sqrt{\mathrm{tr}(M^\top M)} = \sqrt{\sum_{i=1}^d \sum_{j=1}^d M_{ij}^2} = \|M^\flat\|_2$. For symmetric matrices, the latter is equal to the square root of the sum of the squares of its eigenvalues. Moreover, we consider the *Mahalanobis norm* (with respect to a non-degenerate covariance matrix $\Sigma$), both for vectors $\left( \|x\|_\Sigma \coloneqq \left\| \Sigma^{-\frac{1}{2}} x \right\|_2 \right)$ and matrices $\left( \|M\|_\Sigma \coloneqq \left\| \Sigma^{-\frac{1}{2}} M \Sigma^{-\frac{1}{2}} \right\|_F \right)$. Furthermore, we use the notation $\langle \cdot, \cdot \rangle$ to denote *inner products*, either between vectors $x, y \in \mathbb{R}^d$ $\left( \langle x, y \rangle \coloneqq x^\top y \right)$ or between matrices $A, B \in \mathbb{R}^{d \times d}$ $\left( \langle A, B \rangle \coloneqq \mathrm{tr}(AB^\top) \right)$. The inner product of a pair of $d \times d$ matrices $A, B$ can be reduced to an inner product between vectors in $\mathbb{R}^{d^2}$ via the identity $\langle A, B \rangle = \langle A^\flat, B^\flat \rangle$. Additionally, given a pair of symmetric matrices $A, B \in \mathbb{R}^d$, we write $A \succeq B$ if and only if $x^\top (A - B) x \geq 0, \forall x \in \mathbb{R}^d$. Finally, given matrices $A \in \mathbb{R}^{n \times m}$ and $B \in \mathbb{R}^{k \times \ell}$, the *Kronecker product* of $A$ with $B$ is denoted by $A \otimes B \in \mathbb{R}^{(nk) \times (m\ell)}$. More linear algebra facts appear in Appendix E.

## A.1 Privacy Preliminaries

We define differential privacy and state its closure under post-processing property.

**Definition A.2** (Differential Privacy (DP) [DMNS06])**.** A randomized algorithm $M \colon \mathcal{X}^n \to \mathcal{Y}$ satisfies $(\varepsilon, \delta)$-differential privacy $((\varepsilon, \delta)$-DP) if for every pair of neighboring datasets $X, X' \in \mathcal{X}^n$ (i.e., datasets that differ in exactly one entry), we have:

$$\underset{M}{\mathbb{P}} [M(X) \in Y] \leq e^\varepsilon \cdot \underset{M}{\mathbb{P}} [M(X') \in Y] + \delta, \forall Y \subseteq \mathcal{Y}.$$

When $\delta = 0$, we say that $M$ satisfies $\varepsilon$-differential privacy or pure differential privacy.

**Lemma A.3** (Post Processing [DMNS06])**.** *If $M \colon \mathcal{X}^n \to \mathcal{Y}$ is $(\varepsilon, \delta)$-DP, and $P \colon \mathcal{Y} \to \mathcal{Z}$ is any randomized function, then the algorithm $P \circ M$ is $(\varepsilon, \delta)$-DP.*

Finally, we discuss a recent technique from [ASZ21] for proving lower bounds for estimation problems under approximate DP. It is a private version of the well-known Assouad Lemma [Yu97]. Let $\mathcal{P}$ be a family of distributions over $\mathcal{X}^n$, where $\mathcal{X}$ is the data universe and $n$ is the number of

samples. Let $\theta \colon \mathcal{P} \to \Theta$ be the quantity associated with the distribution that we want to estimate, and let $\ell \colon \Theta \times \Theta \to \mathbb{R}_+$ be the pseudo-metric (loss function) for estimating $\theta$. We define the risk of an estimator $\hat{\theta} \colon \mathcal{X}^n \to \Theta$ as $\max\limits_{p \in \mathcal{P}} \mathbb{E}\limits_{X \sim p^{\otimes n}} \left[ \ell \left( \hat{\theta}(X), \theta(p) \right) \right]$. The minimax risk of $(\varepsilon, \delta)$-DP estimators for a statistical task is defined as:

$$R(\mathcal{P}, \ell, \varepsilon, \delta) \coloneqq \min_{\hat{\theta} \text{ is } (\varepsilon, \delta)\text{-DP}} \max_{p \in \mathcal{P}} \mathbb{E}_{X \sim P^{\otimes n}} \left[ \ell \left( \hat{\theta}(X), \theta(p) \right) \right].$$

Let $\mathcal{V} \subseteq \mathcal{P}$ be a subset of distributions indexed by the points in the hypercube $\mathcal{E}_d = \{-1, 1\}^d$. Suppose there exists a $\tau \in \mathbb{R}$, such that for all $u, v \in \mathcal{E}_d$, $\ell\left(\theta(p_u), \theta(p_v)\right) \geq 2\tau \cdot \sum\limits_{i=1}^{d} \mathbb{1}\{u_i \neq v_i\}$. For every coordinate $i \in [d]$, define the following mixture distributions:

$$p_{+i} \coloneqq \frac{2}{|\mathcal{E}_d|} \sum_{e \in \mathcal{E}_d \,:\, e_i = 1} p_e \text{ and } p_{-i} \coloneqq \frac{2}{|\mathcal{E}_d|} \sum_{e \in \mathcal{E}_d \,:\, e_i = -1} p_e.$$

The DP Assouad's method provides a lower bound for the minimax risk by considering the problem of distinguishing between $p_{+i}$ and $p_{-i}$.

**Lemma A.4** (DP Assouad's Method [ASZ21]). *For all $i \in [d]$, let $\phi_i \colon \mathcal{X}^n \to \{-1, 1\}$ be a binary classifier. Then we have the following:*

$$R(\mathcal{P}, \ell, \varepsilon, \delta) \geq \frac{\tau}{2} \cdot \sum_{i=1}^{d} \min_{\phi_i \text{ is } (\varepsilon, \delta)\text{-}DP} \left( \mathbb{P}_{X \sim p_{+i}^{\otimes n}} [\phi_i(X) \neq 1] + \mathbb{P}_{X \sim p_{-i}^{\otimes n}} [\phi_i(X) \neq -1] \right).$$

*Moreover, if $\forall i \in [d]$, there exists a coupling $(X, Y)$ between $p_{+i}$ and $p_{-i}$ with $\mathbb{E}\left[d_{\mathrm{Ham}}(X, Y)\right] \leq D$, then:*

$$R(\mathcal{P}, \ell, \varepsilon, \delta) \geq \frac{d\tau}{2} \cdot \left( 0.9 e^{-10\varepsilon D} - 10\delta D \right).$$

## A.2 Exponential Families

In this work, we make frequent use of exponential families due to their expressive power. This gives us a unified way of dealing with multiple commonly used distributions (e.g., Gaussians, Bernoullis). We first state the definition of exponential families.

**Definition A.5** (Exponential Family). Let $S \subseteq \mathbb{R}^d$ and $h \colon S \to \mathbb{R}_+, T \colon S \to \mathbb{R}^k$. Given $\eta \in \mathbb{R}^k$, we say that the density function $p_\eta$ belongs to the *exponential family* $\mathcal{E}(T, h)$ if its density can be written in the form:

$$p_\eta(x) = h(x) \exp\left( \eta^\top T(x) - Z(\eta) \right), \forall x \in S,$$

where:

$$Z(\eta) = \ln \left( \int\limits_S h(x) \exp\left( \eta^\top T(x) \right) \, dx \right).$$

The functions $h$ and $T$ are referred to as the *carrier measure* and the *sufficient statistics* of the family, respectively. Additionally, $Z$ is known as the *log-partition function* and $\eta$ is the *natural parameter vector*. Finally, we denote the *range of natural parameters* by $\mathcal{H} \subseteq \mathbb{R}^k$, which is the set of values of $\eta$ for which the log-partition function is well-defined ($Z(\eta) < \infty$).[1]

We appeal to the following properties of exponential families.

**Proposition A.6** (see [Jor10]). *Let $\mathcal{E}(T, h)$ be an exponential family parameterized by $\eta \in \mathbb{R}^k$. Then, for all $\eta$ in the interior of $\mathcal{H}$, the following hold:*

1. *The mean of the sufficient statistics is:*

$$\mu_T \coloneqq \mathbb{E}_{X \sim p_\eta} [T(X)] = \nabla_\eta Z(\eta).$$

---

[1] The definition is phrased in terms of continuous distributions, but applies to discrete distributions as well. Indeed, $p_\eta$ will be a *probability mass function*, whereas the integral will be replaced by a sum in the definition of $Z$. We gave this version of the definition to avoid measure-theoretic notation, since it does not add much to our results.

2. *The covariance matrix of the sufficient statistics is:*

$$\Sigma_T := \operatorname*{\mathbb{E}}_{X \sim p_\eta} \left[ (T(X) - \mu_T)(T(X) - \mu_T)^\top \right] = \nabla_\eta^2 Z(\eta).$$

3. *For all $s \in \mathbb{R}^k$ such that $\eta + s \in \mathcal{H}$, the MGF of the sufficient statistics is:*

$$\operatorname*{\mathbb{E}}_{X \sim p_\eta} \left[ \exp \left( s^\top T(X) \right) \right] = \exp \left( Z(\eta + s) - Z(\eta) \right).$$

4. *The natural parameter range $\mathcal{H}$ is a convex set, and the log-partition function $Z$ is convex.*

For a number of facts from probability and statistics (both related and unrelated to exponential families), we refer the reader to Appendix F.

## B  Omitted Proofs from Section 2

### B.1  Omitted Proofs from Section 2.2

*Proof of Lemma 2.2.* Without loss of generality, we assume that all the intervals $I_j$ are non-degenerate. We have:

$$
\begin{aligned}
\operatorname*{\mathbb{E}}_{\eta, X} \left[ Z + \| f(X) - (\eta - m) \|_2^2 \right] &= \operatorname*{\mathbb{E}}_{\eta, X} \left[ \sum_{j \in [k]} \left\{ Z^j + [f_j(X) - (\eta_j - m_j)]^2 \right\} \right] \\
&= \sum_{j \in [k]} \operatorname*{\mathbb{E}}_{\eta, X} \left[ Z^j + [f_j(X) - (\eta_j - m_j)]^2 \right] \\
&= \sum_{j \in [k]} \operatorname*{\mathbb{E}}_{\eta_{-j}} \left[ \operatorname*{\mathbb{E}}_{\eta_j, X} \left[ Z^j + [f_j(X) - (\eta_j - m_j)]^2 \right] \right] \\
&\overset{(a)}{\geq} \sum_{j \in [k]} \operatorname*{\mathbb{E}}_{\eta_{-j}} \left[ \frac{R_j^2}{12} \right] \\
&= \frac{\| R \|_2^2}{12},
\end{aligned}
$$

where $(a)$ is by a direct application of Lemma 2.1.

If there exists a degenerate interval $I_j$, we have $Z^j = f_j(X) - (\eta_j - m_j) = R_j = 0$. Thus, it suffices to apply the same argument as before for the non-degenerate intervals. ∎

### B.2  Omitted Proofs from Section 2.3

This appendix is dedicated to proving Lemma 2.3 and Theorem 2.4.

To prove the former, in addition to the correlation terms defined in Section 2.1, we introduce one more correlation term. Let $X_i'$ be an independently drawn sample from $p_\eta$ and let $X_{\sim i}$ be the dataset that is the same as $X$ with $X_i$ replaced by $X_i'$. We define:

$$s_j := \left[ \frac{R_j^2}{4} - (\eta_j - m_j)^2 \right] [M_j(X_{\sim i}) - (\eta_j - m_j)], \forall j \in [k], \tag{1}$$

$$\widetilde{Z}_i := \sum_{j \in [k]} s_j (T_j(X_i) - \mu_{T,j}). \tag{2}$$

This measures the effect that replacing an individual sample has on the output of the estimator. Observe that, conditioning on $\eta$, $X_{\sim i}$ and $X_i$ are independent. This implies that $\operatorname*{\mathbb{E}}_{X_{\sim i}, X_i, M} \left[ \widetilde{Z}_i \right] = 0$ and $\operatorname*{Var}_{X_{\sim i}, X_i, M} \left( \widetilde{Z}_i \right) = \operatorname*{\mathbb{E}}_{X_{\sim i}, X_i, M} \left[ \widetilde{Z}_i^2 \right]$. Additionally, calculating the variance of $\widetilde{Z}_i$ requires us to

reason about the covariance matrix of the sufficient statistics $\Sigma_T$, since the calculation involves terms of the form $(T_{j_1}(X_i) - \mu_{T,j_1})(T_{j_2}(X_i) - \mu_{T,j_2})$, $j_1, j_2 \in [k]$.

The previous remarks are summarized in the following lemma.

**Lemma B.1.** *Conditioning on $\eta$, it holds that:*

$$\mathop{\mathbb{E}}_{X_{\sim i}, X_i, M}\left[\widetilde{Z}_i\right] = 0,$$

$$\mathop{\mathrm{Var}}_{X_{\sim i}, X_i, M}\left(\widetilde{Z}_i\right) = \mathop{\mathbb{E}}_{X_{\sim i}, X_i, M}\left[\widetilde{Z}_i^2\right] = \mathop{\mathbb{E}}_{X_{\sim i}, M}\left[s^\top \Sigma_T s\right].$$

*Proof.* We start by computing the mean. We have:

$$\mathop{\mathbb{E}}_{X_{\sim i}, X_i, M}\left[\widetilde{Z}_i\right] = \mathop{\mathbb{E}}_{X_{\sim i}, X_i, M}\left[\sum_{j \in [k]} s_j (T_j(X_i) - \mu_{T,j})\right]$$

$$= \sum_{j \in [k]} \mathop{\mathbb{E}}_{X_{\sim i}, X_i, M}\left[s_j (T_j(X_i) - \mu_{T,j})\right]$$

$$\overset{(a)}{=} \sum_{j \in [k]} \mathop{\mathbb{E}}_{X_{\sim i}, M}[s_j] \mathop{\mathbb{E}}_{X_i}[T_j(X_i) - \mu_{T,j}]^{\nearrow 0}$$

$$= 0,$$

where $(a)$ relies on the independence of $s_j = \left[\frac{R_j^2}{4} - (\eta_j - m_j)^2\right][M_j(X_{\sim i}) - (\eta_j - m_j)]$ and $T_j(X_i) - \mu_{T,j}$ conditioned on $\eta$.

We now proceed with the variance:

$$\mathop{\mathrm{Var}}_{X_{\sim i}, X_i, M}\left(\widetilde{Z}_i\right) = \mathop{\mathbb{E}}_{X_{\sim i}, X_i, M}\left[\widetilde{Z}_i^2\right]$$

$$= \mathop{\mathbb{E}}_{X_{\sim i}, X_i, M}\left[\sum_{j_1 \in [k]} \sum_{j_2 \in [k]} s_{j_1} s_{j_2} (T_{j_1}(X_i) - \mu_{T,j_1})(T_{j_2}(X_i) - \mu_{T,j_2})\right]$$

$$= \sum_{j_1 \in [k]} \sum_{j_2 \in [k]} \mathop{\mathbb{E}}_{X_{\sim i}, M}[s_{j_1} s_{j_2}] \mathop{\mathbb{E}}_{X_i}[(T_{j_1}(X_i) - \mu_{T,j_1})(T_{j_2}(X_i) - \mu_{T,j_2})]$$

$$\overset{(a)}{=} \sum_{j_1 \in [k]} \sum_{j_2 \in [k]} \mathop{\mathbb{E}}_{X_{\sim i}, M}[s_{j_1} s_{j_2}] \Sigma_{T,j_1 j_2}$$

$$= \mathop{\mathbb{E}}_{X_{\sim i}, M}\left[\sum_{j_1 \in [k]} \sum_{j_2 \in [k]} s_{j_1} \Sigma_{T,j_1 j_2} s_{j_2}\right]$$

$$= \mathop{\mathbb{E}}_{X_{\sim i}, M}\left[s^\top \Sigma_T s\right].$$

where $(a)$ is by the second part of Proposition A.6. ∎

Now, we show how to upper-bound $\mathop{\mathbb{E}}_{X,M}[Z_i]$ using properties of differential privacy. We appeal to the definition of DP to relate $\mathop{\mathbb{E}}_{X,M}[Z_i]$ with $\mathop{\mathbb{E}}_{X_{\sim i}, X_i, M}\left[\widetilde{Z}_i\right]$ and $\mathop{\mathbb{E}}_{X_{\sim i}, X_i, M}\left[\widetilde{Z}_i^2\right]$. The proof involves splitting $Z_i$ in its positive and negative components and applying the definition of DP to them. The main steps of the proof are standard (e.g., see [SU17]), but we include everything for completeness.

*Proof of Lemma 2.3.* Consider the random variables $Z_{i,+} = \max\{Z_i, 0\}$ and $Z_{i,-} = -\min\{Z_i, 0\}$. These random variables are non-negative, so for any $T > 0$, we have:

$$
\begin{aligned}
\mathbb{E}_{X,M}[Z_i] &= \mathbb{E}_{X,M}[Z_{i,+}] - \mathbb{E}_{\eta,X,M}[Z_{i,-}] \\
&= \int_0^\infty \mathbb{P}_{X,M}[Z_{i,+} > t]\, dt - \int_0^\infty \mathbb{P}_{X,M}[Z_{i,-} > t]\, dt \\
&= \int_0^\infty \mathbb{P}_{X,M}[Z_i > t]\, dt - \int_0^\infty \mathbb{P}_{X,M}[Z_i < -t]\, dt \\
&= \int_0^T \mathbb{P}_{X,M}[Z_i > t]\, dt - \int_0^T \mathbb{P}_{X,M}[Z_i < -t]\, dt \\
&\quad + \int_T^\infty \mathbb{P}_{X,M}[Z_i > t]\, dt - \int_T^\infty \mathbb{P}_{X,M}[Z_i < -t]\, dt
\end{aligned}
\tag{3}
$$

We focus our attention on the first two terms of (3). We consider the sets $S_{+,X_i}, S_{-,X_i} \subseteq \mathbb{R}^k$:

$$
S_{+,X_i} := \left\{ Y \in \mathbb{R}^k : \sum_{j \in [k]} \left[ \frac{R_j^2}{4} - (\eta_j - m_j)^2 \right] [Y_j - (\eta_j - m_j)] \left( T_j(X_i) - \mu_{T,j} \right) \geq t \right\},
$$

$$
S_{-,X_i} := \left\{ Y \in \mathbb{R}^k : \sum_{j \in [k]} \left[ \frac{R_j^2}{4} - (\eta_j - m_j)^2 \right] [Y_j - (\eta_j - m_j)] \left( T_j(X_i) - \mu_{T,j} \right) \leq -t \right\}.
$$

By a direct application of the definition of DP on $S_{+,X_i}$ and $S_{-,X_i}$[2] for the neighboring datasets $X$ and $X_{\sim i}$ we have:

$$
\begin{aligned}
&\int_0^T \mathbb{P}_{X,M}[Z_i > t]\, dt - \int_0^T \mathbb{P}_{X,M}[Z_i < -t]\, dt \\
&= \int_0^T \mathbb{P}_{X,M}[M(X) \in S_{+,X_i}]\, dt - \int_0^T \mathbb{P}_{X,M}[M(X) \in S_{-,X_i}]\, dt \\
&\leq \int_0^T \left( e^\varepsilon \mathbb{P}_{X_{\sim i},X_i,M}[M(X_{\sim i}) \in S_{+,X_i}] + \delta \right) dt \\
&\quad - \int_0^T e^{-\varepsilon} \left( \mathbb{P}_{X_{\sim i},X_i,M}[M(X_{\sim i}) \in S_{-,X_i}] - \delta \right) dt \\
&\leq \int_0^T \left( e^\varepsilon \mathbb{P}_{X_{\sim i},X_i,M}\left[ \widetilde{Z}_i > t \right] + \delta \right) dt - \int_0^T e^{-\varepsilon} \left( \mathbb{P}_{X_{\sim i},X_i,M}\left[ \widetilde{Z}_i < -t \right] - \delta \right) dt \\
&= \left( 1 + e^{-\varepsilon} \right) \delta T + e^\varepsilon \int_0^T \mathbb{P}_{X_{\sim i},X_i,M}\left[ \widetilde{Z}_i > t \right] dt - e^{-\varepsilon} \int_0^T \mathbb{P}_{X_{\sim i},X_i,M}\left[ \widetilde{Z}_i < -t \right] dt.
\end{aligned}
$$

---

[2]These are random subsets of $\mathbb{R}^k$, since they depend on the random vector $X_i$ ($\eta$ is assumed to be fixed). This does not affect the way the definition of privacy is applied.

Under the assumption that $\varepsilon \in [0, 1]$, it holds that $e^{\varepsilon} \le 1 + 2\varepsilon$ and $1 \ge e^{-\varepsilon} \ge 1 - \varepsilon \ge 1 - 2\varepsilon$. Combined with the above, this yields:

$$\int_0^T \mathop{\mathbb{P}}_{X,M}[Z_i > t]\, dt - \int_0^T \mathop{\mathbb{P}}_{X,M}[Z_i < -t]\, dt$$

$$\le 2\delta T + (2\varepsilon + 1)\int_0^T \mathop{\mathbb{P}}_{X_{\sim i}, X_i, M}\left[\widetilde{Z}_i > t\right] dt + (2\varepsilon - 1)\int_0^T \mathop{\mathbb{P}}_{X_{\sim i}, X_i, M}\left[\widetilde{Z}_i < -t\right] dt$$

$$= 2\delta T + 2\varepsilon \int_0^T \left( \mathop{\mathbb{P}}_{X_{\sim i}, X_i, M}\left[\widetilde{Z}_i > t\right] + \mathop{\mathbb{P}}_{X_{\sim i}, X_i, M}\left[\widetilde{Z}_i < -t\right] \right) dt$$

$$+ \int_0^T \left( \mathop{\mathbb{P}}_{X_{\sim i}, X_i, M}\left[\widetilde{Z}_i > t\right] - \mathop{\mathbb{P}}_{X_{\sim i}, X_i, M}\left[\widetilde{Z}_i < -t\right] \right) dt$$

$$= 2\delta T + 2\varepsilon \int_0^T \mathop{\mathbb{P}}_{X_{\sim i}, X_i, M}\left[\left|\widetilde{Z}_i\right| > t\right] dt + \int_0^T \left( \mathop{\mathbb{P}}_{X_{\sim i}, X_i, M}\left[\widetilde{Z}_i > t\right] - \mathop{\mathbb{P}}_{X_{\sim i}, X_i, M}\left[\widetilde{Z}_i < -t\right] \right) dt$$

$$\le 2\delta T + 2\varepsilon \int_0^\infty \mathop{\mathbb{P}}_{X_{\sim i}, X_i, M}\left[\left|\widetilde{Z}_i\right| > t\right] dt + \int_0^T \left( \mathop{\mathbb{P}}_{X_{\sim i}, X_i, M}\left[\widetilde{Z}_i > t\right] - \mathop{\mathbb{P}}_{X_{\sim i}, X_i, M}\left[\widetilde{Z}_i < -t\right] \right) dt$$

$$= 2\delta T + 2\varepsilon \mathop{\mathbb{E}}_{X_{\sim i}, X_i, M}\left[\left|\widetilde{Z}_i\right|\right] + \int_0^T \left( \mathop{\mathbb{P}}_{X_{\sim i}, X_i, M}\left[\widetilde{Z}_i > t\right] - \mathop{\mathbb{P}}_{X_{\sim i}, X_i, M}\left[\widetilde{Z}_i < -t\right] \right) dt$$

$$= 2\delta T + 2\varepsilon \mathop{\mathbb{E}}_{X_{\sim i}, X_i, M}\left[\left|\widetilde{Z}_i\right|\right] + \int_0^\infty \left( \mathop{\mathbb{P}}_{X_{\sim i}, X_i, M}\left[\widetilde{Z}_i > t\right] - \mathop{\mathbb{P}}_{X_{\sim i}, X_i, M}\left[\widetilde{Z}_i < -t\right] \right) dt$$

$$- \int_T^\infty \left( \mathop{\mathbb{P}}_{X_{\sim i}, X_i, M}\left[\widetilde{Z}_i > t\right] - \mathop{\mathbb{P}}_{X_{\sim i}, X_i, M}\left[\widetilde{Z}_i < -t\right] \right) dt.$$

Defining $\widetilde{Z}_{i,+} = \max\left\{\widetilde{Z}_i, 0\right\}$ and $\widetilde{Z}_{i,-} = -\min\left\{\widetilde{Z}_i, 0\right\}$, we get:

$$\int_0^T \mathop{\mathbb{P}}_{X,M}[Z_i > t]\, dt - \int_0^T \mathop{\mathbb{P}}_{X,M}[Z_i < -t]\, dt$$

$$\le 2\delta T + 2\varepsilon \mathop{\mathbb{E}}_{X_{\sim i}, X_i, M}\left[\left|\widetilde{Z}_i\right|\right] + \left( \mathop{\mathbb{E}}_{X_{\sim i}, X_i, M}\left[\widetilde{Z}_{i,+}\right] - \mathop{\mathbb{E}}_{X_{\sim i}, X_i, M}\left[\widetilde{Z}_{i,-}\right] \right)$$

$$- \int_T^\infty \left( \mathop{\mathbb{P}}_{X_{\sim i}, X_i, M}\left[\widetilde{Z}_i > t\right] - \mathop{\mathbb{P}}_{X_{\sim i}, X_i, M}\left[\widetilde{Z}_i < -t\right] \right) dt$$

$$\le 2\delta T + 2\varepsilon \mathop{\mathbb{E}}_{X_{\sim i}, X_i, M}\left[\left|\widetilde{Z}_i\right|\right] + \mathop{\mathbb{E}}_{X_{\sim i}, X_i, M}\left[\widetilde{Z}_i\right]^{\,0}$$

$$- \int_T^\infty \left( \mathop{\mathbb{P}}_{X_{\sim i}, X_i, M}\left[\widetilde{Z}_i > t\right] - \mathop{\mathbb{P}}_{X_{\sim i}, X_i, M}\left[\widetilde{Z}_i < -t\right] \right) dt$$

$$= 2\delta T + 2\varepsilon \mathop{\mathbb{E}}_{X_{\sim i}, X_i, M}\left[\left|\widetilde{Z}_i\right|\right] - \int_T^\infty \left( \mathop{\mathbb{P}}_{X_{\sim i}, X_i, M}\left[\widetilde{Z}_i > t\right] - \mathop{\mathbb{P}}_{X_{\sim i}, X_i, M}\left[\widetilde{Z}_i < -t\right] \right) dt, \quad (4)$$

where we appealed to Lemma B.1 along the way.

The second version of Fact F.4 for $X = \left|\widetilde{Z}_i\right|, Y = 1$, and Lemma B.1 yield:

$$\mathop{\mathbb{E}}_{X \sim i, X_i, M} \left[\left|\widetilde{Z}_i\right|\right] \leq \sqrt{\mathop{\mathbb{E}}_{X \sim i, X_i, M} \left[\widetilde{Z}_i^2\right]} = \sqrt{\mathop{\mathbb{E}}_{X \sim i, M} \left[s^\top \Sigma_T s\right]}. \tag{5}$$

Substituting (4) and (5) into (3), we get:

$$\mathop{\mathbb{E}}_{X, M} \left[Z_i\right] \leq 2\delta T + 2\varepsilon \sqrt{\mathop{\mathbb{E}}_{X \sim i, M} \left[s^\top \Sigma_T s\right]} + \int_T^\infty \left(\mathop{\mathbb{P}}_{X, M} \left[Z_i > t\right] - \mathop{\mathbb{P}}_{X, M} \left[Z_i < -t\right]\right) dt$$

$$+ \int_T^\infty \left(\mathop{\mathbb{P}}_{X \sim i, X_i, M} \left[\widetilde{Z}_i < -t\right] - \mathop{\mathbb{P}}_{X \sim i, X_i, M} \left[\widetilde{Z}_i > t\right]\right) dt$$

$$\leq 2\delta T + 2\varepsilon \sqrt{\mathop{\mathbb{E}}_{X \sim i, M} \left[s^\top \Sigma_T s\right]} + \int_T^\infty \left(\mathop{\mathbb{P}}_{X, M} \left[|Z_i| > t\right] + \mathop{\mathbb{P}}_{X \sim i, X_i, M} \left[\left|\widetilde{Z}_i\right| > t\right]\right) dt. \tag{6}$$

We now show how to bound each of the tail probabilities involving $Z_i$ and $\widetilde{Z}_i$. For $Z_i$, we have:

$$|Z_i| = \left|\sum_{j \in [k]} \left[\frac{R_j^2}{4} - (\eta_j - m_j)^2\right] \left[M_j(X) - (\eta_j - m_j)\right] (T_j(X_i) - \mu_{T,j})\right|$$

$$\leq \sum_{j \in [k]} \left[\frac{R_j^2}{4} - (\eta_j - m_j)^2\right] |M_j(X) - (\eta_j - m_j)| |T_j(X_i) - \mu_{T,j}|$$

$$\overset{(a)}{\leq} \frac{\|R\|_\infty^3}{4} \sum_{j \in [k]} |T_j(X_i) - \mu_{T,j}|$$

$$\overset{(b)}{\leq} \frac{\|R\|_\infty^3}{4} \sqrt{k} \|T(X_i) - \mu_T\|_2,$$

where $(a)$ relies on the assumption $|M_j(X_i) - (\eta_j - m_j)| \leq R_j, \forall j \in [k]$, and $(b)$ uses the first version of Fact F.4 for $x = \vec{1}_k, y_j = |T_j(X_i) - \mu_{T,j}|, \forall j \in [k]$.

The above yields:

$$\int_T^\infty \mathop{\mathbb{P}}_{X, M} \left[|Z_i| > t\right] dt \leq \int_T^\infty \mathop{\mathbb{P}}_{X_i} \left[\|T(X_i) - \mu_T\|_2 > \frac{4t}{\|R\|_\infty^3 \sqrt{k}}\right] dt. \tag{7}$$

Upper-bounding the term involving $\widetilde{Z}_i$ in similar fashion and substituting (7) to (6), we get:

$$\mathop{\mathbb{E}}_{X, M} \left[Z_i\right] \leq 2\delta T + 2\varepsilon \sqrt{\mathop{\mathbb{E}}_{X \sim i, M} \left[s^\top \Sigma_T s\right]} + 2 \int_T^\infty \mathop{\mathbb{P}}_{X_i} \left[\|T(X_i) - \mu_T\|_2 > \frac{4t}{\|R\|_\infty^3 \sqrt{k}}\right] dt.$$

■

Now, we give the complete proof of Theorem 2.4.

*Proof of Theorem 2.4.* By Lemma 2.2 and (3), we have:

$$\mathop{\mathbb{E}}_{\eta, X, M} [Z] \geq \frac{\|R\|_2^2}{24}.$$

Additionally, by Lemma 2.3, we have:

$$\mathbb{E}_{\eta,X,M}[Z]$$

$$= \sum_{i\in[n]} \mathbb{E}_{\eta,X,M}[Z_i]$$

$$= \sum_{i\in[n]} \mathbb{E}_{\eta}\left[\mathbb{E}_{X,M}[Z_i]\right]$$

$$\leq \sum_{i\in[n]} \mathbb{E}_{\eta}\left[2\delta T + 2\varepsilon\sqrt{\mathbb{E}_{X_{\sim i},M}[s^\top \Sigma_T s]} + 2\int_T^\infty \mathbb{P}_{X_i}\left[\|T(X_i) - \mu_T\|_2 > \frac{4t}{\|R\|_\infty^3 \sqrt{k}}\right] dt\right]$$

$$\leq n\left\{2\delta T + 2\varepsilon\mathbb{E}_{\eta}\left[\sqrt{\mathbb{E}_{X_{\sim i},M}[s^\top \Sigma_T s]}\right] + 2\int_T^\infty \mathbb{P}_{X_i}\left[\|T(X_i) - \mu_T\|_2 > \frac{4t}{\|R\|_\infty^3 \sqrt{k}}\right] dt\right\}.$$

Combining the two, we get the desired result. ∎

We conclude this appendix with three remarks about how Theorem 2.4 is applied.

**Remark B.2.** In order to apply Theorem 2.4, it is necessary for the parameter range $\mathcal{H}$ to have a subset that is a (potentially degenerate) axis-aligned hyperrectangle. This is a very mild assumption, since $\mathcal{H}$ is known to be convex (see the last part of Proposition A.6). Moreover, if $\mathcal{H}$ has a subset that is a hyperrectangle, but is not axis-aligned, it suffices to remark that, for any rotation matrix $U \in \mathbb{R}^{k\times k}, \eta^\top T(x) = (U\eta)^\top (UT(x))$. This mapping is bijective and preserves the $\ell_2$-distance between vectors. Thus, we can re-parameterize our exponential family in a way that will make the aforementioned hyperrectangle axis-aligned and lower bounds proven for estimating the natural parameter vector of the new family will hold for the original one as well.

**Remark B.3.** The last term in the upper bound of Lemma 2.3 involves the tail of the random variable $\|T(X_i) - \mu_T\|_2$. The obvious way to bound that term is:

$$\int_T^\infty \mathbb{P}_{X_i}\left[\|T(X_i) - \mu_T\|_2^2 > \frac{16t^2}{\|R\|_\infty^6 k}\right] dt \leq \int_T^\infty \frac{\mathbb{E}_{X_i}\left[\|T(X_i) - \mu_T\|_2^2\right]}{\frac{16t^2}{\|R\|_\infty^6 k}} dt$$

$$= \frac{\|R\|_\infty^6 k\,\mathrm{tr}(\Sigma_T)}{16}\int_T^\infty \frac{dt}{t^2}$$

$$= \frac{\|R\|_\infty^6 k\,\mathrm{tr}(\Sigma_T)}{16T}.$$

Following the steps we sketched previously, this approach leads to a range for $\delta$ of the form $\delta \leq \widetilde{\mathcal{O}}\left(\frac{1}{n^2}\right)$. However, when reasoning about the cost of approximate DP for various statistical tasks, in order to claim that a full characterization has been achieved, it is generally the case that lower bounds must be shown for $\delta$ as large as $\widetilde{\mathcal{O}}\left(\frac{1}{n}\right)$. To achieve this, we will apply the following strategy. First, observe that we can approximate $\|T(X) - \mu_T\|_2$ using an $\varepsilon$-net (see Definition E.1). In particular, if

$\mathcal{N}_{\frac{1}{4}}$ is a $\frac{1}{4}$-net of $\mathbb{S}^{k-1}$ of minimum cardinality, using Facts E.3, E.4, and a union bound, we get that:

$$\int_T^\infty \mathop{\mathbb{P}}_{X_i} \left[ \|T(X_i) - \mu_T\|_2 > \frac{4t}{\|R\|_\infty^3 \sqrt{k}} \right] dt$$

$$\leq \int_T^\infty \mathop{\mathbb{P}}_{X_i} \left[ \frac{1}{1 - \frac{1}{4}} \cdot \sup_{v \in \mathcal{N}_{\frac{1}{4}}} |\langle v, T(X_i) - \mu_T \rangle| > \frac{4t}{\|R\|_\infty^3 \sqrt{k}} \right] dt$$

$$= \int_T^\infty \mathop{\mathbb{P}}_{X_i} \left[ \sup_{v \in \mathcal{N}_{\frac{1}{4}}} |\langle v, T(X_i) - \mu_T \rangle| > \frac{3t}{\|R\|_\infty^3 \sqrt{k}} \right] dt$$

$$\leq \left| \mathcal{N}_{\frac{1}{4}} \right| \int_T^\infty \mathop{\mathbb{P}}_{X_i} \left[ |\langle v, T(X_i) - \mu_T \rangle| > \frac{3t}{\|R\|_\infty^3 \sqrt{k}} \right] dt$$

$$\leq 9^k \int_T^\infty \mathop{\mathbb{P}}_{X_i} \left[ |\langle v, T(X_i) - \mu_T \rangle| > \frac{3t}{\|R\|_\infty^3 \sqrt{k}} \right] dt.$$

Now, to reason about the tail probabilities of $|\langle v, T(X_i) - \mu_T \rangle|$, it suffices to leverage the fact that the MGFs of univariate projections of the sufficient statistics of exponential families are well-defined (see the third part of Proposition A.6). This renders it possible to prove Chernoff-type bounds in the context that we are considering. In practice, we will not have to prove our own bounds, instead appealing to various known bounds from the literature.

**Remark B.4.** In the above discussion, we have largely assumed that we want to prove lower bounds under approximate DP, corresponding to $\delta > 0$. However, applying the method becomes significantly simpler for pure DP. Indeed, the reason we had to split $\mathbb{R}_+$ in the intervals $[0, T]$ and $(T, \infty)$ in the proof of Lemma 2.3 and work separately with each interval is because, if we applied the definition of DP to each of $\mathop{\mathbb{P}}_{X,M}[Z_i > t]$, $\mathop{\mathbb{P}}_{X,M}[Z_i < -t]$ while integrating over all of $\mathbb{R}_+$, we would have to deal with a term of the form $\int_0^\infty \delta \, dt = \infty$. This would render our upper bound on the correlation terms vacuous and it would be impossible to obtain any meaningful lower bounds. On the other hand, for $\delta = 0$, it suffices to integrate over all of $\mathbb{R}_+$ (corresponding to $T = \infty$). This would eventually lead to the bound $n\mathop{\mathbb{E}}_\eta \left[ \sqrt{\mathop{\mathbb{E}}_{X_{\sim i}, M}[s^\top \Sigma_T s]} \right] \geq \frac{\|R\|_2^2}{48\varepsilon}$.

## C  Omitted Proofs from Section 3

### C.1  Omitted Proofs from Section 3.1

The first step of the proof is to write the Gaussian $\mathcal{N}(0, \Sigma)$ as an exponential family. By Fact F.3, we have that the density can be written in the form:

$$\begin{aligned}
p_\eta(x) &= h(x) \exp\left( \eta^\top T(x) - Z(\eta) \right), \\
h(x) &= 1, \\
T(x) &= -\frac{1}{2} \left( xx^\top \right)^\flat, \\
\eta &= \left( \Sigma^{-1} \right)^\flat, \\
Z(\eta) &= \frac{d}{2} \ln(2\pi) - \frac{1}{2} \ln\left( \det\left( \eta^\# \right) \right).
\end{aligned}$$

Observe that the natural parameter vector $\eta$ is the flattening of the *precision matrix* $\Sigma^{-1}$. However, we cannot use this representation, because $\Sigma^{-1}$ is symmetric. Indeed, this implies that the components of $\eta$ cannot be generated independently, as is required by Theorem 2.4. For that reason, we consider an alternative parameterization.

**Lemma C.1.** *Let $\mathcal{N}(0, \Sigma)$ be a $d$-dimensional Gaussian with unknown covariance $\Sigma$. Then, it belongs to an exponential family with $h_0 \equiv h, \eta_0 = 2U^\flat, T_0 \equiv T, Z_0(\eta_0) = \frac{d}{2}\ln(2\pi) - \frac{1}{2}\ln\left(\det\left(\frac{\eta_0^\# + (\eta_0^\#)^\top}{2}\right)\right)$ where $U \in \mathbb{R}^{d \times d}$ is an upper triangular matrix with:*

$$
U_{ij} := \begin{cases} \frac{1}{2}\left(\Sigma^{-1}\right)_{ii}, & i = j \\ \left(\Sigma^{-1}\right)_{ij}, & j > i \\ 0, & i > j \end{cases}.
$$

*In particular, it holds that $Z(\eta) = Z_0(\eta_0)$ and $\eta^\top T(x) = \eta_0^\top T_0(x)$.*

*Proof.* Observe that $\eta^\# = \Sigma^{-1} = U + U^\top = \frac{\eta_0^\# + (\eta_0^\#)^\top}{2} \implies Z(\eta) = Z_0(\eta_0)$. Additionally, we have:

$$
\begin{aligned}
\eta^\top T(x) = \left\langle \left(\Sigma^{-1}\right)^\flat, -\frac{1}{2}\left(xx^\top\right)^\flat \right\rangle = -\frac{1}{2}\left\langle \Sigma^{-1}, xx^\top \right\rangle &= -\frac{1}{2}x^\top \Sigma^{-1} x \\
&= -\frac{1}{2}x^\top \left(U + U^\top\right) x \\
&= -\frac{1}{2}x^\top U x - \frac{1}{2}x^\top U^\top x \\
&\overset{(a)}{=} -x^\top U x \\
&= -\left\langle U, xx^\top \right\rangle \\
&= \left\langle 2U^\flat, -\frac{1}{2}\left(xx^\top\right)^\flat \right\rangle \\
&= \eta_0^\top T_0(x),
\end{aligned}
$$

where $(a)$ used the fact that $x^\top U^\top x \in \mathbb{R} \implies x^\top U^\top x = \left(x^\top U^\top x\right)^\top = x^\top U x$.

Consequently, we have $p_\eta(x) = h(x)\exp\left(\eta^\top T(x) - Z(\eta)\right) = h_0(x)\exp\left(\eta_0^\top T_0(x) - Z_0(\eta_0)\right)$, verifying the equivalence of the two parameterizations. ∎

We now consider the following process to generate $\eta_0$. Roughly speaking, modulo the parameterization issues mentioned above, the process consists of sampling the elements of the precision matrix uniformly at random from intervals of width $\frac{1}{2d}$. For the diagonal elements, these intervals are centered at $\frac{3}{4}$, whereas for the non-diagonal elements they are centered at the origin. Similar constructions have been employed previously, e.g., in [KLSU19].

We proceed to identify some properties of the matrices generated by the above process that will help us to prove Theorem 3.1.

**Lemma C.2.** *Let $(\Sigma, \eta_0) \in \mathbb{R}^{d \times d} \times \mathbb{R}^{d^2}$ be the output of Algorithm 1. We have that:*

1. $\mathbb{I} \preceq \Sigma \preceq 2\mathbb{I}$.

2. *The components of $\eta_0$ are supported on intervals of the form:*

$$
I_\ell = \begin{cases} \left[\frac{3}{4} \pm \frac{1}{4d}\right], & (\ell-1) \bmod d = (\ell-1) \operatorname{div} d \\ \left[\pm \frac{1}{2d}\right], & (\ell-1) \bmod d > (\ell-1) \operatorname{div} d \\ \{0\}, & (\ell-1) \bmod d < (\ell-1) \operatorname{div} d \end{cases}.
$$

3. *The components of $\eta_0$ are independent and it holds that $\|R\|_\infty = \frac{1}{d}$ and $\|R\|_2^2 = \frac{1}{2}\left(1 - \frac{1}{2d}\right) \geq \frac{1}{4}$.*

4. *The vector $m \in \mathbb{R}^{d^2}$ is equal to:*

$$
m_\ell = \begin{cases} \frac{3}{4}, & \ell = (k-1)d + k, \text{ for some } k \in [d] \\ 0, & \text{otherwise} \end{cases}.
$$

---

**Algorithm 1** Covariance Matrix Sampling

**Input:** $d \in \mathbb{N}$.
**Output:** A pair $(\Sigma, \eta_0)$.
1: **procedure** CovSamp($d$)
2:     **for** $i \in [d]$ **do**
3:         **for** $j \in [d] \setminus [i-1]$ **do**
4:             **if** $i = j$ **then**
5:                 Draw an independent sample $\left(\Sigma^{-1}\right)_{ii} \sim \mathcal{U}\left[\frac{3}{4} \pm \frac{1}{4d}\right]$.
6:                 Let $U_{ii} = \frac{1}{2}\left(\Sigma^{-1}\right)_{ii}$.
7:             **else**
8:                 Draw an independent sample $\left(\Sigma^{-1}\right)_{ij} \sim \mathcal{U}\left[\pm \frac{1}{4d}\right]$.
9:                 Let $\left(\Sigma^{-1}\right)_{ji} = \left(\Sigma^{-1}\right)_{ij}$.
10:              Let $U_{ij} = \left(\Sigma^{-1}\right)_{ij}$ and $U_{ji} = 0$.
11:             **end if**
12:         **end for**
13:     **end for**
14:     Let $\eta_0 = 2U^\flat$.
15:     **return** $(\Sigma, \eta_0)$.
16: **end procedure**

---

*Proof.*      1. By a direct application of Theorem E.10, we get that the eigenvalues of $\Sigma^{-1}$ all lie in the interval $\left[\frac{1}{2}, 1\right]$. Thus, we have $\frac{1}{2}\mathbb{I} \preceq \Sigma^{-1} \preceq \mathbb{I} \iff \mathbb{I} \preceq \Sigma \preceq 2\mathbb{I}$.

2. Observe that, the first of the three branches in the definition corresponds to the diagonal elements of $\eta_0^{\#}$, the second corresponds to those above the diagonal, and the third corresponds to those below the diagonal. The result follows directly from this remark.

3. The non-constant components of $\eta_0$ are all independent draws from a uniform distribution, which immediately yields the independence property. In addition to that, we have $d$ non-constant components taking values in an interval of length $\frac{1}{2d}$, and $\frac{d(d-1)}{2}$ non-constant components take values in an interval of length $\frac{1}{d}$. This implies that $\|R\|_\infty = \frac{1}{d}$ and $\|R\|_2^2 = d \cdot \frac{1}{4d^2} + \frac{d(d-1)}{2} \cdot \frac{1}{d^2} = \frac{1}{2}\left(1 - \frac{1}{2d}\right)$.

4. Observe that the non-diagonal elements of $\eta_0^{\#} = 2U$ are either constant and equal to $0$ or take values in a $0$-centered interval. This implies that the only non-zero elements of $m$ are those corresponding to the diagonal elements of $\left(\Sigma^{-1}\right)_{ii}$. This yields the desired result.

∎

**Remark C.3.** Note that Theorem 3.1 holds only for $\alpha = \mathcal{O}(1)$. In contrast, prior fingerprinting lower bounds for mean estimation of product distributions and Gaussians gave non-trivial results in the low-accuracy regime $\alpha = \mathcal{O}\left(\sqrt{d}\right)$ (see, e.g., the results in Section G). In principle, it may be able to achieve similar lower bounds for covariance estimation in Frobenius norm, as the Frobenius diameter of the set of matrices that satisfy $\mathbb{I} \preceq \Sigma \preceq 2\mathbb{I}$ is equal to $\sqrt{d}$. However, by the third part of the above lemma, our construction only has an upper bound on this diameter of $\mathcal{O}(1)$. This makes estimation trivial for any $\alpha$ which is larger, as one could output an arbitrary parameter vector within the set. Thus, to prove a lower bound for all $\alpha = \mathcal{O}\left(\sqrt{d}\right)$, one must consider a different construction, and it is not merely a deficiency of our analysis.

We show that, if one has an estimator for the covariance matrix $\Sigma$, then this implies an estimator for the natural parameter vector $\eta_0$. Therefore, a lower bound for the latter problem implies a lower bound for the former, allowing us to focus on lower bounds for estimating the natural parameter vector. More precisely, we present an $(\varepsilon, \delta)$-DP mechanism $T_M \colon \mathbb{R}^{n \times d} \to \bigotimes_{j \in [d^2]} I_j$ that satisfies

the guarantee $\underset{X,T_M}{\mathbb{E}} \left[ \|T_M(X) - (\eta_0 - m)\|_2^2 \right] \leq \alpha^2$. $T_M$ assumes the existence of an $(\varepsilon, \delta)$-DP mechanism $M : \mathbb{R}^{n \times d} \to \mathbb{R}^{d \times d}$ such that $\underset{X,M}{\mathbb{E}} \left[ \|M(X) - \Sigma\|_\Sigma^2 \right] \leq \frac{\alpha^2}{8}$.

---

**Algorithm 2** From Covariance Estimation to Natural Parameter Estimation

---

    **Input:** $X = (X_1, \ldots, X_n) \sim \mathcal{N}(0, \Sigma)^{\otimes n}$ and a mechanism $M : \mathbb{R}^{n \times d} \to \mathbb{R}^{d \times d}$.
    **Output:** $T_M(X) \in \underset{j \in [d^2]}{\bigotimes} I_j$.

1: **procedure** $T_M(X)$
2:     Let $\widehat{\Sigma} = M(X)$.
3:     **for** $i \in [d]$ **do**        $\triangleright$ $\widehat{\Sigma}^{-1}$ may not be equal to any possible $\Sigma^{-1}$ generated by Algorithm 1.
4:         **for** $j \in [d]$ **do**
5:             Let $\widetilde{\Sigma}_{ij}^{-1}$ be the projection of $\left( \widehat{\Sigma}^{-1} \right)_{ij}$ onto the support of $\Sigma^{-1}$.
6:         **end for**
7:     **end for**
8:     Let $\widetilde{U} \in \mathbb{R}^{d \times d}$ be an upper triangular matrix such that $\widetilde{\Sigma}^{-1} = \widetilde{U} + \widetilde{U}^\top$.
9:     Let $T_M(X) = 2\widetilde{U}^\flat - m$.
10:     **return** $T_M(X)$.
11: **end procedure**

---

**Lemma C.4.** *Let $\Sigma \in \mathbb{R}^{d \times d}$ be a covariance generated by Algorithm 1, and $X \sim \mathcal{N}(0, \Sigma)^{\otimes n}$ be a dataset. If $M : \mathbb{R}^{n \times d} \to \mathbb{R}^{d \times d}$ is an $(\varepsilon, \delta)$-DP mechanism with $\underset{X,M}{\mathbb{E}} \left[ \|M(X) - \Sigma\|_\Sigma^2 \right] \leq \frac{\alpha^2}{8} \leq \frac{1}{32}$, then Algorithm 2 constructs an $(\varepsilon, \delta)$-DP mechanism $T_M : \mathbb{R}^{n \times d} \to \underset{j \in [d^2]}{\bigotimes} I_j$ that satisfies $\underset{X,T_M}{\mathbb{E}} \left[ \|T_M(X) - (\eta_0 - m)\|_2^2 \right] \leq \alpha^2 \leq \frac{1}{4}$.*

*Proof.* The privacy guarantee of $T_M$ is an immediate consequence of the guarantee of $M$ and Lemma A.3. Additionally, the only randomness used by $T_M$ is that used by $M$. Having established that, we focus our attention on the accuracy guarantee. By the definitions of $\eta_0$ and $T_M$, we have:

$$\underset{X,T_M}{\mathbb{E}} \left[ \|T_M(X) - (\eta_0 - m)\|_2^2 \right] = 4 \underset{X,M}{\mathbb{E}} \left[ \left\| \widetilde{U} - U \right\|_F^2 \right]. \tag{8}$$

Additionally, the definitions of $U$ and $\widetilde{U}$ yield:

$$\left\| \widetilde{U} - U \right\|_F^2 = \sum_{j \geq i} \left( \widetilde{U}_{ij} - U_{ij} \right)^2 = \frac{1}{4} \sum_{i \in [d]} \left[ \left( \widetilde{\Sigma}^{-1} \right)_{ii} - \left( \Sigma^{-1} \right)_{ii} \right]^2 + \frac{1}{2} \sum_{i \neq j} \left[ \left( \widetilde{\Sigma}^{-1} \right)_{ij} - \left( \Sigma^{-1} \right)_{ij} \right]^2$$

$$\leq \frac{1}{2} \left\| \widetilde{\Sigma}^{-1} - \Sigma^{-1} \right\|_F^2$$

$$\leq \frac{1}{2} \left\| \widehat{\Sigma}^{-1} - \Sigma^{-1} \right\|_F^2, \tag{9}$$

where the last inequality uses the fact that the projection we perform in Line 5 ensures that $\left| \widetilde{\Sigma}_{ij}^{-1} - \left( \Sigma^{-1} \right)_{ij} \right| \leq \left| \left( \widehat{\Sigma}^{-1} \right)_{ij} - \left( \Sigma^{-1} \right)_{ij} \right|, \forall i, j \in [d]$.

Combining (8) and (9), we get:

$$\underset{X,T_M}{\mathbb{E}} \left[ \|T_M(X) - (\eta_0 - m)\|_2^2 \right] \leq 2 \underset{X,M}{\mathbb{E}} \left[ \left\| \widehat{\Sigma}^{-1} - \Sigma^{-1} \right\|_F^2 \right], \tag{10}$$

Now, observe that:

$$
\begin{aligned}
\left\| \widehat{\Sigma}^{-1} - \Sigma^{-1} \right\|_F = \left\| \Sigma^{-\frac{1}{2}} \left( \mathbb{I} - \Sigma^{\frac{1}{2}} \widehat{\Sigma}^{-1} \Sigma^{\frac{1}{2}} \right) \Sigma^{-\frac{1}{2}} \right\|_F &\overset{(a)}{\leq} \left\| \Sigma^{-\frac{1}{2}} \right\|_2^2 \left\| \mathbb{I} - \Sigma^{\frac{1}{2}} \widehat{\Sigma}^{-1} \Sigma^{\frac{1}{2}} \right\|_F \\
&\overset{(b)}{\leq} \left\| \Sigma^{-1} - \widehat{\Sigma}^{-1} \right\|_{\Sigma^{-1}} \\
&= \left\| \Sigma^{-1} - \widehat{\Sigma}^{-1} \right\|_{\Sigma^{-\frac{1}{2}} \widehat{\Sigma}^{-\frac{1}{2}} \widehat{\Sigma} \widehat{\Sigma}^{-\frac{1}{2}} \Sigma^{-\frac{1}{2}}} \\
&\overset{(c)}{=} \left\| \widehat{\Sigma}^{\frac{1}{2}} \Sigma^{\frac{1}{2}} \left( \Sigma^{-1} - \widehat{\Sigma}^{-1} \right) \Sigma^{\frac{1}{2}} \widehat{\Sigma}^{\frac{1}{2}} \right\|_{\widehat{\Sigma}} \\
&= \left\| \widehat{\Sigma} - \widehat{\Sigma}^{\frac{1}{2}} \Sigma^{\frac{1}{2}} \widehat{\Sigma}^{-1} \Sigma^{\frac{1}{2}} \widehat{\Sigma}^{\frac{1}{2}} \right\|_{\widehat{\Sigma}} \\
&\overset{(d)}{=} \left\| \widehat{\Sigma} - \Sigma \right\|_{\widehat{\Sigma}} \\
&\overset{(e)}{\leq} 2 \left\| \widehat{\Sigma} - \Sigma \right\|_{\Sigma}, \\
&= 2 \left\| T_M (X) - \Sigma \right\|_{\Sigma}, \quad (11)
\end{aligned}
$$

where $(a)$ uses Fact E.6, $(b)$ uses the fact that $\Sigma^{-1} \preceq \mathbb{I}$ (which holds by the first part of Lemma C.4), $(c)$ uses Lemma E.7 for $A = \Sigma^{-\frac{1}{2}} \widehat{\Sigma}^{-\frac{1}{2}}$, $(d)$ is by Lemma E.8, and $(e)$ is by Lemma E.9.

Substituting based on (11) into (10), we get:

$$
\underset{X, T_M}{\mathbb{E}} \left[ \| T_M (X) - (\eta_0 - m) \|_2^2 \right] \leq 8 \underset{X, M}{\mathbb{E}} \left[ \| M (X) - \Sigma \|_{\Sigma}^2 \right],
$$

which yields the desired result. ∎

An immediate consequence of the above is the following:

**Corollary C.5.** *Let $\Sigma \in \mathbb{R}^{d \times d}$ be a covariance generated by Algorithm 1, and let $X \sim \mathcal{N}(0, \Sigma)^{\otimes n}$. If any $(\varepsilon, \delta)$-DP mechanism $T \colon \mathbb{R}^{n \times d} \to \bigotimes_{j \in [d^2]} I_j$ with $\underset{X, T}{\mathbb{E}} \left[ \| T (X) - (\eta_0 - m) \|_2^2 \right] \leq 8\alpha^2 \leq \frac{1}{4}$ requires at least $n \geq n_{\eta_0}$ samples, the same sample complexity lower bound holds for any $(\varepsilon, \delta)$-DP mechanism $M \colon \mathbb{R}^{n \times d} \to \mathbb{R}^{d \times d}$ that satisfies $\underset{X, M}{\mathbb{E}} \left[ \| M (X) - \Sigma \|_{\Sigma}^2 \right] \leq \alpha^2$.*

Having established the reduction from estimating $\eta_0$ with respect to the $\ell_2$-norm to estimating $\Sigma$ with respect to the Mahalanobis norm, it remains now to apply Theorem 2.4 to the former problem. For that reason, we work towards bounding the various quantities involved in it.

First, in accordance with Theorem 2.4, we assume the existence of an $(\varepsilon, \delta)$-DP mechanism $M \colon \mathbb{R}^{n \times d} \to \bigotimes_{j \in [d^2]} I_j$ such that:

$$
\underset{X, M}{\mathbb{E}} \left[ \| M (X) - (\eta_0 - m) \|_2^2 \right] \leq 8\alpha^2 \leq \frac{1}{96} \leq \frac{\| R \|_2^2}{24}, \quad (12)
$$

where we appealed to the third part of Lemma C.2.

The next step to establishing our lower bound is reasoning about the quantity $\underset{X_{\sim i}, M}{\mathbb{E}} \left[ s^\top \Sigma_{T_0} s \right]$.

To do that, we identify an expression for $\Sigma_{T_0}$ for $\mathcal{N}(0, \Sigma)$ and use a result from [DKK+16]. By Lemma C.1, we have $T_0 \equiv T \implies \Sigma_{T_0} = \Sigma_T$. This yields:

$$
\begin{aligned}
\Sigma_T &= \underset{Y \sim \mathcal{N}(0, \Sigma)}{\mathbb{E}} \left[ T (Y) T (Y)^\top \right] - \mu_T \mu_T^\top \\
&= \frac{1}{4} \underset{Y \sim \mathcal{N}(0, \Sigma)}{\mathbb{E}} \left[ (YY^\top)^\flat \left( (YY^\top)^\flat \right)^\top \right] - \frac{1}{4} \Sigma^\flat \left( \Sigma^\flat \right)^\top \\
&= \frac{1}{4} \underset{Y \sim \mathcal{N}(0, \Sigma)}{\mathbb{E}} \left[ (Y \otimes Y)(Y \otimes Y)^\top \right] - \frac{1}{4} \Sigma^\flat \left( \Sigma^\flat \right)^\top \quad (13)
\end{aligned}
$$

Observe now that, based on the definition we gave for $s$ in Section 2 and the characteristics of the particular exponential family that we are considering here, the matrix $s^{\#} \in \mathbb{R}^{d \times d}$ is symmetric. We use this observation to connect our problem with the following result:

**Proposition C.6.** *[Theorem 4.2 from [DKK$^+$16]] Let $\mathcal{S}_{\mathrm{sym}} = \left\{ M^{\flat} \in \mathbb{R}^{d^2} : M = M^{\top} \right\}$ and $X \sim \mathcal{N}(0, \Sigma)$. Let $M$ be the $d^2 \times d^2$ matrix given by $M = \mathbb{E}\left[ (X \otimes X)(X \otimes X)^{\top} \right]$. Then, as an operator on $\mathcal{S}_{\mathrm{sym}}$, we have:*

$$M = 2\Sigma^{\otimes 2} + \left(\Sigma^{\flat}\right)\left(\Sigma^{\flat}\right)^{\top}.$$

Using the above, we prove the following lemma:

**Lemma C.7.** *Let $\Sigma \in \mathbb{R}^{d \times d}$ be a covariance matrix generated by Algorithm 1, and let $X \sim \mathcal{N}(0, \Sigma)^{\otimes n}$ be a dataset. Also, let $M \colon \mathbb{R}^{n \times d} \to \bigotimes_{j \in [d^2]} I_j$ be an $(\varepsilon, \delta)$-DP mechanism such that $\underset{X,M}{\mathbb{E}}\left[ \|M(X) - (\eta_0 - m)\|_2^2 \right] \leq 8\alpha^2$. Then, it holds that:*

$$\underset{X_{\sim i}, M}{\mathbb{E}}\left[ s^{\top} \Sigma_T s \right] \leq \frac{\alpha^2}{d^4}.$$

*Proof.* By Proposition C.6 and (13), we get that:

$$
\begin{aligned}
\underset{X_{\sim i}, M}{\mathbb{E}}\left[ s^{\top} \Sigma_T s \right] &= \frac{1}{4} \underset{X_{\sim i}, M}{\mathbb{E}}\left[ s^{\top} \underset{Y \sim \mathcal{N}(0,\Sigma)}{\mathbb{E}}\left[ (Y \otimes Y)(Y \otimes Y)^{\top} \right] s \right] - \frac{1}{4} \underset{X_{\sim i}, M}{\mathbb{E}}\left[ s^{\top} \Sigma^{\flat} \left(\Sigma^{\flat}\right)^{\top} s \right] \\
&= \frac{1}{4} \underset{X_{\sim i}, M}{\mathbb{E}}\left[ s^{\top} \left( 2\Sigma^{\otimes 2} + \left(\Sigma^{\flat}\right)\left(\Sigma^{\flat}\right)^{\top} \right) s \right] - \frac{1}{4} \underset{X_{\sim i}, M}{\mathbb{E}}\left[ s^{\top} \Sigma^{\flat} \left(\Sigma^{\flat}\right)^{\top} s \right] \\
&= \frac{1}{2} \underset{X_{\sim i}, M}{\mathbb{E}}\left[ s^{\top} \Sigma^{\otimes 2} s \right] \\
&= \frac{1}{2} \underset{X_{\sim i}, M}{\mathbb{E}}\left[ \|s\|_2^2 \frac{s^{\top}}{\|s\|_2} \Sigma^{\otimes 2} \frac{s}{\|s\|_2} \right] \\
&\overset{(a)}{\leq} \frac{1}{2} \left\|\Sigma^{\otimes 2}\right\|_2 \underset{X_{\sim i}, M}{\mathbb{E}}\left[ \|s\|_2^2 \right] \\
&\overset{(b)}{=} \frac{1}{2} \left\|\Sigma\right\|_2^2 \underset{X_{\sim i}, M}{\mathbb{E}}\left[ \|s\|_2^2 \right] \\
&\overset{(c)}{\leq} 2 \underset{X_{\sim i}, M}{\mathbb{E}}\left[ \sum_{j \in [d^2]} \left[ \frac{R_j^2}{4} - (\eta_{0,j} - m_j)^2 \right]^2 [M_j(X_{\sim i}) - (\eta_{0,j} - m_j)]^2 \right] \\
&\leq \frac{\|R\|_{\infty}^4}{16} 2 \underset{X_{\sim i}, M}{\mathbb{E}}\left[ \|M(X_{\sim i}) - (\eta_0 - m)\|_2^2 \right] \\
&\overset{(d)}{\leq} \frac{\|R\|_{\infty}^4}{8} \cdot 8\alpha^2 \\
&= \|R\|_{\infty}^4 \, \alpha^2 \\
&= \frac{\alpha^2}{d^4},
\end{aligned}
$$

where $(a)$ is by the definition of the spectral norm, $(b)$ is by Fact E.5, $(c)$ is by the first part of Lemma C.2, $(d)$ is because $\underset{X,M}{\mathbb{E}}\left[ \|M(X) - (\eta_0 - m)\|_2^2 \right] \leq 8\alpha^2$, and the last equality is by the third part of Lemma C.2. ∎

Now, it remains to reason about the terms involving the tail probabilities of $\|T(X_i) - \mu_T\|_2$. Our approach closely follows the strategy sketched in Remark B.3. The following lemma is devoted to implementing this.

**Lemma C.8.** *Let $\Sigma \in \mathbb{R}^{d \times d}$ be a covariance matrix generated by Algorithm 1, and $X_i \sim \mathcal{N}(0, \Sigma)$. For $T \geq \frac{1}{3d^2}$, we have:*

$$\int\limits_T^\infty \mathop{\mathbb{P}}_{X_i} \left[ \|T(X_i) - \mu_T\|_2 > 4d^2 t \right] \, dt \leq \frac{2}{3cd^2} e^{-d^2(3cT - 2\ln(3))},$$

*where $c \approx 0.036425$.*

*Proof.* Setting $Y := \Sigma^{-\frac{1}{2}} X_i \sim \mathcal{N}(0, \mathbb{I})$, we observe that:

$$\|T(X_i) - \mu_T\|_2 = \frac{1}{2} \left\| \left( X_i X_i^\top \right)^\flat - \Sigma^\flat \right\|_2 = \frac{1}{2} \left\| X_i X_i^\top - \Sigma \right\|_F$$

$$= \frac{1}{2} \left\| \Sigma^{\frac{1}{2}} \left[ \left( \Sigma^{-\frac{1}{2}} X_i \right) \left( \Sigma^{-\frac{1}{2}} X_i \right)^\top - \mathbb{I} \right] \Sigma^{\frac{1}{2}} \right\|_F$$

$$\overset{(a)}{\leq} \frac{\left\| \Sigma^{\frac{1}{2}} \right\|_2^2}{2} \left\| YY^\top - \mathbb{I} \right\|_F$$

$$\overset{(b)}{\leq} \left\| YY^\top - \mathbb{E} \left[ YY^\top \right] \right\|_F,$$

where $(a)$ uses Fact E.6, and $(b)$ uses the assumption that $\Sigma \preceq 2\mathbb{I}$ and the remark that, for symmetric PSD matrices, $\left\| \Sigma^{\frac{1}{2}} \right\|_2 = \sqrt{\|\Sigma\|_2}$.

Thus, we have:

$$\int\limits_T^\infty \mathop{\mathbb{P}}_{X_i} \left[ \|T(X_i) - \mu_T\|_2 > 4d^2 t \right] \, dt \leq \int\limits_T^\infty \mathbb{P} \left[ \left\| YY^\top - \mathbb{E} \left[ YY^\top \right] \right\|_F > 4d^2 t \right] \, dt.$$

Working as we described in Remark B.3, we get:

$$\int\limits_T^\infty \mathop{\mathbb{P}}_{X_i} \left[ \|T(X_i) - \mu_T\|_2 > 4d^2 t \right] \, dt$$

$$\leq \int\limits_T^\infty \mathbb{P} \left[ \left\| \left( YY^\top - \mathbb{E} \left[ YY^\top \right] \right)^\flat \right\|_2 > 4d^2 t \right] \, dt$$

$$\leq 9^{d^2} \int\limits_T^\infty \mathbb{P} \left[ \left| \left\langle M^\flat, \left( YY^\top - \mathbb{E} \left[ YY^\top \right] \right)^\flat \right\rangle \right| > 3d^2 t \right] \, dt$$

$$\leq 9^{d^2} \int\limits_T^\infty \mathbb{P} \left[ \left| \left\langle M, YY^\top - \mathbb{E} \left[ YY^\top \right] \right\rangle \right| > 3d^2 t \right] \, dt$$

$$= 9^{d^2} \int\limits_T^\infty \mathbb{P} \left[ \left| Y^\top M Y - \mathbb{E} \left[ Y^\top M Y \right] \right| > 3d^2 t \right] \, dt$$

$$= 9^{d^2} \int\limits_T^\infty \mathbb{P} \left[ \left| Y^\top \frac{M + M^\top}{2} Y - \mathbb{E} \left[ Y^\top \frac{M + M^\top}{2} Y \right] \right| > 3d^2 t \right] \, dt \tag{14}$$

where $M$ is a matrix with $\|M\|_F = 1$.

We have that $Y \sim \mathcal{N}(0, \mathbb{I})$, and that $\frac{M+M^\top}{2}$ is symmetric, so we can apply Fact F.7 to (14). Since $\left\| \frac{M+M^\top}{2} \right\|_F \leq \frac{\|M\|_F + \|M^\top\|_F}{2} \leq 1$ and $\left\| \frac{M+M^\top}{2} \right\|_2 \leq \left\| \frac{M+M^\top}{2} \right\|_F \leq 1$, we get:

$$\int\limits_T^\infty \mathop{\mathbb{P}}_{X_i} \left[ \|T(X_i) - \mu_T\|_2 > 4d^2 t \right] \, dt \leq 2 \cdot 9^{d^2} \int\limits_T^\infty \exp \left( -c \min \left\{ 9d^4 t^2, 3d^2 t \right\} \right) \, dt.$$

By our assumption that on $T$, we get $\min\left\{9d^4t^2, 3d^2t\right\} = 3d^2t, \forall t \geq T$. This leads to:

$$\int\limits_T^\infty \mathop{\mathbb{P}}\limits_{X_i}\left[\left\|T\left(X_i\right) - \mu_T\right\|_2 > 4d^2t\right]\,dt \leq 2 \cdot 9^{d^2}\int\limits_T^\infty \exp\left(-3cd^2t\right)\,dt = \frac{2 \cdot 9^{d^2}}{3cd^2}e^{-3cd^2T}$$

$$= \frac{2}{3cd^2}e^{-d^2(3cT - 2\ln(3))}.$$

∎

We are now ready to prove the main result of this section.

*Proof of Theorem 3.1.* We assume that the process generating $\Sigma$ is that of Algorithm 1. By the first part of Lemma C.2, the condition $\mathbb{I} \preceq \Sigma \preceq 2\mathbb{I}$ is satisfied. We assume that there exists an $(\varepsilon, \delta)$-DP mechanism $M\colon \mathbb{R}^{n \times d} \to \bigotimes\limits_{j \in [d^2]} I_j$ that satisfies $\mathop{\mathbb{E}}\limits_{X,M}\left[\left\|M\left(X\right) - \left(\eta_0 - m\right)\right\|_2^2\right] \leq 8\alpha^2 \leq \frac{1}{96}$.
Combining the results of Lemmas C.7, and C.8, with Theorem 2.4, we get:

$$n\left(2\delta T + 2\frac{\alpha\varepsilon}{d^2} + \frac{4}{3cd^2}e^{-d^2(3cT - 2\ln(3))}\right) \geq \frac{1}{48}\left(1 - \frac{1}{2d}\right) \geq \frac{1}{96}. \tag{15}$$

It remains to set the values of $T$ and $\delta$ appropriately so that:

$$\delta T \geq \frac{4}{3cd^2}e^{-d^2(3cT - 2\ln(3))} \text{ and } 3n\delta T \leq \frac{1}{192}.$$

The first of these two conditions yields:

$$\delta T \geq \frac{4}{3cd^2}e^{-d^2(3cT - 2\ln(3))} \iff Te^{d^2(3cT - 2\ln(3))} \geq \frac{4}{3cd^2} \cdot \frac{1}{\delta}.$$

We set $T = \frac{1}{3c}\left(2\ln\left(3\right) + \frac{1}{d^2}\ln\left(\frac{1}{\delta}\right)\right)$. This satisfies the constraint of Lemma C.8. Then, the above becomes:

$$\frac{1}{3c}\left(2\ln\left(3\right) + \frac{1}{d^2}\ln\left(\frac{1}{\delta}\right)\right)\frac{1}{\delta} \geq \frac{1}{cd^2} \cdot \frac{1}{\delta} \iff \ln\left(\frac{1}{\delta}\right) \geq 3 - 2\ln\left(3\right)d^2.$$

The RHS of the last inequality is $< 0$ for $d \geq 2$. For $d = 1$, we get the constraint $\delta \leq e^{2\ln(3) - 3} \approx 0.44$. Respecting this constraint, we proceed to identify a range of values for $\delta$ such that $3n\delta T \leq \frac{1}{192}$. This is equivalent to:

$$\frac{\delta}{c}\left(2\ln\left(3\right) + \frac{1}{d^2}\ln\left(\frac{1}{\delta}\right)\right) \leq \frac{1}{192n} \iff \delta\left(1 + \frac{\ln\left(\frac{1}{\delta}\right)}{2\ln\left(3\right)d^2}\right) \leq \frac{c}{384\ln\left(3\right)n}. \tag{16}$$

The above inequality yields a constraint on $\delta$. To ensure that $\ln\left(\frac{1}{\delta}\right) \geq 0$, we need $\frac{c}{384\ln(3)n\delta} - 1 \geq 0 \iff \delta \leq \frac{c}{384\ln(3)n}$. We know from Lemma C.8 value of $c$ is $c \approx 0.036425$. Thus, this last constraint is stricter than $\delta \leq 0.44$.

For (16) to hold, we will show that it suffices to have:

$$\delta \leq \min\left\{\frac{c}{768\ln\left(3\right)n}, \frac{cd^2}{768n\ln\left(\frac{384n}{cd^2}\right)}\right\},$$

which trivially satisfies the previous constraint.

We set $\phi := \frac{cd^2}{384n}$, so the proposed range for $\delta$ becomes $\delta \leq \min\left\{\frac{\phi}{2\ln(3)d^2}, \frac{\phi}{2\ln\left(\frac{1}{\phi}\right)}\right\}$. Additionally, (16) can be equivalently written as:

$$\delta\left(1 + \frac{\ln\left(\frac{1}{\delta}\right)}{2\ln\left(3\right)d^2}\right) \leq \frac{\phi}{\ln\left(3\right)d^2}. \tag{17}$$

Observe that, depending on how $\delta$ compares with $e^{-2\ln(3)d^2}$ determines which of the terms 1 and $\frac{\ln\left(\frac{1}{\delta}\right)}{2\ln(3)d^2}$ dominates. Thus, our strategy to verify our claim is by considering cases based on whether the proposed range for $\delta$ includes $e^{-2\ln(3)d^2}$. This yields:

1. $\min\left\{\frac{\phi}{2\ln(3)d^2}, \frac{\phi}{2\ln\left(\frac{1}{\phi}\right)}\right\} < e^{-2\ln(3)d^2}$. We start by arguing that:

$$\min\left\{\frac{\phi}{2\ln(3)d^2}, \frac{\phi}{2\ln\left(\frac{1}{\phi}\right)}\right\} < e^{-2\ln(3)d^2} \iff \frac{\phi}{2\ln\left(\frac{1}{\phi}\right)} < e^{-2\ln(3)d^2}. \qquad (18)$$

The $\impliedby$ direction is trivial, so we focus on the $\implies$ direction.

We assume that $\min\left\{\frac{\phi}{2\ln(3)d^2}, \frac{\phi}{2\ln\left(\frac{1}{\phi}\right)}\right\} < e^{-2\ln(3)d^2}$ but $\frac{\phi}{2\ln\left(\frac{1}{\phi}\right)} \geq e^{-2\ln(3)d^2}$. Then, it must be the case that $\frac{\phi}{2\ln(3)d^2} < e^{-2\ln(3)d^2}$. Setting $y := \frac{1}{\phi}$, the above system of inequalities can be written in the form:

$$\begin{cases} \frac{e^{2\ln(3)d^2}}{2} \geq y\ln(y) \\ y > \frac{e^{2\ln(3)d^2}}{2\ln(3)d^2} \end{cases}.$$

The function $y\ln(y)$ is increasing for $y \geq e^{-1}$. We have $\frac{e^{2\ln(3)d^2}}{2\ln(3)d^2} \geq e^{-1}, \forall d \geq 1$. Thus, for $y > \frac{e^{2\ln(3)d^2}}{2\ln(3)d^2}$, we get:

$$\frac{e^{2\ln(3)d^2}}{2} \geq y\ln(y) > \frac{e^{2\ln(3)d^2}}{2\ln(3)d^2}\ln\left(\frac{e^{2\ln(3)d^2}}{2\ln(3)d^2}\right) \implies \ln(3)d^2 < \ln\left(2\ln(3)d^2\right).$$

However, the above inequality cannot be satisfied, which leads to a contradiction.

Consequently, we have shown (18). Since we have assumed that $\min\left\{\frac{\phi}{2\ln(3)d^2}, \frac{\phi}{2\ln\left(\frac{1}{\phi}\right)}\right\} < e^{-2\ln(3)d^2}$, it must be the case that $\frac{\phi}{2\ln\left(\frac{1}{\phi}\right)} < e^{-2\ln(3)d^2}$.

We now turn our attention again to (17). We remark that, for $\delta < e^{-2\ln(3)d^2}$, we get $\frac{\ln\left(\frac{1}{\delta}\right)}{2\ln(3)d^2} > 1$. As a result, in order to satisfy (17), it suffices to have:

$$2\delta\frac{\ln\left(\frac{1}{\delta}\right)}{2\ln(3)d^2} \leq \frac{\phi}{\ln(3)d^2} \iff \delta\ln\left(\frac{1}{\delta}\right) \leq \phi. \qquad (19)$$

Observe that the function $\delta\ln\left(\frac{1}{\delta}\right)$ is increasing for $\delta < e^{-2\ln(3)d^2} < e^{-1}$. Combined with our previous remarks, this implies that, in order to verify (19) for the values of $\delta$ we picked, it suffices to show that it holds for $\delta = \frac{\phi}{2\ln\left(\frac{1}{\phi}\right)}$. So, we have to verify the inequality:

$$\left(\frac{\phi}{2\ln\left(\frac{1}{\phi}\right)}\right)\ln\left(\frac{2\ln\left(\frac{1}{\phi}\right)}{\phi}\right) \leq \phi \iff \frac{\ln\left(2\ln\left(\frac{1}{\phi}\right)\right)}{\ln\left(\frac{1}{\phi}\right)} \leq 1.$$

Setting $z := \ln\left(\frac{1}{\phi}\right)$, this becomes equivalent to $\frac{\ln(2z)}{z} < 1$, which holds for all $z > 0 \iff \ln\left(\frac{1}{\phi}\right) > 0 \iff \phi < 1$. This last condition is satisfied by all $\phi$ such that $\frac{\phi}{2\ln\left(\frac{1}{\phi}\right)} < e^{-2\ln(3)d^2}$, so the desired result has been established.

2. $\min\left\{\frac{\phi}{2\ln(3)d^2}, \frac{\phi}{2\ln\left(\frac{1}{\phi}\right)}\right\} \geq e^{-2\ln(3)d^2}$. For this case, we consider two sub-cases, depending on how $\delta$ compares with $e^{-2\ln(3)d^2}$. We have:

   - $\delta < e^{-2\ln(3)d^2}$. As before, we argue that it suffices to have $\delta\ln\left(\frac{1}{\delta}\right) \leq \phi$. Since $\delta\ln\left(\frac{1}{\delta}\right)$ is increasing for $\delta < e^{-2\ln(3)d^2} < e^{-1}$, all we have to do is show that $e^{-2\ln(3)d^2}2\ln(3)d^2 \leq \phi$. This is an immediate consequence of our assumption that $\min\left\{\frac{\phi}{2\ln(3)d^2}, \frac{\phi}{2\ln\left(\frac{1}{\phi}\right)}\right\} \geq e^{-2\ln(3)d^2}$.

- $\delta \geq e^{-2\ln(3)d^2}$. This implies $\frac{\ln\left(\frac{1}{\delta}\right)}{2\ln(3)d^2} \leq 1$. Thus, for (17) to hold, it suffices that:

$$2\delta \leq \frac{\phi}{\ln(3)d^2} \iff \delta \leq \frac{\phi}{2\ln(3)d^2}.$$

This is satisfied, because our proposed range for $\delta$ is $\delta \leq \min\left\{\frac{\phi}{2\ln(3)d^2}, \frac{\phi}{2\ln\left(\frac{1}{\phi}\right)}\right\}$.

We have established that our proposed values of $\delta$ and $T$ imply:

$$\delta T \geq \frac{4}{3cd^2}e^{-d^2(3cT-2\ln(3))} \text{ and } 3n\delta T \leq \frac{1}{192},$$

while respecting all the constraints.

We substitute this to (15) and get $n \geq \frac{d^2}{384\alpha\varepsilon}$. Appealing to Corollary C.5 completes the proof. ∎

## C.2 Omitted Proofs from Section 3.2

*Proof of Theorem 3.2.* First, assume that $\Sigma$ is generated by Algorithm 1 (as was assumed in all of Section 3.1). Additionally, assume that there exists an $(\varepsilon, \delta)$-DP mechanism $M: \mathbb{R}^{n\times d} \to \mathbb{R}^{d\times d}$ with $\varepsilon \in [0,1]$ and $\delta \leq \mathcal{O}\left(\min\left\{\frac{1}{n}, \frac{d^2}{n\log\left(\frac{n}{d^2}\right)}\right\}\right)$ that satisfies $\underset{X,M}{\mathbb{E}}\left[\left\|\Sigma^{-\frac{1}{2}}(M(X)-\Sigma)\Sigma^{-\frac{1}{2}}\right\|_2^2\right] \leq \alpha^2$ and uses $n = o\left(\frac{d^{1.5}}{\alpha\varepsilon}\right)$ samples. Using the standard matrix-norm inequality $\|A\|_F \leq \sqrt{d}\|A\|_2, \forall A \in \mathbb{R}^{k\times\ell}$, we get:

$$\begin{aligned}
\underset{X,M}{\mathbb{E}}\left[\|M(X)-\Sigma\|_\Sigma^2\right] &= \underset{X,M}{\mathbb{E}}\left[\left\|\Sigma^{-\frac{1}{2}}(M(X)-\Sigma)\Sigma^{-\frac{1}{2}}\right\|_F^2\right] \\
&\leq d\underset{X,M}{\mathbb{E}}\left[\left\|\Sigma^{-\frac{1}{2}}(M(X)-\Sigma)\Sigma^{-\frac{1}{2}}\right\|_2^2\right] \\
&\leq d\alpha^2.
\end{aligned}$$

By Theorem 3.1 for $\alpha \to \sqrt{d}\alpha$, we get that any mechanism with the characteristics of $M$ that estimates a Gaussian covariance matrix in Mahalanobis error $\sqrt{d}\alpha$ requires $n \geq \Omega\left(\frac{d^{1.5}}{\alpha\varepsilon}\right)$ samples. This leads to a contradiction, implying the desired sample complexity lower bound. ∎

# D Omitted Proof from Section

*Proof of Theorem 4.1.* We define the following family of distributions, such that the second moment of each distribution is upper-bounded by 1. The loss function between the true distribution and the estimate is the squared $\ell_2$-distance between their means. For each $v \in \mathcal{E}_d = \{-1,1\}^d$, we define the distribution $\mathcal{D}_v$ over $\mathbb{R}^d$ as follows. Let $t > 0$ and $0 < p < 1$. For any $X \sim \mathcal{D}_v$, we have:

$$X_i = \begin{cases} v_i t, & \text{with probability } p \\ 0, & \text{with probability } 1-p \end{cases}, \forall i \in [d].$$

Then $\underset{X\sim\mathcal{D}_v}{\mathbb{E}}[X] = ptv$. We show that the second moment of this distribution is upper-bounded by $pt^2$, that is, $\underset{X\sim\mathcal{D}_v}{\mathbb{E}}\left[\langle X - ptv, u\rangle^2\right] \leq pt^2, \forall u \in \mathbb{S}^{d-1}$. Note that, for any $i \in [d]$, we have $\mathbb{E}\left[(X_i - ptv_i)^2\right] = p(1-p)t^2$. We omit the subscript of the expectation when the context is clear.

We have:

$$
\mathbb{E}\left[\langle X - ptv, u\rangle^2\right] = \mathbb{E}\left[\left(\sum_{i=1}^{d} (X_i - ptv_i)\, u_i\right)^2\right]
$$

$$
= \sum_{i=1}^{d} \mathbb{E}\left[(X_i - ptv_i)^2\right] u_i^2 + \sum_{i\neq j} \mathbb{E}\left[X_i - ptv_i\right] \mathbb{E}\left[X_j - ptv_j\right] u_i u_j
$$

$$
= \sum_{i=1}^{d} p\,(1-p)\, t^2 u_i^2
$$

$$
= p\,(1-p)\, t^2
$$

$$
\leq pt^2.
$$

We want this to be upper-bounded by 1:

$$
pt^2 = 1. \tag{20}
$$

Next, for any $u, v \in \mathcal{E}_d$, we bound the loss function between $\mathcal{D}_{v_1}$ and $\mathcal{D}_{v_2}$. As mentioned before, we define $\ell\left(\theta\left(\mathcal{D}_u\right), \theta\left(\mathcal{D}_v\right)\right) = \|\theta\left(\mathcal{D}_u\right) - \theta\left(\mathcal{D}_v\right)\|_2^2$. We have the following:

$$
\|\theta\left(\mathcal{D}_u\right) - \theta\left(\mathcal{D}_v\right)\|_2^2 = \sum_{i=1}^{d} p^2 t^2 \left(v_i - u_i\right)^2
$$

$$
= p^2 t^2 \sum_{i:\, u_i \neq v_i} \left(v_i - u_i\right)^2
$$

$$
= 4p^2 t^2 \sum_{i=1}^{d} \mathbb{1}\left\{v_i \neq u_i\right\}.
$$

Using the notation in Lemma A.4, we have $\tau = 2p^2 t^2$. Using the same lemma, we have:

$$
R\left(\mathcal{P}, \ell, \varepsilon, \delta\right) \geq \frac{d\tau}{2} \cdot \left(0.9 e^{-10\varepsilon D} - 10\delta D\right)
$$

$$
\geq \frac{d\tau}{2} \cdot \left[0.9\left(1 - 10\varepsilon D\right) - 10\delta D\right]
$$

$$
= \frac{d\tau}{2} \cdot \left[0.9 - D\left(9\varepsilon + 10\delta\right)\right]
$$

$$
\geq \frac{d\tau}{2} \cdot \left[0.9 - 10D\left(\varepsilon + \delta\right)\right].
$$

Setting $R\left(\mathcal{P}, \ell, \varepsilon, \delta\right) \leq \alpha^2$ and rearranging the above, we get:

$$
D \geq \frac{1}{10\left(\varepsilon + \delta\right)} \cdot \left(0.9 - \frac{2\alpha^2}{d\tau}\right). \tag{21}
$$

Now, from Equation (20), $\tau = 2p^2 t^2 = 2p$. Using this and (21), and setting $p = \frac{2\alpha^2}{d}$, we have the following.

$$
D \geq \frac{0.04}{\varepsilon + \delta}. \tag{22}
$$

For $i \in [d]$, we define the mixture distributions $p_{+i}$ and $p_{-i}$ as in Lemma A.4. We define a coupling $(X, Y)$ between $p_{+i}$ and $p_{-i}$ as follows. For every row $X_j$ in $X$, if $X_j^i \neq 0$, then $Y_j^i = -X_j^i$, otherwise $Y_j^i = X_j^i$. For all other coordinates $k \neq i$, $Y_j^k = X_j^k$. Notice that, for all $k \neq i$, the distributions for coordinate $k$ for both $p_{+i}$ and $p_{-i}$ are identical because all coordinates are independent. Therefore, by setting $Y_j^k = X_j^k$, these coordinates of $X_j$ and $Y_j$ have identical distributions for all $j \in [n]$. The coordinate $i$ of $X_j$ is $t$ with probability $p$, and 0 with probability $1 - p$. Hence, the coordinate $i$ of $Y_j$ is $-t$ with probability $p$, and 0 with probability $1 - p$. This is

true for all $j \in [n]$. Since all $n$ rows in both $X$ (hence, in $Y$) are chosen independently at random, the distribution of $X$ is $p_{+i}$ and the distribution of $Y$ is $p_{-i}$. Therefore, $(X, Y)$ is a valid coupling of $p_{+i}$ and $p_{-i}$. Next, we have to determine the value of $D = \mathbb{E}\left[d_{\mathrm{Ham}}(X, Y)\right]$. Note that for each $j \in [n]$, $X_j$ can only differ from $Y_j$ when $X_j^i \neq 0$. This can only happen with probability $p$. Thus, by linearity of expectation, $D = np$.

Finally, using the fact that $p = \frac{2\alpha^2}{d}$ and $t = \frac{\sqrt{d}}{\sqrt{2}\alpha}$, and substituting this in Inequality (22), we have:

$$np \geq \frac{0.04}{\varepsilon + \delta} \implies n \geq \frac{d}{50\left(\varepsilon + \delta\right)\alpha^2}.$$

When $\varepsilon \geq \delta$, this gives us the desired lower bound. $\blacksquare$

## E  Facts from Linear Algebra and Geometry

In this appendix we collect some results from linear algebra and geometry which are referenced throughout this work.

**Definition E.1** ($\varepsilon$-net)**.** Let $(T, d)$ be a metric space. Consider a subset $K \subset T$ and let $\varepsilon > 0$. A subset $\mathcal{N}_\varepsilon \subseteq K$ is called an *$\varepsilon$-net* of $K$ if the following holds:

$$\forall x \in K, \exists y \in \mathcal{N}_\varepsilon : d(x, x_0) \leq \varepsilon.$$

**Definition E.2** (Covering Number)**.** The smallest possible cardinality of an $\varepsilon$-net of $K$ is called the *covering number* of $K$ and denoted by $\mathcal{N}(K, d, \varepsilon)$. Equivalently, $\mathcal{N}(K, d, \varepsilon)$ is the smallest number of closed balls with centers in $K$ and radii $\varepsilon$ whose union covers $K$.

**Fact E.3** (Covering numbers of $\mathbb{S}^{d-1}$)**.** *The covering numbers of the unit Euclidean sphere $\mathbb{S}^{d-1}$ satisfy the following for any $\varepsilon > 0$:*

$$\left(\frac{1}{\varepsilon}\right)^d \leq \mathcal{N}\left(\mathbb{S}^{d-1}, \varepsilon\right) \leq \left(1 + \frac{1}{\varepsilon}\right)^d.$$

**Fact E.4** (Exercise 4.4.2 from [Ver18])**.** *Let $x \in \mathbb{R}^d$ and $\mathcal{N}_\varepsilon$ be an $\varepsilon$-net on $\mathbb{S}^{d-1}$. It holds that:*

$$\sup_{y \in \mathcal{N}_\varepsilon} |\langle x, y \rangle| \leq \|x\|_2 \leq \frac{1}{1 - \varepsilon} \cdot \sup_{y \in \mathcal{N}_\varepsilon} |\langle x, y \rangle|.$$

**Fact E.5** (Spectral Norm of Kronecker Product)**.** *Let $A \in \mathbb{R}^{n \times m}$ and $B \in \mathbb{R}^{k \times \ell}$. Then, it holds that $\|A \otimes B\|_2 = \|A\|_2 \|B\|_2$.*

**Fact E.6** (Frobenius Norm of Matrix Product)**.** *Let $M, N \in \mathbb{R}^{d \times d}$. Then, $\|MN\|_F \leq \|M\|_2 \|N\|_F$, and $\|MN\|_F \leq \|N\|_2 \|M\|_F$.*

For a proof of the first of the two inequalities, see Lemma 4.26 of [DKK$^+$16]. The proof for the second one is analogous but we have to work with the rows of $M$ instead of the columns of $N$.

**Lemma E.7.** *Let $A, \Sigma \in \mathbb{R}^{d \times d}$ be non-singular matrices and let $X \in \mathbb{R}^{d \times d}$. Then, it holds that $\|X\|_{A\Sigma A^\top} = \left\|A^{-1} X \left(A^\top\right)^{-1}\right\|_\Sigma.$*

For a proof of this statement for symmetric $A$ (the proof does not change), we refer the reader to Section 2.2 of [Li19].

**Lemma E.8.** *Let $A, B \in \mathbb{R}^{d \times d}$ be symmetric positive definite matrices. Then, it holds that:*

$$\left\|A - A^{\frac{1}{2}} B^{\frac{1}{2}} A^{-1} B^{\frac{1}{2}} A^{\frac{1}{2}}\right\|_A = \|A - B\|_A.$$

*Proof.* We have that:

$$
\left\| A - A^{\frac{1}{2}} B^{\frac{1}{2}} A^{-1} B^{\frac{1}{2}} A^{\frac{1}{2}} \right\|_A^2 = \left\| \mathbb{I} - B^{\frac{1}{2}} A^{-1} B^{\frac{1}{2}} \right\|_F^2
$$

$$
= \operatorname{tr} \left( \left( \mathbb{I} - B^{\frac{1}{2}} A^{-1} B^{\frac{1}{2}} \right)^2 \right)
$$

$$
= \operatorname{tr} \left( \mathbb{I} - 2 B^{\frac{1}{2}} A^{-1} B^{\frac{1}{2}} + B^{\frac{1}{2}} A^{-1} B A^{-1} B^{\frac{1}{2}} \right)
$$

$$
\overset{(a)}{=} \operatorname{tr} \left( \mathbb{I} - 2 A^{-\frac{1}{2}} B A^{-\frac{1}{2}} + \left( A^{-\frac{1}{2}} B A^{-\frac{1}{2}} \right)^2 \right)
$$

$$
= \operatorname{tr} \left( \left( \mathbb{I} - A^{-\frac{1}{2}} B A^{-\frac{1}{2}} \right)^2 \right)
$$

$$
= \left\| \mathbb{I} - A^{-\frac{1}{2}} B A^{-\frac{1}{2}} \right\|_F^2
$$

$$
= \| A - B \|_A^2,
$$

where in $(a)$ we use the cyclic property of the trace in individual terms. ∎

**Lemma E.9.** *Let $\Sigma_0, \Sigma_1 \in \mathbb{R}^{d \times d}$ be symmetric positive definite matrices. If $\|\Sigma_1 - \Sigma_0\|_{\Sigma_0} < \frac{1}{2}$, it holds that $\|\Sigma_1 - \Sigma_0\|_{\Sigma_1} \leq 2 \|\Sigma_1 - \Sigma_0\|_{\Sigma_0}$.*

*Proof.* By the definition of the Mahalanobis norm, we have:

$$
\|\Sigma_1 - \Sigma_0\|_{\Sigma_0}^2 = \left\| \mathbb{I} - \Sigma_0^{-\frac{1}{2}} \Sigma_1 \Sigma_0^{-\frac{1}{2}} \right\|_F
$$

$$
= \operatorname{tr} \left( \left( \mathbb{I} - \Sigma_0^{-\frac{1}{2}} \Sigma_1 \Sigma_0^{-\frac{1}{2}} \right)^2 \right)
$$

$$
= \operatorname{tr} \left( \mathbb{I} - 2 \Sigma_0^{-\frac{1}{2}} \Sigma_1 \Sigma_0^{-\frac{1}{2}} + \Sigma_0^{-\frac{1}{2}} \Sigma_1 \Sigma_0^{-1} \Sigma_1 \Sigma_0^{-\frac{1}{2}} \right)
$$

$$
= d - 2 \operatorname{tr} \left( \Sigma_0^{-1} \Sigma_1 \right) + \operatorname{tr} \left( \left( \Sigma_0^{-1} \Sigma_1 \right)^2 \right). \tag{23}
$$

Similarly, we have:

$$
\|\Sigma_1 - \Sigma_0\|_{\Sigma_1}^2 = d - 2 \operatorname{tr} \left( \Sigma_1^{-1} \Sigma_0 \right) + \operatorname{tr} \left( \left( \Sigma_1^{-1} \Sigma_0 \right)^2 \right). \tag{24}
$$

The eigenvalues of the matrix $\Sigma_0^{-1} \Sigma_1$ are positive. Indeed, the previous matrix can be written as $\Sigma_0^{-\frac{1}{2}} \left( \Sigma_0^{-\frac{1}{2}} \Sigma_1 \Sigma_0^{-\frac{1}{2}} \right) \Sigma_0^{\frac{1}{2}}$, implying that it is similar to the symmetric matrix $\Sigma_0^{-\frac{1}{2}} \Sigma_1 \Sigma_0^{-\frac{1}{2}}$. This last matrix is positive definite, since $u^\top \Sigma_0^{-\frac{1}{2}} \Sigma_1 \Sigma_0^{-\frac{1}{2}} u = \left( \Sigma_0^{-\frac{1}{2}} u \right)^\top \Sigma_1 \left( \Sigma_0^{-\frac{1}{2}} u \right)$ and $\Sigma_1 \succ 0$. Thus, if $\lambda_1, \ldots, \lambda_d > 0$ are the eigenvalues of $\Sigma_0^{-1} \Sigma_1$, (23) and (24) can be written as:

$$
\|\Sigma_1 - \Sigma_0\|_{\Sigma_0}^2 = d - 2 \cdot \sum_{i=1}^d \lambda_i + \sum_{i=1}^d \lambda_i^2 = \sum_{i=1}^d (\lambda_i - 1)^2,
$$

$$
\|\Sigma_1 - \Sigma_0\|_{\Sigma_1}^2 = d - 2 \cdot \sum_{i=1}^d \frac{1}{\lambda_i} + \sum_{i=1}^d \frac{1}{\lambda_i^2} = \sum_{i=1}^d \left( \frac{1}{\lambda_i} - 1 \right)^2.
$$

Therefore, proving the desired upper bound reduces to the following maximization problem:

$$
\max_{\lambda_i} \quad \sum_{i=1}^d \left( \frac{1}{\lambda_i} - 1 \right)^2
$$

$$
\text{s.t.} \quad \sum_{i=1}^d (\lambda_i - 1)^2 = \|\Sigma_1 - \Sigma_0\|_{\Sigma_0}^2 < \frac{1}{4},
$$

where we have omitted the constraint $\lambda_i > 0$, since we necessarily have $\lambda_i \geq 1 - \|\Sigma_1 - \Sigma_0\|_{\Sigma_0} > \frac{1}{2}, \forall i \in [d]$. The above problem is solvable exactly via the KKT conditions, yielding:

$$\|\Sigma_1 - \Sigma_0\|_{\Sigma_1} \leq \frac{\|\Sigma_1 - \Sigma_0\|_{\Sigma_0} \sqrt{d}}{\sqrt{d} - \|\Sigma_1 - \Sigma_0\|_{\Sigma_0}} \leq 2 \|\Sigma_1 - \Sigma_0\|_{\Sigma_0},$$

for $\|\Sigma_1 - \Sigma_0\|_{\Sigma_0} < \frac{1}{2}$ and $d \geq 1$. ∎

**Theorem E.10** (Gershgorin Circle Theorem (see [Ger31])). *Let $A \in \mathbb{R}^{d \times d}$. For any eigenvalue $\lambda$ of $A$, there exists an $i \in [d]$, such that:*

$$|\lambda - a_{ii}| \leq \sum_{j \neq i} |a_{ij}|.$$

# F Facts from Probability & Statistics

In this appendix we collect some results from probability and statistics which are referenced throughout this work.

**Fact F.1.** *The Bernoulli distribution* $\mathrm{Be}(p)$ *is an exponential family with support* $S = \{0, 1\}$ *and* $h(x) = 1, \eta = \ln\left(\frac{1-p}{p}\right), T(x) = x, Z(\eta) = \ln(1 + e^\eta)$.

**Fact F.2.** *The $d$-dimensional Gaussian distribution with unit covariance and unknown mean* $\mathcal{N}(\mu, \mathbb{I})$ *is an exponential family with support* $S = \mathbb{R}^d$ *and* $h(x) = \frac{1}{(2\pi)^{\frac{d}{2}}} e^{-\frac{\|x\|_2^2}{2}}, \eta = \mu, T(x) = x, Z(\eta) = \frac{\|\eta\|_2^2}{2}$.

**Fact F.3.** *The $d$-dimensional Gaussian distribution with mean $0$ and unknown covariance* $\mathcal{N}(0, \Sigma)$ *is an exponential family with support* $S = \mathbb{R}^d$ *and* $h(x) = 1, \eta = \left(\Sigma^{-1}\right)^\flat, T(x) = -\frac{1}{2}\left(xx^\top\right)^\flat, Z(\eta) = \frac{d}{2}\ln(2\pi) - \frac{1}{2}\ln\left(\det\left(\eta^\#\right)\right)$.

**Fact F.4** (Cauchy-Schwarz). *We consider the following variants of the Cauchy-Schwarz inequality:*

1. *For every $x, y \in \mathbb{R}^d$, we have $|\langle x, y \rangle| \leq \|x\|_2 \|y\|_2$.*

2. *Let $X, Y$ be random variables over $\mathbb{R}$. Then:*

$$\mathbb{E}[|XY|] \leq \sqrt{\mathbb{E}[X^2]}\sqrt{\mathbb{E}[Y^2]},$$

   *assuming all the quantities involved are well-defined.*

**Fact F.5.** *We denote the* complementary error function *by:*

$$\mathrm{erfc}(x) := \frac{2}{\sqrt{\pi}} \int_x^\infty e^{-t^2}\, dt, x \geq 0.$$

*It holds that* $\mathrm{erfc}(x) \leq e^{-x^2}, \forall x \geq 0$.

**Fact F.6** (Lemma 1 of [LM00]). *If $X \sim \chi^2(k)$, and $\beta \in [0, 1]$, then:*

$$\mathbb{P}\left[X - k \geq 2\sqrt{k \log\left(\frac{1}{\beta}\right)} + 2\log\left(\frac{1}{\beta}\right)\right] \leq \beta,$$

*and:*

$$\mathbb{P}\left[k - X \geq 2\sqrt{k \log\left(\frac{1}{\beta}\right)}\right] \leq \beta.$$

*Equivalently, the above can be written as:*

$$\mathbb{P}[X \geq t] \leq e^{-\frac{\left(\sqrt{2t-k} - \sqrt{k}\right)^2}{4}}, \forall t \geq k,$$

*and:*

$$\mathbb{P}\left[X \le t\right] \le e^{-\frac{(k-t)^2}{4}}, \forall t \le k.$$

*Thus, if* $Y \sim \mathcal{N}\left(0, \mathbb{I}\right)$, *then* $\mathbb{P}\left[\|Y\|_2^2 \ge d + 2\sqrt{d \log\left(\frac{1}{\beta}\right)} + 2\log\left(\frac{1}{\beta}\right)\right] \le \beta.$

**Fact F.7** (Gaussian Hanson-Wright Inequality - Theorem 1 and Proposition 1 from [Mos21])**.** *Let* $X \in \mathbb{R}^d$ *be a random vector with* $X \sim \mathcal{N}\left(0, \mathbb{I}\right)$. *For any non-zero symmetric matrix* $A \in \mathbb{R}^{d \times d}$:

$$\mathbb{P}\left[\left|X^\top A X - \mathbb{E}\left[X^\top A X\right]\right| \ge t\right] \le 2\exp\left(-c\min\left\{\frac{t^2}{\|A\|_F^2}, \frac{t}{\|A\|_2}\right\}\right), \forall t \ge 0,$$

*where* $c$ *is an absolute constant with* $c \approx 0.036425$.

# G   Basic Applications: Recovering Existing Lower Bounds

We show how to use Theorem 2.4 to recover existing lower bounds from [KLSU19]: $(1)$ mean estimation of binary product distributions, and $(2)$ mean estimation of high-dimensional Gaussians. Note that both classes of distributions are exponential families. The error metric for these lower bounds would be the mean-squared-error (MSE), as opposed to constant-probability-error, as in [KLSU19], but the bounds could be converted to constant-probability bounds at the cost of a $\log(d)$ factor in the sample complexity (as was sketched in Section 1.1).

## G.1   Mean Estimation of Binary Product Distributions

We start by stating the theorem for mean estimation of binary product distributions over $\{0, 1\}^d$.

**Theorem G.1** (Product Distribution Mean Estimation)**.** *There exists a distribution* $\mathcal{D}$ *over vectors* $p \in \left[\frac{1}{3}, \frac{2}{3}\right]^d$ *such that, given* $p \sim \mathcal{D}$ *and* $n$ *independent samples* $X \coloneqq (X_1, \ldots, X_n)$ *from a binary product distribution* $P$ *over* $\{0, 1\}^d$ *with mean* $p$, *for any* $\alpha = \mathcal{O}\left(\sqrt{d}\right)$ *and any* $(\varepsilon, \delta)$-*DP mechanism* $M\colon \{0, 1\}^{n \times d} \to \left[\frac{1}{3}, \frac{2}{3}\right]^d$ *with* $\varepsilon, \delta \in [0, 1]$, *and* $\delta = \mathcal{O}\left(\frac{1}{n}\right)$ *that satisfies* $\underset{X,M}{\mathbb{E}}\left[\|M(X) - p\|_2^2\right] \le \alpha^2$, *it holds that* $n = \Omega\left(\frac{d}{\alpha\varepsilon}\right)$.

Fact F.1 establishes that Bernoulli distributions are an exponential family. Using that, we have that the probability mass function of binary product distributions can be written as $p_\eta = h(x)\exp\left(\eta^\top T(x) - Z(\eta)\right), \forall x \in \{0, 1\}^d$ with:

$$h(x) = 1,$$
$$T(x) = x,$$
$$\eta = \left(\ln\left(\frac{1 - p_1}{p_1}\right), \ldots, \ln\left(\frac{1 - p_d}{p_d}\right)\right)^\top,$$
$$Z(\eta) = \prod_{j \in [d]} \ln\left(1 + e^{\eta_j}\right).$$

Our process to generate $\eta$ involves drawing independently $\eta_j \sim \mathcal{U}\left[\pm\ln(2)\right]$. We have that $\eta_j = \ln\left(\frac{1}{p_j} - 1\right) \iff p_j = \frac{1}{1 + e^{\eta_j}}$, yielding $\eta_j \in \left[\pm\ln(2)\right] \iff p_j \in \left[\frac{1}{3}, \frac{2}{3}\right]$. Thus, we have $I_j = \left[\pm\ln(2)\right], R_j = 2\ln(2), \forall j \in [d] \implies \|R\|_2^2 = 4\ln^2(2)d, \|R\|_\infty = 2\ln(2)$, and $m = 0$.

We now show how to reduce estimating $\eta$ with an $\ell_2$-guarantee under $(\varepsilon, \delta)$-DP to estimating $p$ with an $\ell_2$-guarantee under the same constraint.

**Lemma G.2.** *Let* $p \in \left[\frac{1}{3}, \frac{2}{3}\right]^d$ *be a randomly generated vector and let* $X \sim P^{\otimes n}$ *be a dataset drawn from a binary product distribution over* $\{0, 1\}^d$ *with mean* $p$. *If* $M\colon \{0, 1\}^{n \times d} \to \left[\frac{1}{3}, \frac{2}{3}\right]^d$ *is an* $(\varepsilon, \delta)$-*DP mechanism satisfying* $\underset{X,M}{\mathbb{E}}\left[\|M(X) - p\|_2^2\right] \le \alpha^2 \le \frac{d}{9}$, *then there exists a* $T_M\colon \{0, 1\}^{n \times d} \to \left[\pm\ln(2)\right]^d$ *that is also* $(\varepsilon, \delta)$-*DP and satisfies* $\underset{X,T_M}{\mathbb{E}}\left[\|T_M(X) - \eta\|_2^2\right] \le \frac{81}{4}\alpha^2$.

*Proof.* We define $T_M$ to be $T_{M,j}(X) = \ln\left(\frac{1-M_j(X)}{M_j(X)}\right), \forall j \in [d]$. $T_M$ satisfies $(\varepsilon, \delta)$-DP by Lemma A.3 and the only randomness employed by $T_M$ is that of $M$. Consider now the function $g\colon (0,1) \to \mathbb{R}$ with $g(x) = \ln\left(\frac{1}{x} - 1\right)$ and $g'(x) = -\frac{1}{x(1-x)}$. We have $\eta_j = g(p_j)$ and $T_{M,j}(X) = g(T_j(X))$. Observe now that, by the Mean Value Theorem, we have for some $\xi_j$ between $\eta_j$ and $M_j(X)$ (implying that $\xi_j \in \left[\frac{1}{3}, \frac{2}{3}\right]$):

$$|g(M_j(X)) - g(p_j)| = |g'(\xi_j)|\,|M_j(X) - p_j| \le \frac{9}{2}|M_j(X) - p_j|.$$

Applying this coordinate-wise, we get $\underset{X, T_M}{\mathbb{E}}\left[\|T_M(X) - \eta\|_2^2\right] \le \frac{81}{4}\underset{X,M}{\mathbb{E}}\left[\|M(X) - p\|_2^2\right]$, yielding the desired result. ∎

We now present a restatement of the previous lemma from the point of view of lower bounds.

**Corollary G.3.** *Let $p \in \left[\frac{1}{3}, \frac{2}{3}\right]^d$ be a randomly generated vector and let $X \sim P^{\otimes n}$ be a dataset drawn from a binary product distribution over $\{0,1\}^d$ with mean $p$. If any $(\varepsilon, \delta)$-DP mechanism $T\colon \{0,1\}^{n \times d} \to [\pm \ln(2)]^d$ satisfying $\underset{X,T}{\mathbb{E}}\left[\|T(X) - \eta\|_2^2\right] \le \frac{81}{4}\alpha^2 \le \frac{9}{4}d$ requires at least $n \ge n_\eta$ samples, the same sample complexity lower bound holds for any $(\varepsilon, \delta)$-DP mechanism $M\colon \{0,1\}^{n \times d} \to \left[\frac{1}{3}, \frac{2}{3}\right]^d$ that satisfies $\underset{X,M}{\mathbb{E}}\left[\|M(X) - p\|_2^2\right] \le \alpha^2$.*

In contrast to other lower bounds proven in this work using Theorem 2.4, we do not need to study the concentration properties of $\|T(X_i) - \mu_T\|_2 = \|X_i - p\|_2$. Indeed, since $T(X) = X \in \{0,1\}^d$, it suffices to pick a large enough threshold $T > 0$ so that $\underset{X_i}{\mathbb{P}}\left[\|X_i - p\|_2 > \frac{t}{2\ln^3(2)\sqrt{d}}\right] = 0, \forall t \ge T$.

Based on the above, we are now ready to prove the main theorem of this section.

*Proof of Theorem G.1.* Let $M\colon \{0,1\}^{n\times d} \to [\pm \ln(2)]^d$ be an $(\varepsilon, \delta)$-DP mechanism with:

$$\underset{X,M}{\mathbb{E}}\left[\|M(X) - \eta\|_2^2\right] \le \frac{81}{4}\alpha^2 \le \frac{\ln^2(2)d}{6} = \frac{\|R\|_2^2}{24}.$$

By Theorem 2.4 for product distributions with $\eta_j \in [\pm \ln(2)], \forall j \in [d]$, we get:

$$n\left\{2\delta T + 2\varepsilon\underset{\eta}{\mathbb{E}}\left[\sqrt{\underset{X_{\sim i},M}{\mathbb{E}}[s^\top \Sigma_T s]}\right] + 2\int_T^\infty \underset{X_i}{\mathbb{P}}\left[\|X_i - p\|_2 > \frac{t}{2\ln^3(2)\sqrt{d}}\right]dt\right\} \ge \frac{\ln^2(2)d}{6}. \tag{25}$$

Observe that, since we have a product distribution and $T(x) = x$, it holds that $\mu_T = p$ and:

$$\Sigma_T = \Sigma = \operatorname{diag}\{p_1(1-p_1), \ldots, p_d(1-p_d)\} \preceq \frac{1}{4}\mathbb{I},$$

for $p_i \in \left[\frac{1}{3}, \frac{2}{3}\right], \forall i \in [d]$.

Thus, we have:

$$\underset{X_{\sim i},M}{\mathbb{E}}[s^\top \Sigma_T s] = \frac{1}{4}\underset{X_{\sim i},M}{\mathbb{E}}\left[\|s\|_2^2\right] \le \frac{\|R\|_\infty^4}{64}\alpha^2 = \frac{\ln^4(2)\alpha^2}{4}. \tag{26}$$

Furthermore, we remark that $\|X_i - p\|_2 \le \frac{2}{3}\sqrt{d}$. Thus, setting $T = \frac{4\ln^3(2)d}{3}$ yields:

$$\int_T^\infty \underset{X_i}{\mathbb{P}}\left[\|X_i - p\|_2 > \frac{t}{2\ln^3(2)\sqrt{d}}\right]dt = 0. \tag{27}$$

Substituting (26) and (27) into (25), we get:

$$n\left(\frac{8\ln^3(2)d\delta}{3} + \frac{\ln^2(2)\alpha\varepsilon}{2}\right) \ge \frac{\ln^2(2)d}{6}.$$

Setting $\delta \le \frac{1}{32\ln(2)n}$ results in $n \ge \frac{d}{12\alpha\varepsilon}$, so appealing to Corollary G.3 completes the proof. ∎

## G.2 Mean Estimation of High-Dimensional Gaussians

Our techniques also apply to recovering lower bounds for mean estimation of a Gaussian with known covariance. By Fact F.2, we have that this class of distributions is an exponential family with $\eta = \mu$ and $T(x) = x$ (implying that $\mu_T = \mu, \Sigma_T = \Sigma$). Therefore, we do not need to resort to a reduction-based approach (as we did with binary product distributions).

**Theorem G.4** (Gaussian Mean Estimation)**.** *Given $\mu \sim \mathcal{U}\left([\pm 1]^d\right)$ and $X \sim \mathcal{N}(\mu, \mathbb{I})^{\otimes n}$, for any $\alpha = \mathcal{O}\left(\sqrt{d}\right)$ and any $(\varepsilon, \delta)$-DP mechanism $M \colon \mathbb{R}^{n \times d} \to [\pm 1]^d$ with $\varepsilon, \delta \in [0, 1]$, and $\delta \leq \mathcal{O}\left(\min\left\{\frac{1}{n}, \frac{\sqrt{d}}{n\sqrt{\log\left(\frac{n}{\sqrt{d}}\right)}}\right\}\right)$ that satisfies $\underset{X, M}{\mathbb{E}}\left[\|M(X) - \mu\|_2^2\right] \leq \alpha^2$, it holds that $n = \Omega\left(\frac{d}{\alpha\varepsilon}\right)$.*

Before proving Theorem G.4, we start with a lemma that is concerned with upper-bounding the term involving the tail probability of $\|T(X_i) - \mu_T\|_2$. The lemma does not rely on the approach of Remark B.3, instead using the concentration properties of the $\chi^2$-distribution (Fact F.6).

**Lemma G.5.** *Let $\mu \in [\pm 1]^d$, and $X_i \sim \mathcal{N}(\mu, \mathbb{I})$. Assuming that $T \geq 2d$, we have:*

$$\int_T^\infty \underset{X_i}{\mathbb{P}}\left[\|T(X_i) - \mu_T\|_2 > \frac{t}{2\sqrt{d}}\right] dt \leq \sqrt{2\pi d}\, e^{-\frac{\left(\sqrt{T^2 - 2d^2} - \sqrt{2}d\right)^2}{8d}}.$$

*Proof.* Since $X_i \sim \mathcal{N}(\mu, \mathbb{I})$, we have $T(X_i) - \mu_T = X_i - \mu$. We observe that $X_i - \mu \sim \mathcal{N}(0, \mathbb{I}) \implies \|X_i - \mu\|_2^2 \sim \chi^2(d)$. By our assumption that $T \geq 2d \implies \frac{t^2}{4d} \geq \frac{T^2}{4d} \geq d$, we can apply Fact F.6 with $k = d$. This yields:

$$\int_T^\infty \underset{X_i}{\mathbb{P}}\left[\|X_i - \mu\|_2^2 > \frac{t^2}{4d}\right] dt \leq \int_T^\infty \exp\left[-\frac{1}{4}\left(\sqrt{\frac{t^2}{2d} - d} - \sqrt{d}\right)^2\right] dt$$

$$\overset{z^2 = \frac{1}{4}\left(\sqrt{\frac{t^2}{2d} - d} - \sqrt{d}\right)^2}{=} 2\sqrt{d} \int_{\frac{\sqrt{\frac{T^2}{2d} - d} - \sqrt{d}}{2}}^\infty \frac{2z + \sqrt{d}}{\sqrt{z^2 + \left(z + \sqrt{d}\right)^2}} e^{-z^2}\, dz$$

$$\overset{(a)}{\leq} 2\sqrt{2d} \int_{\frac{\sqrt{\frac{T^2}{2d} - d} - \sqrt{d}}{2}}^\infty e^{-z^2}\, dz$$

$$= \sqrt{2\pi d}\,\mathrm{erfc}\left(\frac{\sqrt{\frac{T^2}{2d} - d} - \sqrt{d}}{2}\right)$$

$$\overset{(b)}{\leq} \sqrt{2\pi d}\, e^{-\frac{\left(\sqrt{\frac{T^2}{2d} - d} - \sqrt{d}\right)^2}{4}}$$

$$= \sqrt{2\pi d}\, e^{-\frac{\left(\sqrt{T^2 - 2d^2} - \sqrt{2}d\right)^2}{8d}},$$

where $(a)$ is by the inequality $a + b \leq \sqrt{2}\sqrt{a^2 + b^2}$ for $a = z, b = z + \sqrt{d}$, and $(b)$ is by Fact F.5. $\blacksquare$

We are now ready to prove the main theorem of this section.

*Proof of Theorem G.4.* Let $M \colon \mathbb{R}^{n \times d} \to [\pm 1]^d$ be an $(\varepsilon, \delta)$-DP mechanism with:

$$\underset{X, M}{\mathbb{E}}\left[\|M(X) - \mu\|_2^2\right] \leq \alpha^2 \leq \frac{d}{6} = \frac{\|R\|_2^2}{24}.$$

By Theorem 2.4 for Gaussians $\mathcal{N}\left(\mu, \mathbb{I}\right)$ with $\mu^j \in [\pm 1], \forall j \in [d]$, we get:

$$n\left\{2\delta T + 2\varepsilon\mathbb{E}_{\mu}\left[\sqrt{\mathbb{E}_{X_{\sim i}, M}\left[\|s\|_2^2\right]}\right] + 2\int_T^\infty \mathbb{P}_{X_i}\left[\|X_i - \mu\|_2 > \frac{t}{2\sqrt{d}}\right]dt\right\} \geq \frac{d}{6}. \tag{28}$$

We have:

$$\mathbb{E}_{X_{\sim i}, M}\left[\|s\|_2^2\right] \leq \frac{\|R\|_\infty^4}{16}\alpha^2 = \alpha^2. \tag{29}$$

We assume that $T \geq 2d$, as is required by Lemma G.5. Later in the proof, when we pick a specific value for $T$, we will verify that this condition is satisfied. Substituting based on that lemma and (29) into (28), we get:

$$n\left[2\delta T + 2\alpha\varepsilon + 2\sqrt{2\pi}de^{-\frac{\left(\sqrt{T^2 - 2d^2} - \sqrt{2}d\right)^2}{8d}}\right] \geq \frac{d}{6}. \tag{30}$$

It remains to set the values of $T$ and $\delta$ appropriately so that:

$$\delta T \geq 2\sqrt{2\pi}de^{-\frac{\left(\sqrt{T^2 - 2d^2} - \sqrt{2}d\right)^2}{8d}} \quad \text{and} \quad 3n\delta T \leq \frac{d}{12}.$$

The first of these two conditions yields:

$$Te^{\frac{\left(\sqrt{T^2 - 2d^2} - \sqrt{2}d\right)^2}{8d}} \geq \frac{2\sqrt{2\pi}d}{\delta}.$$

We set $T = 2\sqrt{d}\sqrt{\ln\left(\frac{1}{\delta}\right) + \left(\sqrt{\ln\left(\frac{1}{\delta}\right)} + \sqrt{d}\right)^2}$. Since $\delta \leq 1$, we have $\ln\left(\frac{1}{\delta}\right) \geq 0 \implies T \geq 2d$, thus satisfying the condition of Lemma G.5. Substituting to the previous yields:

$$2\sqrt{d}\sqrt{\ln\left(\frac{1}{\delta}\right) + \left(\sqrt{\ln\left(\frac{1}{\delta}\right)} + \sqrt{d}\right)^2} \cdot \frac{1}{\delta} \geq \frac{2\sqrt{2\pi}d}{\delta}$$

$$\iff \sqrt{\ln\left(\frac{1}{\delta}\right) + \left(\sqrt{\ln\left(\frac{1}{\delta}\right)} + \sqrt{d}\right)^2} \geq \sqrt{2\pi}$$

$$\iff \ln\left(\frac{1}{\delta}\right) + \sqrt{d}\sqrt{\ln\left(\frac{1}{\delta}\right)} + \frac{1}{2}\left(d - 2\pi\right) \geq 0.$$

Setting $x \coloneqq \sqrt{\ln\left(\frac{1}{\delta}\right)} \geq 0$, the above is a quadratic inequality with respect to $x$. Depending on whether the trinomial in the LHS has any positive roots, this might imply a constraint on $\delta$. The discriminant is $\Delta = 4\pi - d$, so taking cases based on $d$, the strictest constraint we get is $\delta \leq e^{-\pi + \frac{\sqrt{4\pi - 1}}{2}} \approx 0.23$, which is for $d = 1$. Respecting this constraint, we now must identify a range of values for $\delta$ so that we have $3n\delta T \leq \frac{d}{12}$. We have:

$$3n\delta T \leq \frac{d}{12} \iff 6n\delta\sqrt{d}\sqrt{\ln\left(\frac{1}{\delta}\right) + \left(\sqrt{\ln\left(\frac{1}{\delta}\right)} + \sqrt{d}\right)^2} \leq \frac{d}{12}.$$

It holds that $\ln\left(\frac{1}{\delta}\right) + \left(\sqrt{\ln\left(\frac{1}{\delta}\right)} + \sqrt{d}\right)^2 \leq 2\left(\sqrt{\ln\left(\frac{1}{\delta}\right)} + \sqrt{d}\right)^2$, so it suffices to satisfy:

$$6\sqrt{2}n\delta\sqrt{d}\left(\sqrt{\ln\left(\frac{1}{\delta}\right)} + \sqrt{d}\right) \leq \frac{d}{12} \iff \delta\left(1 + \sqrt{\frac{\ln\left(\frac{1}{\delta}\right)}{d}}\right) \leq \frac{1}{72\sqrt{2}n}. \tag{31}$$

The previous inequality imposes a condition on $\delta$. Indeed, to ensure that $\sqrt{\frac{\ln\left(\frac{1}{\delta}\right)}{d}} \geq 0$, it is necessary that $\frac{1}{72\sqrt{2}n\delta} - 1 \geq 0 \iff \delta \leq \frac{1}{72\sqrt{2}n}$, which is stronger than $\delta \leq 0.23$.

For (31) to hold, we will show that it suffices to have:

$$\delta \leq \min \left\{ \frac{1}{144\sqrt{2}n}, \frac{\sqrt{d}}{288\sqrt{2}n\sqrt{\ln\left(\frac{144\sqrt{2}n}{\sqrt{d}}\right)}} \right\},$$

which trivially satisfies the previous constraint.

We set $\phi := \frac{\sqrt{d}}{144\sqrt{2}n}$, so the proposed range for $\delta$ becomes $\delta \leq \min\left\{\frac{\phi}{\sqrt{d}}, \frac{\phi}{2\sqrt{\ln\left(\frac{1}{\phi}\right)}}\right\}$. Additionally, (31) can be equivalently written as:

$$\delta\left(1 + \sqrt{\frac{\ln\left(\frac{1}{\delta}\right)}{d}}\right) \leq \frac{2\phi}{\sqrt{d}}. \tag{32}$$

To verify our claim, we consider the following two cases:

1. $\min\left\{\frac{\phi}{\sqrt{d}}, \frac{\phi}{2\sqrt{\ln\left(\frac{1}{\phi}\right)}}\right\} \leq e^{-d}$. We start by arguing that:

$$\min\left\{\frac{\phi}{\sqrt{d}}, \frac{\phi}{2\sqrt{\ln\left(\frac{1}{\phi}\right)}}\right\} \leq e^{-d} \iff \frac{\phi}{2\sqrt{\ln\left(\frac{1}{\phi}\right)}} \leq e^{-d}. \tag{33}$$

The $\impliedby$ direction is trivial, so we focus on the $\implies$ direction.

We assume that $\min\left\{\frac{\phi}{\sqrt{d}}, \frac{\phi}{2\sqrt{\ln\left(\frac{1}{\phi}\right)}}\right\} \leq e^{-d}$ but $\frac{\phi}{2\sqrt{\ln\left(\frac{1}{\phi}\right)}} > e^{-d}$. Then, it must be the case that $\frac{\phi}{\sqrt{d}} \leq e^{-d}$. Setting $y := \frac{1}{\phi}$, the above system of inequalities can be written in the form:

$$\begin{cases} \frac{e^{2d}}{4} > y^2 \ln(y) \\ y \geq \frac{e^d}{\sqrt{d}} \end{cases}.$$

The function $y^2 \ln(y)$ is increasing for $y \geq e^{-\frac{1}{2}}$. We have $\frac{e^d}{\sqrt{d}} > e^{-\frac{1}{2}}, \forall d \geq 1$. Thus, for $y \geq \frac{e^d}{\sqrt{d}}$, we get:

$$\frac{e^{2d}}{4} > y^2 \ln(y) \geq \frac{e^{2d}}{d}\ln\left(\frac{e^d}{\sqrt{d}}\right) \implies 3d < 2\ln(d).$$

However, this leads to a contradiction, because $3d < 2\ln(d) \leq 2d - 2 \implies d < -2$.

Consequently, we have shown (33). Since we have assumed that $\min\left\{\frac{\phi}{\sqrt{d}}, \frac{\phi}{2\sqrt{\ln\left(\frac{1}{\phi}\right)}}\right\} \leq e^{-d}$, it must be the case that $\frac{\phi}{2\sqrt{\ln\left(\frac{1}{\phi}\right)}} \leq e^{-d}$.

We now turn our attention again to (32). We remark that, for $\delta \leq e^{-d}$, we get $\sqrt{\frac{\ln\left(\frac{1}{\delta}\right)}{d}} \geq 1$. As a result, in order to satisfy (32), it suffices to have:

$$2\delta\sqrt{\frac{\ln\left(\frac{1}{\delta}\right)}{d}} \leq \frac{2\phi}{\sqrt{d}} \iff \delta\sqrt{\ln\left(\frac{1}{\delta}\right)} \leq \phi. \tag{34}$$

Observe that the function $\delta\sqrt{\ln\left(\frac{1}{\delta}\right)}$ is increasing for $\delta \leq e^{-d} < e^{-\frac{1}{2}}$. Combined with our previous remarks, this implies that, in order to verify (34) for the values of $\delta$ we picked, it

suffices to show that it holds for $\delta = \frac{\phi}{2\sqrt{\ln\left(\frac{1}{\phi}\right)}}$. So, we have to verify the inequality:

$$\left(\frac{\phi}{2\sqrt{\ln\left(\frac{1}{\phi}\right)}}\right)\sqrt{\ln\left(\frac{2\sqrt{\ln\left(\frac{1}{\phi}\right)}}{\phi}\right)} \leq \phi \iff \frac{\ln\left(2\sqrt{\ln\left(\frac{1}{\phi}\right)}\right)}{\ln\left(\frac{1}{\phi}\right)} \leq 3.$$

Setting $z := \ln\left(\frac{1}{\phi}\right)$, this becomes equivalent to $\frac{\ln\left(2\sqrt{z}\right)}{z} \leq 3$, which holds for all $z > 0 \iff \ln\left(\frac{1}{\phi}\right) > 0 \iff \phi < 1$. This last condition is satisfied by all $\phi$ such that $\frac{\phi}{2\sqrt{\ln\left(\frac{1}{\phi}\right)}} \leq e^{-d}$, so the desired result has been established.

2. $\min\left\{\frac{\phi}{\sqrt{d}}, \frac{\phi}{2\sqrt{\ln\left(\frac{1}{\phi}\right)}}\right\} > e^{-d}$. For this case, we consider two sub-cases, depending on how $\delta$ compares with $e^{-d}$. We have:

- $\delta \leq e^{-d}$. As before, we argue that it suffices to have $\delta\sqrt{\ln\left(\frac{1}{\delta}\right)} \leq \phi$. Since $\delta\sqrt{\ln\left(\frac{1}{\delta}\right)}$ is increasing for $\delta \leq e^{-d} < e^{-\frac{1}{2}}$, all we have to do is show that $e^{-d}\sqrt{d} \leq \phi$. This, however, is an immediate consequence of our assumption that $\min\left\{\frac{\phi}{\sqrt{d}}, \frac{\phi}{2\sqrt{\ln\left(\frac{1}{\phi}\right)}}\right\} > e^{-d}$.

- $\delta > e^{-d}$. This implies $\frac{\ln\left(\frac{1}{\delta}\right)}{d} < 1$. Thus, for (32) to hold, it suffices that:

$$2\delta \leq \frac{2\phi}{\sqrt{d}} \iff \delta \leq \frac{\phi}{\sqrt{d}}.$$

This is satisfied, because our proposed range for $\delta$ is $\delta \leq \min\left\{\frac{\phi}{\sqrt{d}}, \frac{\phi}{2\sqrt{\ln\left(\frac{1}{\phi}\right)}}\right\}$.

We have established that:

$$T = 2\sqrt{d}\sqrt{\ln\left(\frac{1}{\delta}\right) + \left(\sqrt{\ln\left(\frac{1}{\delta}\right)} + \sqrt{d}\right)^2} \text{ and } \delta \leq \mathcal{O}\left(\min\left\{\frac{1}{n}, \frac{\sqrt{d}}{n\sqrt{\log\left(\frac{n}{\sqrt{d}}\right)}}\right\}\right),$$

imply:

$$\delta T \geq 2\sqrt{2\pi}de^{-\frac{\left(\sqrt{T^2 - 2d^2} - \sqrt{2}d\right)^2}{8d}} \text{ and } 3n\delta T \leq \frac{d}{12},$$

while respecting all the constraints.

We substitute this to (30) and get $n \geq \frac{d}{24\alpha\varepsilon}$, which yields the desired result. ∎