# OpenReview forum: "New Lower Bounds for Private Estimation and a Generalized Fingerprinting Lemma"
_NeurIPS.cc/2022/Conference — NeurIPS 2022 Accept_

### Official Review · Reviewer_dVh8 · 2022-07-06

**Rating:** 7
**Confidence:** 3
**Soundness:** 3 good
**Presentation:** 3 good
**Contribution:** 3 good

**Summary:**

The goal of this paper is to prove lower bounds on the sample complexity needed for private statistical estimation. In the case of pure eps-DP, one can often use packing lower bounds to prove lower bounds. Lower bounds for approximate (eps, delta)-DP are often much more involved with one well-developed technique being the fingerprinting lemma. However, past works using the fingerprinting lemma required that underlying distribution be a product distribution. This left open lower bounds for some fundamental distributions such as Gaussian distributions.

The main technical contribution of this paper is a generalized fingerprinting lemma for exponential families. As an important application, they apply this new tool to derive Omega(d^2/alpha * eps) lower bound on the sample complexity needed to learn a single d-dimensional Gaussian, thus closing the gap for this distribution. As a second application, they provide a lower bound for mean-estimation of distributions with bounded second moments.

**Questions:**

I only have a couple of rather minor comments.
- The paragraph around line 146 and line 164 has some repetition w.r.t. discussion about T. Also T is not defined here and it is only defined in the appendix.
- Line 154: The second line is confusing to me: the authors say they assume the existence of vectors eta^1, eta^2 but for what purpose? What properties do these vectors need to satisfy? Is the eta drawn from some distribution that is supported between these two vectors?

**Limitations:**

Limitations discussion is fine.

**Strengths And Weaknesses:**

**Strengths.**
This is a very fundamental problem and the literature was clearly missing a key tool that would allow us to prove optimal results in this area. This paper provided an important contribution in this area by giving us a new tool (based on fingerprinting) which allowed us to answer some fundamental questions in this area. Personally, I would like to dive further and learn more about the techniques in this paper.

**Weaknesses.**
No concrete weaknesses.

---

> ### Author Response · Authors · 2022-08-02
> **Response to Reviewer dVh8**
>
> We thank the reviewer for their time and positive evaluation of our work. Regarding the questions raised, we have the following comments.
>
> >The paragraph around line 146 and line 164 has some repetition w.r.t. discussion about T. Also T is not defined here and it is only defined in the appendix.
>
> $T$ is a function associated with exponential families, known as the ``sufficient statistics’’ of the family. We omitted its definition (as well as the definition of exponential families) from the main body due to space limitations. In a future update to the manuscript, we will try our best to move some of the material from the appendix to the main body and improve the clarity of the presentation.
>
> >Line 154: The second line is confusing to me: the authors say they assume the existence of vectors eta^1, eta^2 but for what purpose? What properties do these vectors need to satisfy? Is the eta drawn from some distribution that is supported between these two vectors?
>
> The reviewer is correct: $\eta$ is drawn uniformly from the box defined by opposite corners $\eta^1$ and $\eta^2$. In more detail: most arguments for proving lower bounds in these settings take a common form. The algorithm is uncertain about the value of some parameter of interest, and is given data from a distribution where the parameter is selected randomly from this uncertainty set. It is safe to assume that the algorithm is aware of this uncertainty set, and what distribution the natural parameter follows over this set. The vectors $\eta^1$ and $\eta^2$ define this uncertainty set: independently for each coordinate $j$, we let the $j$-th coordinate of the natural parameter vector $\eta$ be drawn uniformly at random from the interval $[\eta^1_j, \eta^2_j]$.
> The only necessary condition is that the Cartesian product of the previous intervals must be a subset of the range of natural parameters of the exponential family (again, defined in the appendix). However, in order to obtain strong lower bounds, one must choose $\eta^1$ and $\eta^2$ appropriately on a problem-by-problem basis.

---

> > ### Comment · Reviewer_dVh8 · 2022-08-08
> > **response**
> >
> > Thanks to the authors for addressing my questions. I have no other questions or concerns for now.

---

### Official Review · Reviewer_eSVJ · 2022-07-07

**Rating:** 6
**Confidence:** 3
**Soundness:** 3 good
**Presentation:** 3 good
**Contribution:** 3 good

**Summary:**

This paper provides new tight lower bounds for private estimating the covariance matrix of a Gaussian and private mean estimation of heavy-tailed distributions over $\mathbb{R}^d$.
Previous results gave similar lower bounds under simplified assumptions (e.g., pure DP or product distributions). This paper fills a number of gaps by presenting lower bounds for non-product distributions under approximate DP.
Most of the presentation is devoted to the former lower bound (covariance matrix estimation), which is achieved using a new generalized “Fingerprinting Lemma” for exponential families.

The fingerprinting lemma [Bun, Steinke, Ullman, SODA 17] gives a way to sample n codewords in dimension $d=O(n^2)$ such that every algorithm that estimates the average of the points “too well”, must be “too correlated” with one of the codewords, and thus cannot be private.
The main bottleneck of the above approach is that the codewords are sampled from product distribution, and therefore provide lower bounds only for that cases.

This paper provides a new FP lemma (Lemma 2.2) that extends the original one to exponential families which capture a wider range of distributions. Moreover, it does not focus on estimating the average of the points, but on estimating the “natural parameter vector” $\eta$ of the family, which captures the parameters of a specific family (E.g., if the family is all the Gaussians of the form $N(0,\Sigma)$, then $\eta$ is essentially a vector representation of the covariance matrix $\Sigma$).

The new FP lemma is used to prove a new lower bound for privately estimating the natural parameter vector of a distribution from a general exponential family (Thm 2.5).
Applying Thm 2.5 to the problem of estimating the covariance matrix yields (with some additional work) a tight $\tilde{\Omega}(d^2)$ lower bound for estimating the covariance matrix in Frobenius norm (Thm 3.1). The latter is then used for proving a tight lower bound of $\tilde{\Omega}(d^1.5)$ for estimating the covariance matrix in the Spectral norm.

Finally, in section 4 (page 9), the $\tilde{\Omega}(d)$ lower bound for private mean estimation of heavy-tailed distributions is given, which is based on a different known method of Assouad (rather than fingerprinting).


**Questions:**

1) In lines 56-59, you mention that the lower bound in Frobenius norm (Thm 1.1) matches the non-private sample complexity, while the lower bound in Spectral norm (Thm 1.2) does not (there is a $\sqrt{d}$ gap). First, can you explain (or provide reference) why the sample complexity of estimating the covariance matrix in spectral norm without privacy is $O(d)$?
Second, how can it be that Thm 1.1 is tight (w.r.t. d) also for non-private algorithms while Thm 1.2 is not? I’m asking because Thm 1.2 is proven directly from Thm 1.1, so why the same arguments do not follow in the non-private case (providing a non-private analog of Thm 1.2)? What am I missing?

2) Can you please explain how Thm 1.3 is different from other known lower bounds for DP mean estimation? For example, [Kamath, Li, Singhal, Ullman, COLT 19] proved that an approximate DP algorithm that accurately estimates the mean of a Gaussian $N(\mu,I_{dxd})$ must incur an additive error of $\tilde{\Omega}(d)$. Since $N(\mu,I_{dxd})$ has a second moment bounded by 1, it seems stronger than your statement. Is that correct?

Minor issues/typoes:
1) In the second equality after line 192: should $\eta_j$ be t?
2) In line 203: change Lemma 2.2 to Lemma 2.1.
3) In Lemma 2.3: I think that $s$ is not defined at this point (only later).


**Limitations:**

In the “Conclusion and Open Problems” section (section 5), the authors address the limitations of their results.

**Strengths And Weaknesses:**

Strengths:
1) First tight lower bounds for the important problem of estimating the covariance matrix of a Gaussian under approximate DP.
2) Interesting FP lemma for exponential families that can find other applications.
3) New lower bound for mean estimation using Assouad’s method.
4) Overall, good writing quality, well-organized, and looks sound (all missing proofs are given in the appendix).

Weaknesses:
1) The proof of the FP lemma for exponential families, which seems like the main technical contribution of this paper, is achieved using a natural extension of the proof of the original FP lemma [Bun, Steinke, Ullman, SODA 17]. This extension is definitely not straightforward and requires some technical work, but I don’t see something conceptually new in the proof technique. This similarity is currently not mentioned at all. It should be emphasized in the paper for making the presentation fairer, and for helping the reader to understand all the long calculations (which look much nicer in the simpler setting of [BSU17]).
2) A comparison of the mean estimation lower bound (Thm 1.3) with previous lower bounds is missing. It is not clear to me if the statement itself is new or only the proof technique (Assouad’s method rather than FP lemma). See my second question in the next part.

Overall, the strengths (especially the end results for covariance matrix estimation) seem to outweigh the weaknesses, and therefore I lean towards accepting this paper.

---

> ### Author Response · Authors · 2022-08-02
> **Response to Reviewer eSVJ**
>
> We thank the reviewer for their time and helpful feedback. We address their comments here.
>
> >The proof of the FP lemma for exponential families, which seems like the main technical contribution of this paper, is achieved using a natural extension of the proof of the original FP lemma [Bun, Steinke, Ullman, SODA 17]. This extension is definitely not straightforward and requires some technical work, but I don’t see something conceptually new in the proof technique. This similarity is currently not mentioned at all. It should be emphasized in the paper for making the presentation fairer, and for helping the reader to understand all the long calculations (which look much nicer in the simpler setting of [BSU17]).
>
> There were various challenges involved in proving our results. The first one has to do with the fact that it was not clear a-priori that using fingerprinting was the ``right’’ way to get a lower bound for covariance estimation.
> The second has to do with the fact that, even after identifying the appropriate approach, coming up with the correct formulation for our fingerprinting result was not obvious. All previous results focus on the problem of mean estimation, whereas we argue that focus should be moved from mean estimation and towards estimating the parameter vector of exponential families. Additionally, in fingerprinting-style proofs, prior works considered the trade-off between accuracy and privacy by directly comparing the output of the estimator with the input samples, while in our lemma the samples are replaced by the corresponding sufficient statistics. Moreover, our choice to have the estimator estimate the deviation of the parameter vector from a point rather than the vector itself was not obvious. All the aforementioned issues are not reflected when looking at the final result itself, but they were crucial and non-trivial during the process of formulating and proving the statement.
> The last challenge involves appropriately leveraging the properties of exponential families at crucial points when proving the intermediate lemmas that build towards our main result. In conclusion, while we agree that the basic steps of the proof follow a structure that is similar to that in prior fingerprinting-style results, we argue that there were a number of obstacles which necessitated thinking out of the box at some important stages.
> That said, we would like to emphasize that we did acknowledge the similarities with the prior work in our manuscript, and tried our best to highlight the differences between the prior work and ours, while staying within the page limit. For example, in lines 215-218, we acknowledge the similarities between the proof of lemma 2.3 and similar results in prior work. More broadly, we believe that section 2.1 also identifies the conceptual similarities between our technique and previous results, while also pointing out the differences. For the final version, we could add a more detailed discussion about the ideas that were similar to or different from the prior work.
>
> >A comparison of the mean estimation lower bound (Thm 1.3) with previous lower bounds is missing. It is not clear to me if the statement itself is new or only the proof technique (Assouad’s method rather than FP lemma). See my second question in the next part.
>
> Here, we address both Weakness 2 and Question 2. The novelty in the lower bound of Theorem 1.3 lies primarily in the result/statement. We compare with two prior lower bounds:
> 1) [4] prove an $(\varepsilon,\delta)$-DP lower bound for estimating the mean of a Gaussian distribution $N(\mu,\mathbb{I})$. They show a sample complexity lower bound of $\Omega(d/\varepsilon\alpha)$, which is specific to Gaussians. Our lower bound applies to a broader class of distributions – those with bounded second moment. We show that this broader class is harder to estimate, requiring $\Omega(d/\varepsilon\alpha^2)$ samples.
> 2) [5, 6] prove an $(\varepsilon,0)$-DP lower bound for estimating the mean of any distribution with bounded second moments. They show a sample complexity lower bound of $\Omega(d/\varepsilon\alpha^2)$. We show the same sample complexity lower bound for the same problem under the weaker constraint of $(\varepsilon,\delta)$-DP.
>
> We included discussion of 2) in lines 78-83, and will add a comparison there with 1) as well.

---

> > ### Author Response · Authors · 2022-08-02
> > **Response to Reviewer eSVJ (Continued)**
> >
> > >In lines 56-59, you mention that the lower bound in Frobenius norm (Thm 1.1) matches the non-private sample complexity, while the lower bound in Spectral norm (Thm 1.2) does not (there is a $\sqrt{d}$ gap). First, can you explain (or provide reference) why the sample complexity of estimating the covariance matrix in spectral norm without privacy is $O(d)$? Second, how can it be that Thm 1.1 is tight (w.r.t. d) also for non-private algorithms while Thm 1.2 is not? I’m asking because Thm 1.2 is proven directly from Thm 1.1, so why the same arguments do not follow in the non-private case (providing a non-private analog of Thm 1.2)? What am I missing?>
> >
> > First, regarding the reference about the sample complexity of non-private spectral estimation of Gaussian covariances, we refer the reviewer to Example 6.3 in [1]. However, under privacy constraints, all known algorithms required $\Omega(d^{3/2})$ (see the paragraph "The Covariance Estimation Bottleneck" in [2] for a discussion about this). A matching lower bound was known only for worst-case data and it was an open problem to show this for a natural class of distributions (like Gaussians). Our work is the first to show this lower bound, thus verifying the existence of a gap between the non-private and private sample complexities for this problem. Conversely, such a gap does not exist when comparing the private and non-private sample complexities of Frobenius/Mahalanobis estimation of Gaussian covariances. Indeed, the non-private sample complexity for that problem is $\widetilde{\Theta}(d^2)$, which is also the sample complexity attained by the algorithm in [3] (which, in turn, nearly matches our LB in Theorem 3.1). Consequently, one can get privacy ``for free” for Frobenius/Mahalanobis estimation, but not for spectral estimation where, for $\varepsilon \le 1, \alpha = \mathcal{O}(1/\sqrt{d})$, the cost increases by a dimension-dependent factor. We stress that the assumption $\varepsilon \le 1$ is crucial, since it appears in line 595. Having a larger $\varepsilon$ would imply that we are in the low privacy regime, in which case the sample complexities end up being dominated by the non-private terms ($d^2/\alpha^2$ and $d/\alpha^2$ for Frobenius and spectral estimation, respectively), which correspond to the case where $\varepsilon = \infty$.
> > Finally, we point out that our reduction for spectral estimation would give the appropriate lower bound for the non-private setting, as well ($\Omega(d/\alpha^2)$). The lower bound for Frobenius estimation without privacy is $\Omega(d^2/\alpha^2)$. Therefore, the same step (as in our reduction for DP spectral estimation) of setting $\alpha$ to $\alpha \sqrt{d}$ would yield the aforementioned lower bound.
> >
> > >Minor issues/typos
> >
> > The reviewer is correct regarding all the three typos they have identified. We will update the manuscript accordingly.
> >
> > **References:**
> > [1] Martin J. Wainwright. High-Dimensional Statistics: A Non-Asymptotic Viewpoint. Cambridge University Press, 2019.
> > [2] Gavin Brown, Marco Gaboardi, Adam Smith, Jonathan Ullman, and Lydia Zakynthinou. Covariance-aware private mean estimation without private covariance estimation. arXiv preprint arXiv:2106.13329, 2021.
> > [3] Hassan Ashtiani and Christopher Liaw. Private and polynomial time algorithms for learning Gaussians and beyond. arXiv preprint arXiv:2111.11320, 2021.
> > [4] Gautam Kamath, Jerry Li, Vikrant Singhal, and Jonathan Ullman. Privately learning high dimensional distributions. In Proceedings of the 32nd Annual Conference on Learning Theory, COLT ’19, pages 1853–1902, 2019.
> > [5] Rina Foygel Barber and John C Duchi. Privacy and statistical risk: Formalisms and minimax bounds. arXiv preprint arXiv:1412.4451, 2014.
> > [6] Gautam Kamath, Vikrant Singhal, and Jonathan Ullman. Private mean estimation of heavy-tailed distributions. In Proceedings of the 33rd Annual Conference on Learning Theory, COLT ’20, pages 2204–2235, 2020.

---

> > > ### Comment · Reviewer_eSVJ · 2022-08-06
> > > **response**
> > >
> > > Thanks for the detailed explanation.

---

> > ### Comment · Reviewer_eSVJ · 2022-08-06
> > **response**
> >
> > I understand the challenges you mentioned, and I have no doubt that your extension of the FP lemma to exponential families is completely non-trivial. Yet, you cannot ignore the fact that the structure of the proof of your Lemma 2.1 is very similar to the structure of the proof of Lemma 3.6 in BSU17 https://arxiv.org/pdf/1604.04618 (Appendix A). For instance, your $g_j(\eta)$ is analog to their $g(p)$, and your $Z^j$ is analog to their $f(x) \sum_i (x_i-p)$. This similarity, which in my opinion, is important for understanding the proof of your fingerprinting lemma, is currently not mentioned at all. As a reader, it helped me a lot to understand your proof steps because I also read the proof steps of BSU17 in their much simpler setting. You have a good result, and there is no reason to hide this similarity unless you think I'm wrong. The prior work you mentioned in lines 215-218 is about Lemma 2.3 which is something else.
> >
> > Regarding your lower bound for averaging, thanks for clarifying it to me, but I am still missing something.
> > You say that your lower (Thm 4.1) applies for any distribution with second moment bounded by 1, but in your proof you only focus on a specific distribution family $\set{D_v}$. Why is it ok? (i.e., why does this imply a lower bound for any distribution with second moment bounded by 1?)

---

> > > ### Author Response · Authors · 2022-08-07
> > > **Response**
> > >
> > > >I understand the challenges you mentioned, and I have no doubt that your extension of the FP lemma to exponential families is completely non-trivial. Yet, you cannot ignore the fact that the structure of the proof of your Lemma 2.1 is very similar to the structure of the proof of Lemma 3.6 in BSU17 https://arxiv.org/pdf/1604.04618 (Appendix A). For instance, your $g_j(\eta)$ is analog to their $g(p)$, and your $Z^j$ is analog to their $f(x)\sum_i(x_i-p)$. This similarity, which in my opinion, is important for understanding the proof of your fingerprinting lemma, is currently not mentioned at all. As a reader, it helped me a lot to understand your proof steps because I also read the proof steps of BSU17 in their much simpler setting. You have a good result, and there is no reason to hide this similarity unless you think I'm wrong. The prior work you mentioned in lines 215-218 is about Lemma 2.3 which is something else.
> > >
> > > We thank the reviewer for their positive and constructive comments about our result. The reviewer is correct in the comparison they are drawing between the proof of Lemma 2.1 and the corresponding lemma from [BSU17]. We would like to note that [BSU17] is not the only paper to feature a statement and proof of this kind (e.g., see Lemmata 6.3 and 6.8 from [1], with a more detailed list of the main works on fingerprinting being in the introduction). Having stated these prior works in the introduction, along with the technical differences in Section 2.1, for brevity, we initially chose to not include a pointer to [BSU17] (or to any other work featuring a fingerprinting lemma) in the technical sections. We will add references to these works in the technical sections in the final version to enable better understanding for the readers.
> > >
> > > >Regarding your lower bound for averaging, thanks for clarifying it to me, but I am still missing something. You say that your lower (Thm 4.1) applies for any distribution with second moment bounded by 1, but in your proof you only focus on a specific distribution family {$D_v$}. Why is it ok? (i.e., why does this imply a lower bound for any distribution with second moment bounded by 1?)
> > >
> > > The reason why we focus on a specific hard instance is that the lower bounds we prove in the paper are those in the *minimax sense*. In particular, when estimating a parameter $\theta$ for a class of distributions $\mathcal{P}$ with respect to a loss function $\ell$ (here, the error of the estimator), we define the minimax risk of the problem as the expected value of $\ell$ when the "best" estimator "competes'' with the "hardest" distribution in the class. Thus, if there exists a subset of the family of distributions considered which is “hard to estimate”, then the lower bound holds. The formal definition for this is given in Lines 535-540 of our submission (also see Chapter 7 of [2] for more information). Thus, the correct way to interpret the phrasing of Theorem 4.1 is "there exist hard distributions in the class of distributions with bounded second moments, such that no matter how good an $(\varepsilon, \delta)$-DP estimator is, it will need at least $\Omega(d/(\alpha^2 \varepsilon))$ samples to have MSE less than $\alpha^2$".
> > >
> > > **References:**
> > > [1] Gautam Kamath, Jerry Li, Vikrant Singhal, and Jonathan Ullman. Privately learning high dimensional distributions. In Proceedings of the 32nd Annual Conference on Learning Theory, COLT ’19, pages 1853–1902, 2019.
> > > [2] John Duchi. Lecture Notes for Statistics 311/Electrical Engineering 377. https://web.stanford.edu/class/stats311/lecture-notes.pdf

---

### Official Review · Reviewer_KYiN · 2022-07-12

**Rating:** 6
**Confidence:** 2
**Soundness:** 3 good
**Presentation:** 3 good
**Contribution:** 3 good

**Summary:**

This paper generalizes the fingerprinting method of BUV14 to exponential families, and uses this result to prove lower bounds on approximate DP Gaussian covariance estimation and heavy-tailed mean estimation.

**Questions:**

Heavy-tailed mean estimation:
-Can you clarify what you mean by second moment bounded by 1? (e.g. coordinate-wise? central??) Also, the work https://arxiv.org/pdf/2106.01336.pdf gave a lower bound for strongly convex DP SCO (under zCDP -- not quite approximate DP) that seems to essentially be a reduction to mean estimation. So is your main contribution on this problem to be a strengthening of their construction from zCDP to approximate DP? Or am I misunderstanding something here?


**Limitations:**

Limitations - yes
Societal impacts - not addressed

**Strengths And Weaknesses:**

Strengths:

-The results are very interesting and seem to expand the DP lower bound toolbox.

Weaknesses:

-Insufficient clarity on the distinction between the proposed method and existing methods. It is explained at a high level in the introduction, but still not as clear or precise as I would like. For example: I don't think it's made clear enough in what sense you generalize the fingerprinting method. Your method applies to exponential families, but what class did the method of prior works apply to? Also, a comparison/discussion of your lemmas and theorems with corresponding results in prior work would be very helpful.

-Related to the above, I'm not sure I understand the main challenge in proving your results and how your proof approach/techniques differs from prior work.

-Minor: Notation with $\eta^i$ is a bit confusing at first glance because it looks like you're raising a vector to the power $i$. Subscript or parenthetical superscript would be clearer.

---

> ### Author Response · Authors · 2022-08-02
> **Response to Reviewer KYiN**
>
> We thank the reviewer for their time and thoughtful comments. We address their questions here.
>
> >Insufficient clarity on the distinction between the proposed method and existing methods. It is explained at a high level in the introduction, but still not as clear or precise as I would like. For example: I don't think it's made clear enough in what sense you generalize the fingerprinting method. Your method applies to exponential families, but what class did the method of prior works apply to? Also, a comparison/discussion of your lemmas and theorems with corresponding results in prior work would be very helpful.
>
> When we say that we generalize the fingerprinting technique, we mean that we formulate and prove a fingerprinting-style result for a very large and useful family of distributions (see the table in [1] for a list of the basic exponential families), which covers prior results (see Appendix G), and provides a way to prove lower bounds for DP estimation of different parameters of many other useful distributions. Older fingerprinting-style arguments constructed hard instances for problems (e.g. query release) by resorting either to binary product distributions or Gaussians with independent marginals. Both of these distributions are instances of exponential families, but they constitute a very small subset of what can be expressed as exponential families, with the need for independent marginals being especially restrictive (it was the main obstacle to obtaining a lower bound for Gaussian covariance estimation under $(\varepsilon,\delta)$-DP, which was an open question going back to [9]). Thus, we believe that we significantly contributed to strengthening the toolbox of lower bounds for differential privacy. Indeed, the majority of lower bounds under approx-DP were applications fingerprinting. The main papers that used fingerprinting include [2-6] (which use binary product distributions) and [7-9] (which consider both binary product distributions and Gaussians with independent marginals).
>
> >Related to the above, I'm not sure I understand the main challenge in proving your results and how your proof approach/techniques differs from prior work.
>
> There were various challenges involved in proving our results. The first has to do with the fact that it was not clear a-priori that using fingerprinting was the ``right’’ way to get a lower bound for covariance estimation.
> The second has to do with the fact that, even after identifying the appropriate approach, coming up with the correct formulation for our fingerprinting result was not obvious. All previous results focus on the problem of mean estimation, whereas we argue that focus should be moved from mean estimation and towards estimating the parameter vector of exponential families. Additionally, in fingerprinting-style proofs, prior works considered the trade-off between accuracy and privacy by directly comparing the output of the estimator with the input samples, while in our lemma the samples are replaced by the corresponding sufficient statistics. Moreover, our choice to have the estimator estimate the deviation of the parameter vector from a point rather than the vector itself was not obvious. All the aforementioned issues are not reflected when looking at the final result itself, but they were crucial and non-trivial during the process of formulating and proving the statement.
> The last challenge involves appropriately leveraging the properties of exponential families at crucial points when proving the intermediate lemmas that build towards our main result. In conclusion, while we agree that the basic steps of the proof follow a structure that is similar to that in prior fingerprinting-style results, we argue that there were a number of obstacles which necessitated thinking out of the box at some important stages.

---

> > ### Author Response · Authors · 2022-08-02
> > **Response to Reviewer KYiN (Continued)**
> >
> > >Minor: Notation with $\eta^i$ is a bit confusing at first glance because it looks like you're raising a vector to the power $i$. Subscript or parenthetical superscript would be clearer.
> >
> > We will switch to parenthetical superscripts for $\eta^i$ to make the notation clearer.
> >
> > >Heavy-tailed mean estimation: -Can you clarify what you mean by second moment bounded by 1? (e.g. coordinate-wise? central??) Also, the work https://arxiv.org/pdf/2106.01336.pdf gave a lower bound for strongly convex DP SCO (under zCDP -- not quite approximate DP) that seems to essentially be a reduction to mean estimation. So is your main contribution on this problem to be a strengthening of their construction from zCDP to approximate DP? Or am I misunderstanding something here?
> >
> > Regarding [10], the result given in that paper assumes that coordinate-wise moments are upper-bounded by $1$. On the other hand, our work assumes that the stronger moment assumption that the second moment of *any* single-dimensional projection is upper-bounded by $1$, under which it is more challenging to prove lower bounds. This is clarified in lines 506-507 (in the appendix about the preliminaries). We remark that this is the same moment assumption used in the works cited in line 79, all of which involve results that hold under pure differential privacy.
> >
> > >Limitations - yes Societal impacts - not addressed
> >
> > Finally, regarding the fact that the reviewer claimed that we didn't address the societal implications of our work, we would like to point out that our submission is about lower bounds and is purely theoretical. For that reason, we do not expect it to have any obvious societal impact.
> >
> > **References:**
> > [1] Wikipedia, Exponential Family, Table of Distributions https://en.wikipedia.org/wiki/Exponential_family#Table_of_distributions
> > [2] Mark Bun, Jonathan Ullman, and Salil Vadhan. Fingerprinting codes and the price of approximate differential privacy. In Proceedings of the 46th Annual ACM Symposium on the Theory of Computing, STOC ’14, pages 1–10, New York, NY, USA, 2014. ACM.
> > [3] Raef Bassily, Adam Smith, and Abhradeep Thakurta. Private empirical risk minimization: Efficient algorithms and tight error bounds. In Proceedings of the 55th Annual IEEE Symposium on Foundations of Computer Science, FOCS ’14, pages 464–473, Washington, DC, USA, 2014. IEEE 363 Computer Society.
> > [4] Thomas Steinke and Jonathan Ullman. Interactive fingerprinting codes and the hardness of preventing false discovery. In Proceedings of the 28th Annual Conference on Learning Theory, COLT ’15, pages 1588–1628, 2015.
> > [5] Thomas Steinke and Jonathan Ullman. Between pure and approximate differential privacy. The Journal of Privacy and Confidentiality, 7(2):3–22, 2017.
> > [6] Thomas Steinke and Jonathan Ullman. Tight lower bounds for differentially private selection. In Proceedings of the 58th Annual IEEE Symposium on Foundations of Computer Science, FOCS ’17, pages 552–563, Washington, DC, USA, 2017. IEEE Computer Society.
> > [7] Cynthia Dwork, Adam Smith, Thomas Steinke, Jonathan Ullman, and Salil Vadhan. Robust traceability from trace amounts. In Proceedings of the 56th Annual IEEE Symposium on Foundations of Computer Science, FOCS ’15, pages 650–669, Washington, DC, USA, 2015. IEEE Computer 394 Society.
> > [8] Gautam Kamath, Jerry Li, Vikrant Singhal, and Jonathan Ullman. Privately learning high dimensional distributions. In Proceedings of the 32nd Annual Conference on Learning Theory, COLT ’19, pages 1853–1902, 2019.
> > [9] T. Tony Cai, Yichen Wang, and Linjun Zhang. The cost of privacy: Optimal rates of convergence for parameter estimation with differential privacy. arXiv preprint arXiv:1902.04495, 2019.
> > [10] Gautam Kamath, Xingtu Liu, and Huanyu Zhang. Improved rates for differentially private stochastic convex optimization with heavy-tailed data. arXiv preprint arXiv:2106.01336, 2021.

---

### Meta-Review · Area_Chair_vuKj · 2022-08-29

**Recommendation:** Accept
**Confidence:** Certain

**Metareview:**

This paper establishes improved and near-optimal lower bounds for private statistical estimation, specifically for private covariance estimation of a Gaussian and heavy-tailed mean estimation. The first result leverages a novel technical result, proved in this paper: a generalization of the fingerprint lemma (Bun, Steinke, Ullman' 17) to exponential families. The second result relies on a private version of Assouad's lemma (developed in recent work). The reviewers agreed that this is a technically novel and interesting work that clearly merits acceptance.

**Award:**

No

---

### Decision · Program_Chairs · 2022-09-14

Accept